# Progressive plasticity during colorectal cancer metastasis

Andrew Moorman[1,13], Elizabeth K. Benitez[2,3,13], Francesco Cambulli[2,10,13], Qingwen Jiang[2], Ahmed Mahmoud[2,4], Melissa Lumish[2,5,11], Saskia Hartner[2], Sasha Balkaran[2], Jonathan Bermeo[2], Simran Asawa[2], Canan Firat[6], Asha Saxena[5], Fan Wu[7], Anisha Luthra[7], Cassandra Burdziak[1], Yubin Xie[1,8], Valeria Sgambati[2], Kathleen Luckett[2,3], Yanyun Li[6,12], Zhifan Yi[6], Ignas Masilionis[1], Kevin Soares[7], Emmanouil Pappou[7], Rona Yaeger[5], T. Peter Kingham[7], William Jarnagin[7], Philip B. Paty[7], Martin R. Weiser[7], Linas Mazutis[1], Michael D'Angelica[7], Jinru Shia[6], Julio Garcia-Aguilar[7], Tal Nawy[1], Travis J. Hollmann[6,12], Ronan Chaligné[1], Francisco Sanchez-Vega[7], Roshan Sharma[1], Dana Pe'er[1,9 ✉] & Karuna Ganesh[2,5 ✉]

As cancers progress, they become increasingly aggressive—metastatic tumours are less responsive to first-line therapies than primary tumours, they acquire resistance to successive therapies and eventually cause death[1,2]. Mutations are largely conserved between primary and metastatic tumours from the same patients, suggesting that non-genetic phenotypic plasticity has a major role in cancer progression and therapy resistance[3–5]. However, we lack an understanding of metastatic cell states and the mechanisms by which they transition. Here, in a cohort of biospecimen trios from same-patient normal colon, primary and metastatic colorectal cancer, we show that, although primary tumours largely adopt LGR5+ intestinal stem-like states, metastases display progressive plasticity. Cancer cells lose intestinal cell identities and reprogram into a highly conserved fetal progenitor state before undergoing non-canonical differentiation into divergent squamous and neuroendocrine-like states, a process that is exacerbated in metastasis and by chemotherapy and is associated with poor patient survival. Using matched patient-derived organoids, we demonstrate that metastatic cells exhibit greater cell-autonomous multilineage differentiation potential in response to microenvironment cues compared with their intestinal lineage-restricted primary tumour counterparts. We identify *PROX1* as a repressor of non-intestinal lineage in the fetal progenitor state, and show that downregulation of *PROX1* licenses non-canonical reprogramming.

As surgical resection of metastases is uncommon, cancer cell-state transitions during tumour progression in patients remain largely unresolved. Colorectal cancer (CRC), for which the simultaneous resection of matched trios of primary tumour, normal colon and metastasis (frequently from liver) is a standard of care, offers a unique opportunity to study phenotypic transitions. While CRC largely initiates from LGR5+ intestinal stem cells (ISCs)[6,7], metastasis-initiating cells at the invasion front enter an LGR5low (refs. 8,9), L1CAM+ (ref. 10) tumour regenerative state. However, the phenotypic states, trajectories and dependencies of macrometastases emerging from L1CAM+ progenitors remain poorly characterized despite their paramount clinical importance.

To study metastatic progression in patients, we prospectively collected matched trios of normal colon, primary colorectal cancer and metastatic tissue from a cohort of 31 patients undergoing synchronous hemicolectomy and metastasectomy, including both treatment-naive patients and those who received pre-operative chemotherapy. Using single-cell RNA-seq (scRNA-seq), multiplexed immunofluorescence and epithelial organoid generation from matched trios, we found that CRC progression involves three distinct, ordered cell-state transitions: (1) from differentiated intestinal states in normal colon to an *LGR5+* ISC-like state enriched in primary tumour; (2) developmental reprogramming to a highly plastic progenitor-like fetal state associated with epithelial injury; and (3) expression of non-intestinal lineage gene programs, including squamous and neuroendocrine, which are enriched in metastases. Organoids derived from profiled trios reveal that metastatic cells possess greater cell-intrinsic plasticity in vitro relative to primary tumour cells from the same patient, allowing them to adapt to distinct microenvironments in the colon and liver in vivo.

[1]Computational and Systems Biology Program, Memorial Sloan Kettering Cancer Center, New York, NY, USA. [2]Molecular Pharmacology Program, Memorial Sloan Kettering Cancer Center, New York, NY, USA. [3]Weill Cornell/Rockefeller/Sloan Kettering Tri-Institutional MD-PhD Program, New York, NY, USA. [4]Pharmacology Program, Weill Cornell Graduate School, New York, NY, USA. [5]Department of Medicine, Memorial Sloan Kettering Cancer Center, New York, NY, USA. [6]Department of Pathology and Laboratory Medicine, Memorial Sloan Kettering Cancer Center, New York, NY, USA. [7]Department of Surgery, Memorial Sloan Kettering Cancer Center, New York, NY, USA. [8]Tri-Institutional PhD Program in Computational Biology and Medicine, New York, NY, USA. [9]Howard Hughes Medical Institute, Chevy Chase, MD, USA. [10]Present address: New York Genome Center, New York, NY, USA. [11]Present address: Case Western Reserve University, Cleveland, OH, USA. [12]Present address: Bristol Myers Squibb, Princeton, NJ, USA. [13]These authors contributed equally: A. Moorman, E. K. Benitez, F. Cambulli. ✉e-mail: peerd@mskcc.org; ganeshk@mskcc.org

We find that the transcriptional repressor *PROX1* is coordinately induced with the fetal progenitor state across multiple patients and functions to repress non-intestinal lineage genes. Loss of *PROX1*-dependent lineage restriction during tumour progression licenses differentiation into non-canonical lineages. Together, our data support a two-stage model of metastatic plasticity, whereby metastasis promotes highly plastic cell states that can be induced to differentiate along diverse trajectories by cues from the tumour microenvironment.

We collected primary CRC, adjacent normal colon and metastasis tissue (peritoneum, lung, chest wall and 29 liver specimens) from 31 patients with microsatellite-stable mismatch-repair-proficient (MSS/pMMR) CRC undergoing synchronous resection of colorectal tumour and metastasectomy at Memorial Sloan Kettering (MSK; Fig. 1a, Extended Data Fig. 1, Supplementary Fig. 1 and Supplementary Table 1). Nine patients received no treatment, while 22 patients received 5-fluorouracil-based chemotherapy before surgery. We performed scRNA-seq (83 samples from 31 patients), derived organoids from the single-cell suspension used for scRNA-seq (29 samples from 15 patients) and performed multiplex immunofluorescence when tissue was available (72 samples from 21 patients). Additional tissue was collected from a second, metachronous metastasectomy for six patients. Our scRNA-seq data captured 47,437 high-quality epithelial cell profiles after data processing (Methods). We clustered[11] epithelial cells, used InferCNV[12] to distinguish cancer from normal cells and identified expected intestinal cell types (Extended Data Fig. 2).

## Decreased ISC program in CRC metastases

The untreated patient trios provide a unique opportunity to characterize tumour progression to metastasis without the confounding influence of therapy. We therefore restricted our initial analysis to the nine untreated patients, comprising 13,935 cells (Extended Data Fig. 3a). Principal component analysis (PCA) of tumour epithelial cells from these untreated samples revealed that the strongest axis of variation (the first PC) corresponds to ISC signatures (false-discovery rate (FDR) $q < 0.04$; Supplementary Table 2), consistent with observations that primary tumours contain higher fractions of *LGR5*-expressing cells[13,14]. Using a de novo ISC signature based our untreated normal colon data (Extended Data Fig. 3b,c and Supplementary Table 2), we found that tumours express high levels of the ISC signature relative to differentiated intestinal cells; however, PC1 represents a distinct trend of gene programs that further increase in cancer cells relative to ISCs, including WNT-signalling genes (*LGR5*, *EPHB2*, *ASCL2*, *TCF7*), embryonic developmental genes (*BMP7*, *SOX4*, *CYP2W1*) and stress-response genes (*UPR1*, *MTORC1*) (Extended Data Fig. 3d–f and Supplementary Table 3).

While normal cells display strict cell-type-specific gene expression as expected (Extended Data Fig. 3g and Supplementary Table 3), untreated cancer cells co-express absorptive and secretory intestinal cell type programs, as well as ISC-specific genes in the same cells (Extended Data Fig. 3g–j). This considerable lineage promiscuity indicates dysregulation of physiological intestinal hierarchies and gain of tumour-specific programs in CRC[15], consistent with observations in other cancers[11,16]. Expanding our analysis to the full cohort of 31 patients and 47,437 cells revealed that, in contrast to the ISC dependency described in mouse CRC metastases[8,17], human metastases express the ISC program at lower levels than primary tumours—particularly in chemotherapy-treated tumours (Extended Data Fig. 4). Together, our patient data show that untreated CRC tumours are enriched in ISC-like programs with primitive developmental and mixed lineage features, while ISC programs are depleted in metastatic tumours.

## Non-canonical programs in metastases

The weak within-cell-type gene correlation structure in cancer cells (Extended Data Fig. 3g) reflects substantial dysregulation, hindering

standard annotation approaches and motivating an unsupervised search for modules of covarying genes (gene programs). We assume that covarying gene expression reflects coordinated gene regulation needed for biological function[18,19], and that expression in cancer is highly context dependent—varying by patient, local environment and other factors. We therefore used Hotspot[20] to search for modules of genes that covary only within cell subsets representing salient cell states. Unlike measures such as Pearson correlation that identify global relationships between features across the entire dataset[21,22], Hotspot finds modules of genes with significant autocorrelation within local cellular neighbourhoods of the phenotypic manifold. These modules are shared across patients and accommodate gene pleiotropy in cancer.

Hotspot identified 37 gene programs across epithelial cells from all tumours, which we manually curated, annotated and grouped (based on biological coherence) into ten modules that are shared across multiple patients (Fig. 1b, Methods, Extended Data Fig. 5a–d and Supplementary Table 4). The local modules are highly robust to varying parameter values and data downsampling, and they are context specific and would therefore be missed by standard analysis (Supplementary Fig. 2).

We identified four modules corresponding to canonical intestinal states, including ISC-like (*LGR5*, *ASCL2*), differentiated absorptive (*FABP2*, *KRT20*) and secretory (*TFF3*, *TFF1*) states. Moreover, six non-canonical modules contain tightly co-regulated genes corresponding to non-intestinal differentiated cell states (Fig. 1b). One module co-expresses *L1CAM* and *EMP1*, which have been independently shown to mark *LGR5*^low CRC metastasis-initiating cells[9,10], and additional genes associated with regeneration and therapy resistance, including *TACSTD2* (encoding TROP2)[23,24], *CD70*[25] and *OSMR*[26]. These observations support the emergence of a discrete metastasis-initiating cell population with tumour regenerative properties, distinct from homeostatic ISCs in advanced human cancer. Two modules cluster closely with this injury-repair module, one expressing canonical epithelial–mesenchymal transition (EMT) markers (*CDH2*, *VIM*) and one expressing endodermal development genes (*WNT5B*, *BMP4*). Notably, three modules express genes associated with differentiated non-intestinal cell states, representing squamous-like (*KRT5*, *ELF5*), neuroendocrine-like (*NEUROD1*, *CHGB*) and osteoblast-like (*MSX1*, *DLX5*) lineages. While the normal colon contains some enteroendocrine cells, differences in gene expression (for example, lack of intestinal transcription factors *CDX1* and *CDX2*) and prevalence in cancer cells indicate a distinct tumour neuroendocrine population.

The non-canonical differentiated squamous-like and neuroendocrine-like modules are present in many patients in our cohort, albeit to varying degrees (Fig. 1c). Most cells express multiple modules—commonly combinations of tumour ISC-like, injury-repair, EMT and endoderm modules—whereas more differentiated modules such as absorptive intestine tend to be more uniquely expressed in a cell (Extended Data Fig. 5e–j). The prominence of cells that have lost intestinal and gained differentiated non-canonical identities prompted us to visualize these spatially and to validate non-canonical marker expression at the protein level. Multiplex immunofluorescence analysis of 74 tissue sections (Extended Data Fig. 6a–f) showed that, while primary tumours display dysplastic crypt architecture, they retain expression of the intestinal lineage-defining transcription factor CDX2[27]. Moreover, differentiated intestinal marker CK20 and ISC marker OLFM4 are downregulated, while the injury-repair marker TROP2 is upregulated in metastases compared with synchronously resected primary tumours (Extended Data Fig. 6a,c–e).

## Non-canonical states associate with poor outcomes

In matched tumour pairs, we found that metastases have significantly more cells with non-canonical expression than simultaneously resected primary tumours ($P = 0.001$, rank-sum test; Fig. 1d,e). In an independent five-patient cohort of matched primary tumour and liver

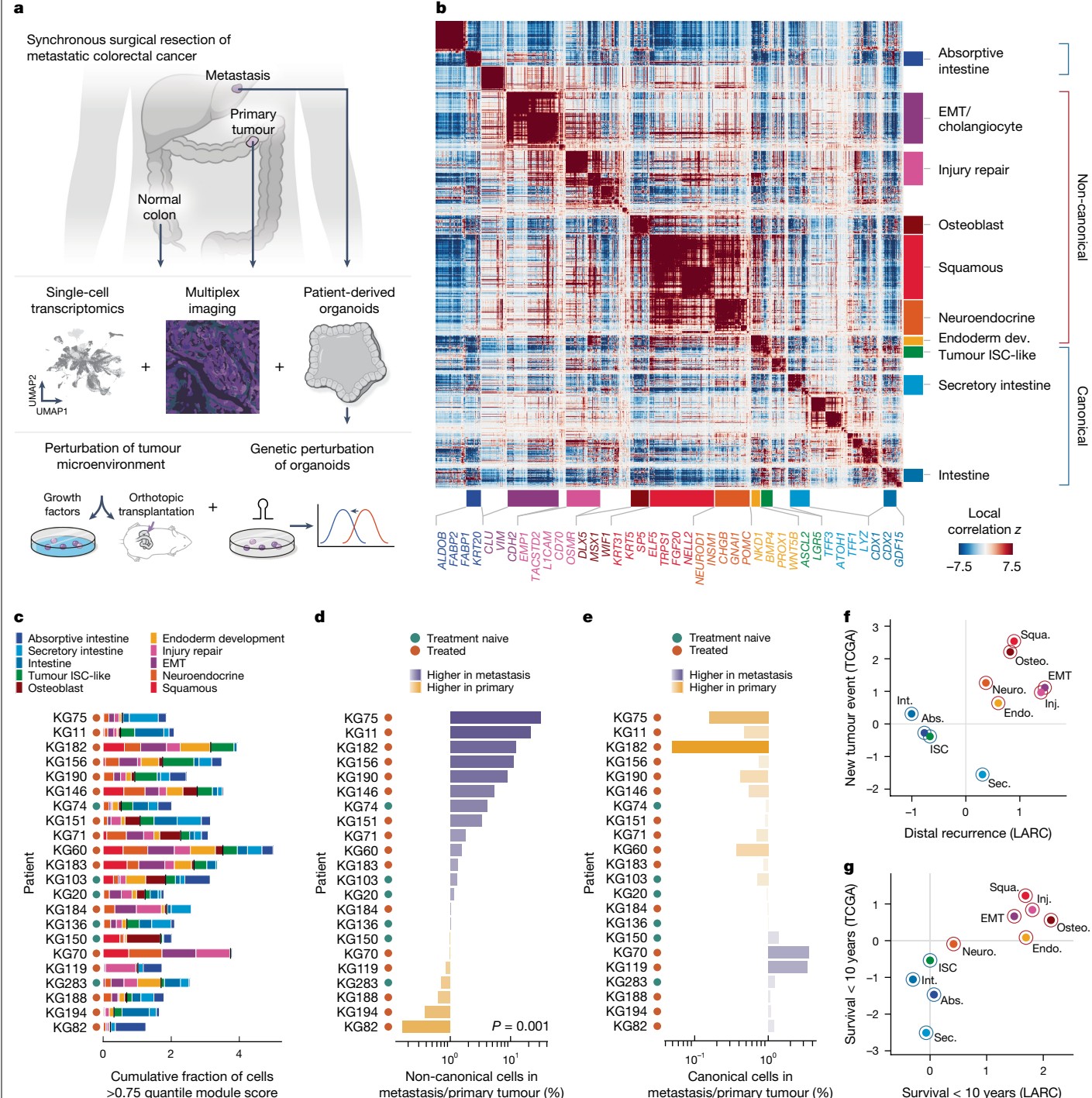

**Fig. 1 | Non-canonical transcriptional programs in CRC are associated with metastasis and poor outcomes. a**, Study design. Matched biospecimen trios of normal colon, primary CRC and metastasis were collected from 31 patients, processed fresh for single-cell transcriptomics and organoid generation, and formalin-fixed and paraffin-embedded for multiplexed immunofluorescence analysis. Organoids were used for functional studies in vitro or through orthotopic xenotransplantation into the caecum or liver. **b**, Hotspot[20] modules in all CRC tumour cells. The heat map comprises 2,003 highly variable genes with significant autocorrelation (FDR < 0.01), grouped into 4 canonical and 6 non-canonical CRC-derived gene modules (Methods and Supplementary Table 4). Dev., development. **c**, The distribution of module proportions in metastatic tumours from treated (red dots) and untreated (green dots) patients; module labels are based on the >0.75 quantile score in a given cell for the gene module (Methods). The vertical line divides canonical intestinal (right) from other (left) cell types. **d,e**, The log-ratio of metastasis-to

primary-derived tumour module proportions in each patient sample, based on the accumulation of non-canonical (**d**) or canonical (**e**) modules. Metastatic tumours are significantly enriched for cells expressing non-canonical modules, whereas primary tumours are enriched for cells expressing canonical modules (*P* = 0.001, one-sided rank-sum test; Methods). **f,g**, Associations of module enrichments with tumour recurrence (**f**) and the survival status (**g**) of patient donors in two independent clinical cohorts. Associations are shown for 108 patients with rectal adenocarcinoma (LARC[32]) and for 445 patients with colon adenocarcinoma (TCGA[31]). Enrichment scores were calculated using ssGSEA of bulk transcriptomic data from each patient (Methods). Each gene module is plotted according to the Mann–Whitney *U*-test statistic of distal recurrence (*x* axis) and new tumour event or survival under 10 years (*y* axis). Abs., absorptive intestine; endo., endoderm development; inj., injury repair; int., intestine; neuro., neuroendocrine; osteo., osteoblast; sec., secretory intestine; squa., squamous.

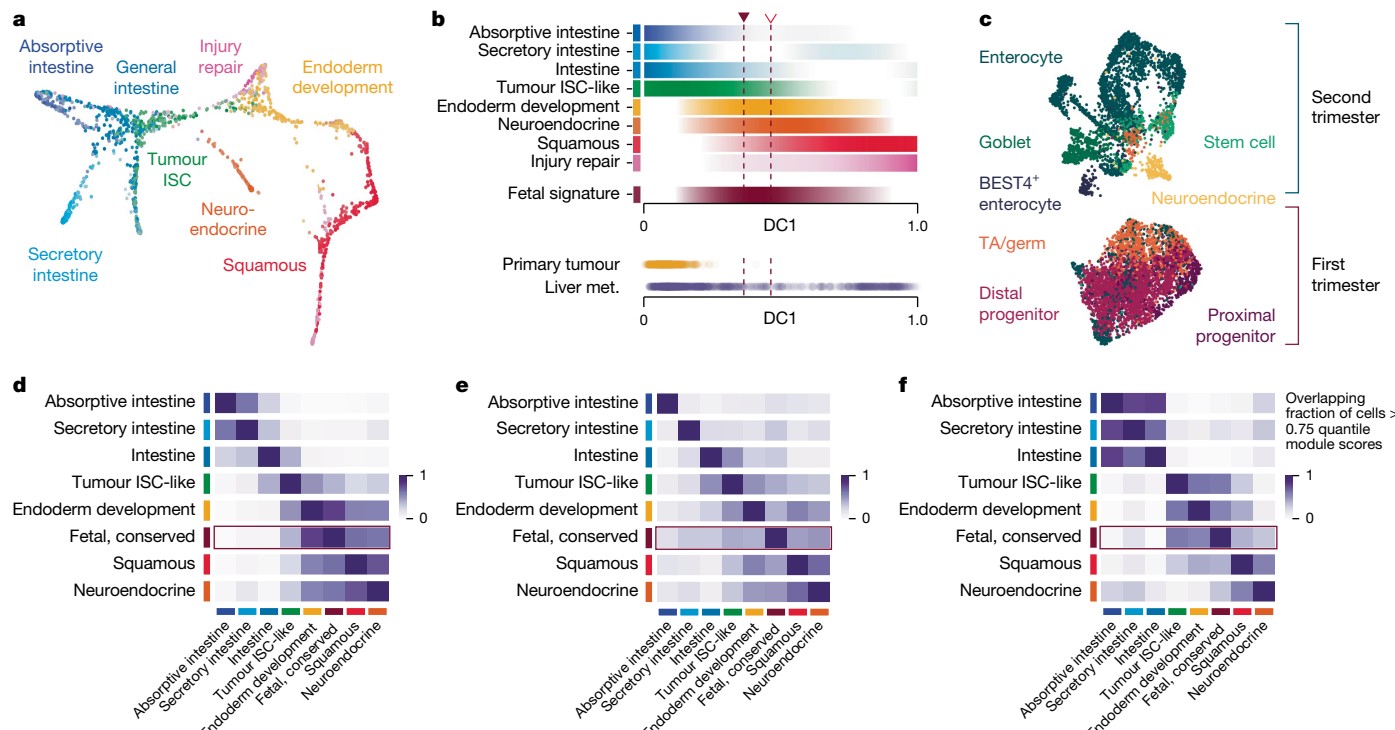

**Fig. 2 | A conserved fetal progenitor intermediate bridges canonical and non-canonical states in CRC. a**, Force-directed layout of cells from liver metastasis (met.) from patient KG146 (1,279 cells), showing a diversity of canonical and non-canonical states. Each cell is coloured according to its maximum module score. **b**, Trends in gene module scores along DC1 observed in all tumour cells from patient KG146 (top) (Methods). Each row depicts the module score along DC1 from the 20th percentile value (white) to the maximum value (highest saturation). Expression of the fetal signature peaks before the non-canonical peak. The closed and open arrowheads correspond to the 75th percentile of fetal and predominant terminal non-canonical module scores,

respectively. Bottom, positions of tumour cells along DC1. **c**, Uniform manifold approximation and projection (UMAP) embedding of human fetal colon cells[40]. First-trimester (6–11 weeks after conception) and second-trimester (12–17 weeks after conception) cells are coloured by cell type according to published annotations. TA, transit-amplifying cells. **d**–**f**, The faction of cells expressing >0.75 quantile score for a given module (or the core fetal signature, maroon rectangle) in samples from the four patients with most non-canonical cells (KG146, KG182, KG150 and KG183); **d**, all samples in our cohort (**e**) and samples from the ref. 28 cohort (**f**). Shared module expression reveals a consistent progression from canonical to non-canonical fates across patients and cohorts.

metastasis samples from a previous study[28], we also detected extensive non-canonical module expression (at least 47% of cells expressing squamous or neuroendocrine modules in each of three patients) (Extended Data Fig. 6g), and similar enrichment of non-canonical module-expressing cells in metastatic tumours ($P$ = 0.008, rank-sum test). Importantly, we observed neuroendocrine-like and squamous-like gene expression even in untreated patient tumours (Fig. 1c), suggesting that therapy is not a prerequisite for entry into non-canonical states. Consistent with our single-cell data, we observed protein expression of neuroendocrine marker CHGA and squamous marker CK5 in multiple metastatic tumour sections (Extended Data Fig. 6a,b). Non-canonical module expression is associated with metastasis and previous chemotherapy and corresponds to CRC consensus molecular subtype 4 (CMS4), which is associated with poor outcomes, whereas canonical modules correspond to CMS2 or CMS3[29] (Extended Data Figs. 5e–h and 6h,i). Together, our data demonstrate a progressive loss of canonical intestinal lineage identity and a gain of non-canonical gene expression in the transition to CRC metastasis.

To determine whether non-canonical module expression could be used as a biomarker of clinical outcome, we performed single-sample gene set enrichment analysis (ssGSEA)[30] of two independent bulk RNA-seq datasets of pretreatment primary tumours: The Cancer Genome Atlas (TCGA) cohort of 445 patients with stage I–IV colon adenocarcinoma[31] and an MSK cohort of 108 patients with locally advanced (stage II–III) rectal cancers (LARC)[32]. Expression of non-canonical modules is associated with poor outcomes in both cohorts, including tumour relapse after surgery and overall survival of less than 10 years

(Fig. 1f,g and Extended Data Fig. 6j,k). Multivariable regression analysis, correcting for clinical confounders, further validates the association of absorptive intestinal module expression with increased disease-free and overall survival in the TCGA cohort, and endoderm module expression with the opposite (Supplementary Fig. 3). Our analyses collectively suggest that subpopulations expressing non-canonical modules can exist within untreated primary tumours and undergo enrichment during metastasis, and that they are associated with negative clinical outcomes.

## Stereotyped cell-state transitions in CRC

To elucidate the canonical to non-canonical transition, we used our matched samples that span metastatic progression, beginning with patient KG146, whose tumour cells contain the broadest range of phenotypic states (Fig. 2a). We found that the diffusion component (DC) representing the strongest axis of variation in these data (DC1) corresponds to a canonical-to-non-canonical progression—from differentiated intestinal lineages to an ISC state, non-canonical endodermal development and, finally, differentiated neuroendocrine-like and squamous-like states in metastasis (Fig. 2a,b). The three additional patients with sufficient cells in non-canonical states for robust analysis display similar cell-state progression to KG146 along a top DC (Extended Data Fig. 7a).

The fact that the endoderm development module is intermediate between canonical and non-canonical states caught our attention, as transient reversion to more developmentally primitive states has been

associated with regeneration[23,33,34] as well as tumour biology[16,35–37]. Most previous research on fetal dedifferentiation is based on mouse studies[23,33,34,38,39]; therefore, to systematically characterize the human endodermal state, we used a fetal development dataset from the Human Cell Atlas[40,41]. In the first trimester, colon epithelial cells assume a fetal progenitor-like state, while, by the second trimester, ISC and differentiated intestinal states can be clearly discerned (Fig. 2c). We identified 113 genes that are differentially expressed in first-trimester progenitors relative to second-trimester mature colonocytes and together define a human intestinal fetal signature, which deviates substantially from mouse signatures (Extended Data Fig. 7b,c). Plotting our fetal progenitor signature along the tumour progression axis finds that it marks a clear intermediate state between canonical and non-canonical differentiated states (Fig. 2b).

Many fetal signature genes are highly upregulated in the tumours from patients KG146, KG182, KG150 and KG183 (Supplementary Table 5). A core set of 14 genes, including WNT genes (*TCF7*, *PTK7*), is shared by all four patients and is significantly enriched in metastases ($P = 0.0004$, rank-sum test), suggesting that these genes facilitate the transition to non-canonical fates (Extended Data Fig. 7d,e). They do not overlap substantially with existing dedifferentiation signatures, which are more intestinal and were largely derived from mice[23,33,34,38,39] (Extended Data Fig. 7b,c).

Every tumour containing enough non-canonical cells to evaluate progression exhibits a stepwise advance from differentiated intestinal states to a tumour ISC-like state, fetal progenitor and finally, differentiated non-canonical states (Fig. 2a,b and Extended Data Fig. 7a). To analyse progression in the remainder of the cohort, we reasoned that, if a large fraction of cells co-express two Hotspot modules, it suggests a pseudo-ordering of the two cell states and a transition between them. Co-occurrence analysis of all module pairs revealed the same step-wise progression across all patients in our cohort, as well as in the independent ref. 28 dataset (Fig. 2d–f). Our data therefore suggest that cancer progression involves developmental reversion, characterized by a primitive dedifferentiated state observed in first trimester colonic progenitors, with profound loss of intestinal lineage identity and upregulation of WNT-associated early developmental programs. In both the LARC and TCGA cohorts, high expression of the fetal progenitor signature in pretreatment primary tumours is associated with decreased disease-free survival (Extended Data Fig. 7g,h). Despite inter- and intratumoural heterogeneity within and across patients, the fetal progenitor state appears to function as a convergent tumour-regenerative intermediate, bridging canonical states with the aggressive non-canonical states associated with poor clinical outcomes.

Palantir trajectory inference[42] applied independently to patients KG146, KG182 and KG150 placed the fetal signature between the squamous and neuroendocrine branches (Methods, Extended Data Fig. 8a,b and Supplementary Table 5) and suggested possible driver genes. WNT-signalling genes are upregulated in both branches, whereas squamous differentiation correlates with YAP and IL-2 signalling, and neuroendocrine differentiation correlates with TGFβ signalling, including *TGFB1* and *TGFBR1* upregulation and TGFβ inhibitor *SMAD7* downregulation (Extended Data Fig. 8c).

## Determinants of non-canonical states

To address whether non-canonical differentiation is driven by cancer-cell-autonomous changes or by differences between colonic and metastatic microenvironments, we generated matched organoid models, which retain mutations from their corresponding patient tumours[43,44] (Extended Data Fig. 9a). To control for microenvironment, we propagated organoids in standard human intestinal stem cell (HISC) medium containing ISC-sustaining niche factors (Fig. 3a). We first focused on organoids derived from patient KG146 primary

tumour (OKG146P) and liver metastasis (OKG146Li), as the originating primary tumour comprises largely canonical states, while the metastasis spans the full canonical-to-non-canonical spectrum (Fig. 2b and Extended Data Fig. 9b,c).

Using a mutually nearest-neighbour approach[45] to map phenotypic states from tumours to their derived organoids, we found that OKG146P retains a largely ISC-like phenotype, whereas OKG146Li adopts ISC-like and endodermal progenitor states with little of the non-canonical gene expression observed in vivo (Fig. 3b and Extended Data Fig. 9b–d). We therefore tested whether growth factors in ISC medium might inhibit non-canonical differentiation, and found that their removal indeed causes OKG146Li cells to lose intestinal and gain non-canonical expression, while OKG146P cells retain intestinal epithelial gene expression—albeit with higher levels of differentiated intestine markers (Fig. 3b and Extended Data Fig. 9d). In liver metastases from two other patients (OKG182CW2, OKG183Li2), switching from HISC to intestinal growth-factor-free (IGFF) medium likewise decreases the expression of the ISC marker *LGR5* and increases the expression of non-canonical and canonical differentiation markers (Extended Data Fig. 9e,f). Thus, cancer cells in organoids derived from tumours with greater non-canonical gene expression in situ possess greater cell-autonomous plasticity and can upregulate genes for a variety of non-canonical lineages in response to environmental signals.

To determine whether different intrinsic abilities to adapt to colon and liver microenvironments are retained in vivo, we xenografted organoids into the caecum (intestinal microenvironment) of NOD scid gamma (NSG) mice. Both primary- and metastasis-derived organoids grew into orthotopic caecal tumours with similar kinetics (Fig. 3c), but OKG146P organoids did not grow after intrahepatic injection, whereas OKG146Li organoids readily adapted to the liver microenvironment (Fig. 3d). Intrahepatic xenografts of organoids derived from more canonical KG136 tumours exhibited similar results (Extended Data Fig. 9g,h). Multiplex immunofluorescence showed that OKG146Li-derived xenografts retain the ability to differentiate into all canonical and non-canonical states (Extended Data Fig. 9i), similar to patient liver metastases (Extended Data Fig. 6a,b).

To investigate the relationship between therapy and non-canonical gene expression, we treated primary and metastasis-derived organoids with their half-median inhibitory concentration dose of a first-line chemotherapeutic for metastatic CRC, irinotecan, for 7 days, and we found an increased expression of non-canonical modules and the fetal signature (Extended Data Fig. 10). Thus, both metastasis and chemotherapy—forms of epithelial injury that disrupt contacts between neighbouring epithelial cells—are associated with highly plastic cell states with multilineage tumour regenerative potential.

## *PROX1* represses non-canonical fates in fetal state

We hypothesized that a transcription factor acts in the fetal state to restrict plasticity in primary tumours, and identified five transcription factors that are tightly co-expressed with the fetal signature along the canonical-to-non-canonical DC (Methods and Extended Data Fig. 11a). Among these, only three are upregulated by irinotecan chemotherapy, which induces injury repair and fetal states (Extended Data Fig. 11b). The pleiotropic homeobox transcription factor *PROX1* is the most induced and is clearly correlated with non-canonical module and anti-correlated with canonical module genes in our data and the Wang et al.[28] dataset (Extended Data Fig. 11c). Multiplex immunofluorescence confirmed that PROX1 is expressed in poorly differentiated CDX2[low] cancer cells at the primary tumour invasion front, suggesting that it may act to repress non-canonical differentiation in disrupted intestinal epithelia (Extended Data Fig. 11d). Indeed, *PROX1* is expressed in injury-induced regenerative cells in normal intestinal epithelium and is required for redifferentiation into canonical intestinal states

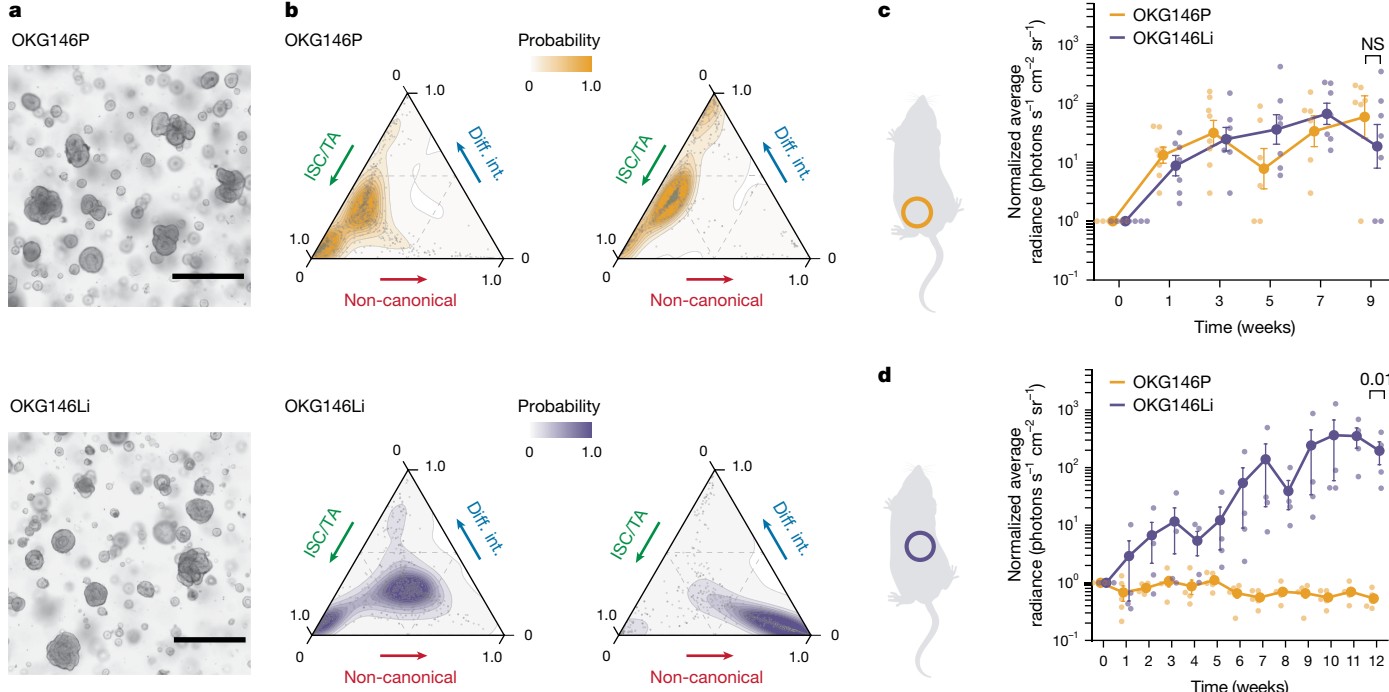

**Fig. 3 | Organoid models reveal distinct contributions of tumour and microenvironment to metastatic plasticity. a**, Bright-field microscopy showing the morphology of paired primary rectal-tumour-derived (OKG146P) and liver-metastasis-derived (OKG146Li) organoids grown in HISC medium, 7 days after seeding as single cells (2,000 cells per 40 μl Matrigel). Scale bars, 500 μm. **b**, Cell-state assignment probabilities per cell, calculated as Markov absorption probabilities (Methods) for OKG146P and OKG146Li organoids grown in HISC (left) and IGFF (right) medium. The lines indicate density contours within the 5th and 95th percentiles of each distribution, and the dots indicate individual cells. Diff. int., differentiated intestinal states. **c,d**, The normalized average radiance was measured by weekly ex vivo bioluminescence imaging after caecal injection of 200,000 cells (**c**) or intrahepatic injection of 500,000 cells (**d**) from OKG146P (primary tumour) and OKG146Li (metastasis) lines in NSG mice, normalized to the signal immediately after injection (week 0). Data are mean ± s.e.m. of *n* = 6 (OKG146P) and *n* = 7 (OKG146Li) mice (**c**) and *n* = 5 (OKG146P) and *n* = 4 (OKG146Li) mice (**d**). For **c** and **d**, statistical analysis was performed using two-sided Mann–Whitney rank-sum tests; *P* = 0.2246 (**c**) and 0.0143 (**d**) comparing between signal at end point.

during repair[46]. In mouse CRC models, increasing primary tumour aggressiveness is associated with increasing PROX1 transcriptional repressor activity[47–49].

We therefore hypothesized that cancer cells must overcome *PROX1*-dependent lineage restriction to enable non-canonical differentiation in metastasis. Accordingly, *PROX1* knockdown causes a substantial induction of non-canonical genes in OKG146P organoids, but only a small set of intestinal and non-canonical differentiation genes in OKG146Li, suggesting dampened *PROX1* repression of non-canonical genes in this context (Fig. 4a, Extended Data Fig. 12a–c,h and Supplementary Table 6). To further investigate the role of cell-state context, we engineered five additional organoid lines and assayed key genes that changed in the OKG146 scRNA-seq data. *PROX1* knockdown induces multiple non-canonical genes (such as squamous markers *KRT23* and *ELF5* and neuroendocrine markers *GNAI1* and *POMC*) in more canonical organoids (OKG136P, OKG136Li and OKG146P), but not in the most non-canonical organoid lines (OKG182CW2 and OKG183Li2), suggesting that *PROX1* repression of non-canonical gene expression depends on how canonical the underlying tumour cell state is (Fig. 4b). Only a small subset of non-canonical genes, which retain *PROX1* sensitivity in the OKG146Li *PROX1* knockdown scRNA-seq data, is upregulated by *PROX1* knockdown even in the most non-canonical organoids (Fig. 4b). Across organoid lines, *PROX1* knockdown does not alter the expression of canonical genes (*TFF3*, *FABP1*) consistently. Functionally, *PROX1* downregulation is not sufficient to induce outgrowth of liver metastasis in mice transplanted with canonical primary-tumour-derived organoids (OKG146P and OKG136P) and followed for 12 weeks, suggesting that *PROX1*-driven non-canonical differentiation cooperates with other phenotypic drivers to promote metastatic outgrowth (Extended Data Fig. 1d,e).

We hypothesized that a permissive cell state is necessary for *PROX1* to inhibit non-canonical differentiation and thereby enable re-entry into an ISC state during organoid formation. We therefore assessed the ability of short hairpin control (shControl) and shPROX1 lines spanning the canonical to non-canonical spectrum to regenerate organoids from single cells—an established assay of ISC function. Indeed, we found that *PROX1*-dependent ISC capacity depends on the cell-state context; lines with more-canonical cell states (OKG136P, OKG136Li, OKG173Li and OKG146P) form fewer organoids after *PROX1* knockdown, whereas more-non-canonical lines (OKG146Li, OKG182CW2 and OKG183Li2) are *PROX1* independent (Extended Data Fig. 12f,g). Together, our data support a model of context-specific dependency on *PROX1* lineage restriction, ISC function and tumour regeneration, with metastasis and treatment selecting for progressively less-*PROX1*-dependent, increasingly lineage unrestricted states (Fig. 4c).

## Discussion

Our unique resource of biospecimen trios from normal colon, primary and metastatic CRC enabled us to characterize the acquisition of plasticity and progression to metastasis in individual patients. Despite substantial heterogeneity among patient samples, we identified a broadly conserved succession of cell states—intestinal states first dedifferentiate into an *LGR5*[+] ISC-like state in the primary tumour, and invasive cancer cells enter an injured state marked by high L1CAM/EMP1 expression, previously associated with metastasis-initiating cells[9,10]. Cells then transition to a highly conserved fetal progenitor state with capacity for lineage reprogramming into squamous-like, neuroendocrine-like and other states, which are highly enriched in human CRC metastases. Our organoid experiments suggest a model whereby the loss of

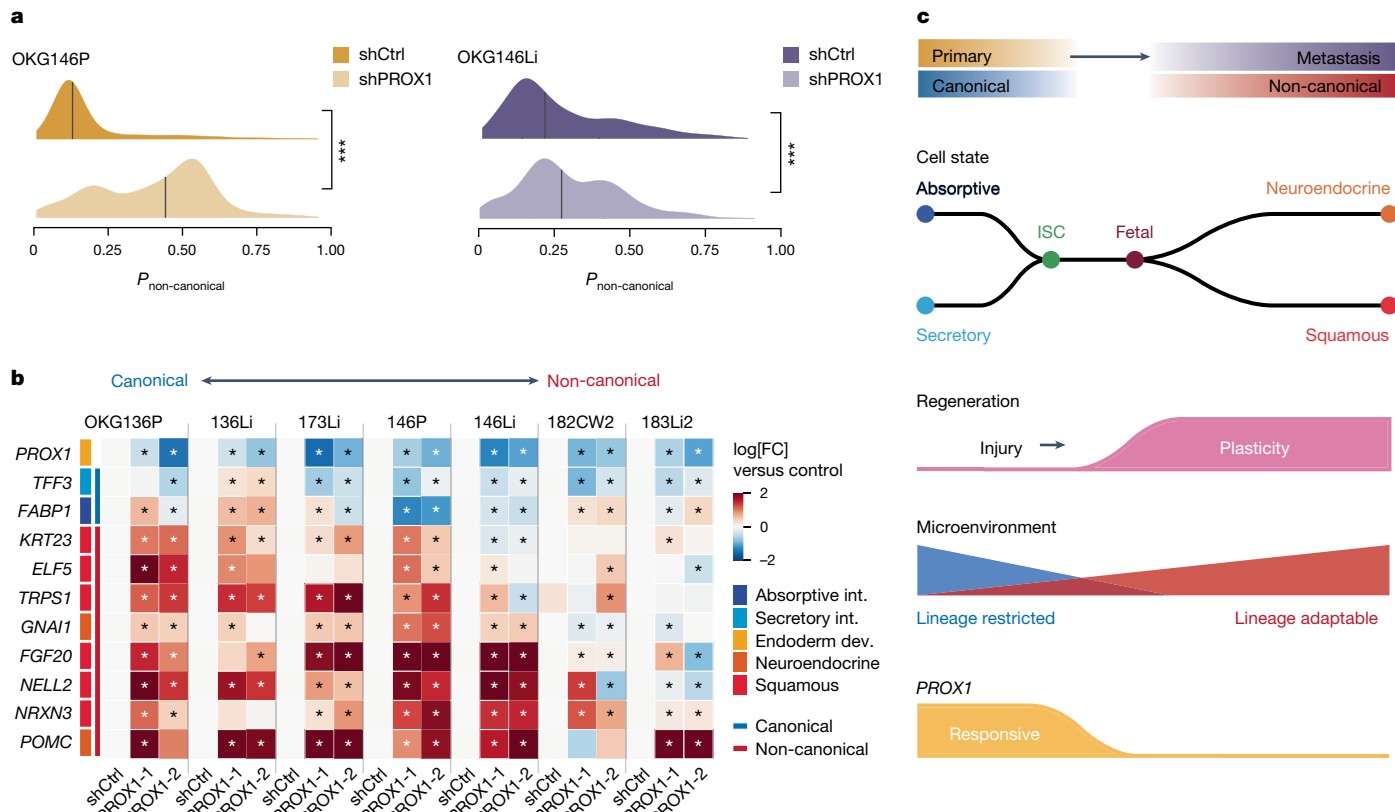

**Fig. 4 | *PROX1* encodes a fetal-state-associated transcription factor that inhibits non-canonical transdifferentiation. a**, The probability of classifying cells as non-canonical in scRNA-seq data from 146 primary (left) and liver metastasis (right) shControl and shPROX1 organoid lines, 7 days after induction with doxycycline. Cells were classified as canonical or non-canonical using a manifold-based classifier that combines methods from Harmony[51] and PhenoGraph[11] (Methods). The vertical black line indicates the median. Statistical analysis was performed using two-sided $t$-tests; ***$P < 0.001$; $P = 0.0$ (left) and $P = 2.31 \times 10^{-7}$ (right). **b**, The relative expression of canonical and non-canonical differentiation markers in organoid lines expressing shRNAs targeting *PROX1* or control shRNA (shCtrl) cultured in HISC medium containing $2 \, \mu g \, ml^{-1}$ doxycycline for 7 days. Quantitative PCR with reverse transcription (RT–qPCR) data are normalized to *GAPDH* mRNA expression. $n = 4$ replicates per group. Statistical analysis was performed using two-sided $t$-tests with Benjamini–Hochberg correction; *$P < 0.05$. **c**, Model of cell-state transitions during metastatic tumour progression in CRC. Cancer cells in the primary tumour first enter an ISC-like state, then cells at the tumour invasion front undergo developmental reversion into a fetal progenitor-like state, enabling differentiation into divergent non-canonical states, including neuroendocrine- and squamous-like, that are enriched during metastatic outgrowth. Entry into the highly plastic fetal progenitor state is triggered by epithelial injury during tumour dissemination or after therapy, allowing tumour regenerative cells to express non-canonical gene programs and adapt to diverse stresses. Induction of *PROX1* inhibits non-canonical gene expression in injured normal epithelia, enabling tissue regeneration, whereas *PROX1*-responsive intestinal lineage restriction is progressively lost during cancer progression, licensing non-canonical differentiation.

epithelial intercellular contacts first induces a highly plastic, multipotent state that is enriched in metastases, then tumour microenvironmental factors drive differentiation towards diverse intestinal and non-intestinal lineages. The greater cell-autonomous plasticity that we identify in metastatic tumours suggests a mechanistic explanation for the close link between metastasis and the ability to adapt to and evade therapy.

Non-canonical state signatures are associated with poor outcomes in two independent cohorts of pretreatment primary CRC. Thus, cells with the ability to enter non-canonical states emerge in primary tumours and may be further induced or selected for during tumour progression and by therapy. Our finding that non-canonical gene signatures can be used as prognostic biomarkers of future disease relapse and poor survival could be used to identify the patients who are most likely to benefit from neoadjuvant or adjuvant therapy targeting non-canonical states to prevent macrometastasis.

Primary tumour cells are more likely to remain in an intestinal lineage than their more plastic metastatic counterparts, in part due to the repressor activity of *PROX1*. Our data are consistent with a model of context-dependent *PROX1* function along a continuum of canonical differentiation to dedifferentiation to non-canonical differentiation during tumour progression (Fig. 4a,b). In early tumorigenesis, *PROX1* functions as a tumour promoter, with increasing levels of *PROX1* reinforcing ISC-like states by inhibiting canonical differentiation[47]. As tumour cells at the invasion front progress into injury-repair and fetal-like states, *PROX1* represses non-canonical differentiation, allowing redifferentiation to canonical intestinal fates in response to niche factors (accompanied by *PROX1* downregulation). By contrast, metastatic colonization may select for cells that have developed insensitivity to *PROX1* repression.

The non-canonical states that we identified in patient metastases are not captured in mouse models of CRC metastasis[9,17], possibly reflecting growth over longer time scales, with many more cell divisions to reach larger sizes in patients. Our data further suggest that epithelial injury can induce non-canonical differentiation (Extended Data Fig. 10). Patient metastases probably undergo rounds of proliferation and immune editing, resulting in multiple injury–repair cycles and clinical dormancy that enable or select for non-canonical differentiation before the eventual outgrowth of macrometastases[1]. Most clinical cancer genomics studies focus on primary tumours, even though new cancer

therapeutics are almost always first tested in patients with advanced metastatic disease[1,50]. The enrichment of distinct cell states in metastases that we identify underscores the limitations of extrapolating from primary tumours, and highlights the need to study metastatic tissue and patient metastasis-derived ex vivo models to delineate therapy response and plasticity mechanisms.

Phenotypic plasticity poses a major challenge to cancer therapy[50], but the identification of cell states and trajectories that are conserved across multiple patients suggest future opportunities to target plasticity, either by targeting emergent tumour regenerative injury repair or fetal-like states, or by blocking the molecular machinery that enables dynamic entry into resistant states. While identifying mechanisms that are unique to cancer cell reprogramming and dispensable for normal tissue homeostasis remains an important challenge, here we provide a roadmap for understanding and eventually targeting progressive plasticity in advanced cancer.

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

## Methods

### Patient biospecimen procurement and processing

**Tissue collection.** Patients undergoing synchronous colorectal resection and metastasectomy at MSKCC were identified by chart review, and those who had signed pre-procedure informed consent to MSK IRB protocols 06-107, 12-245, 14-244 and 22-404 for biospecimen collection were selected for this study. No statistical method was used to pre-determine sample size. Freshly resected surgical tissue in surplus of clinical diagnostic requirements was processed into single-cell suspensions for scRNA-seq analysis and, where sufficient tissue was available, processed to generate organoids. Portions were also fixed in formalin and embedded in paraffin. Tissue was generally processed within 1 h of surgical resection. Archival formalin-fixed, paraffin-embedded (FFPE) clinical tissue blocks for immunostaining were identified by database search and chart review. Tissue processing and histopathological data interpretation were overseen by an expert gastrointestinal pathologist (J.S.). Where trios of normal colon, primary CRC and metastatic CRC were successfully collected, the patient was longitudinally tracked through their clinical course at MSK using MSK Darwin[52], and tumour tissue surplus to diagnostic requirements was collected from any subsequent procedures.

**Patient metadata.** Clinical data, including baseline demographic data and previous treatments (Supplementary Table 1 and Supplementary Fig. 1), were abstracted through manual review of patient electronic medical records by board certified medical oncologists (M.L. and K.G.), collected as part of institutional review board approved protocols (MSK IRB, 14-244 and 22-404). The time to each treatment event was calculated from the date of diagnosis to allow for comparison across patients. Study data were collected and managed using REDCap electronic data capture tools hosted at MSKCC on secure central servers. 17 out of 31 patients had multiple metastatic sites at the time of surgery, and had >50% of tumour sites remaining after surgery. 17 out of 31 patients had early-onset CRC (age of diagnosis, <50 years). Clinical MSK-IMPACT targeted exon sequencing was performed on tumour/normal tissue from 27 out of 31 patients and revealed expected mutations[53] (Extended Data Fig. 1b). Consistent with the low percentage (<5%) of metastatic CRC that is mismatch repair deficient/microsatellite instability high, only one patient in our cohort had an microsatellite instability indeterminate tumour. Clinical data collection was censored on 30 September 30 2022.

**Tissue processing.** We collected 50–300 mg of freshly resected surgical tissue in 5 ml of IGFF organoid medium (Advanced DMEM/F12 (AdDF12; Thermo Fisher Scientific), GlutaMAX (2 mM, Thermo Fisher Scientific), HEPES (10 mM, Thermo Fisher Scientific), $N$-acetyl-L-cysteine (1 mM, Sigma-Aldrich), B27 supplement with vitamin A (Thermo Fisher Scientific)) supplemented with primocin (100 µg ml$^{-1}$, InvivoGen), plasmocin (50 µg ml$^{-1}$, InvivoGen), penicillin–streptomycin (100 µg ml$^{-1}$, Thermo Fisher Scientific), Amphotericin B (2.5 µg ml$^{-1}$, Cytiva), nystatin (250 U ml$^{-1}$, Millipore Sigma). For primary and metastatic tumours, specimens were placed into a 15 cm Petri dish using sterile forceps and washed three times with DPBS (Thermo Fisher Scientific) supplemented with the above-described antibiotic cocktail, and minimally chopped with sharp sterile blades to enable transfer of tumour fragments using a pre-wet 25 ml serological pipette.

Tumour fragments were transferred into a gentleMACS type C tube (Miltenyi) pre-filled with 5 ml of IGFF medium supplemented with antibiotics, DNase I (100 U ml$^{-1}$, Millipore Sigma) and a commercial cocktail of tissue digestion enzymes (Tumour Dissociation Kit, Miltenyi). Tumours were digested using the gentleMACS Octo Dissociator according to the manufacturer's 37C_h_TDK_1 protocol for a maximum of 30 min. Considering the heterogeneity of tissue specimens with respect to cell viability, immune infiltration, blood content, necrosis and calcification, the digestion state of the tumour fragments was assessed

every 10 min under an inverted microscope. Digestion was interrupted before 30 min if at least 50% of the tumour material appeared broken into clusters of 1 to 10 cells. Next, cell cluster solutions were filtered through a 100 µm cell strainer, and washed three times with DPBS supplemented with antibiotics, with each centrifugation step performed at 100$g$ for 3 min at room temperature. The final cell suspension was filtered through a 100 µm cell strainer, washed and centrifuged for 5 min at 500$g$ and 4 °C.

Non-tumour tissue was transferred into a 50 ml tube pre-filled with 25 ml dissociation/chelation buffer (8 mM EDTA, 0.5 mM DTT, DNase I (100 U ml$^{-1}$, Millipore Sigma)). Mucosal fragments were incubated with gentle rotation at 4 °C for a maximum of 30 min. The dissociation state of the tissue fragments was assessed every 10 min under an inverted microscope. Dissociation was interrupted before 30 min if at least 30% of the mucosal material appeared broken into clusters of 1 to 5 colonic crypts. Next, the crypt solution was filtered through a 1 mm cell strainer (PluriSelect) to separate individual crypts or small crypt clusters from large chunks of undissociated mucosa. Dissociation was quenched using an equal volume of DPBS supplemented with antibiotics. At this point, the 1 mm filter was flipped and inverted into a fresh 50 ml tube. Up to 25 ml of DPBS supplemented with antibiotics was flashed through the inverted filter to recover the undissociated mucosal tissue. After manually shaking the suspension of mucosal tissue fragments (approximately 5 times), the collection of clusters of colonic crypts was reattempted as described above. Based on the iteration of filtration and manual agitation steps, up to three additional fractions of crypt suspensions were collected. Crypt suspensions were washed three times with DPBS supplemented with antibiotics, with each centrifugation step carried out at 100$g$ for 3 min at room temperature. Based on visual inspection under an inverted microscope, one or more crypt suspensions were selected for subsequent processing according to the size and integrity of the crypts, and either processed separately or pooled together if individual suspensions were assessed to have low crypt content.

For both tumour and normal tissue, if blood traces were visible under an inverted microscope, the cell pellet was resuspended in 1–5 ml ACK lysis buffer (Lonza), according to the pellet size and incubated for 5 min at room temperature. Quenching was performed with three volumes of DPBS supplemented with antibiotics, followed by an additional wash to remove ACK traces. The resulting cell pellet was further processed for either scRNA-seq, organoid generation or both. Tissue processing protocols were extensively and iteratively optimized to maximize retrieval of high quality (low mitochondrial and ribosomal content) viable single-cell suspensions for downstream analyses.

**scRNA-seq.** Cell suspensions were filtered through a 40 µm cell strainer and incubated in FACS buffer (10 mM HEPES, 0.1 mM EDTA, 0.1% FBS) with DAPI (1 µg ml$^{-1}$, Thermo Fisher Scientific) and calcein AM (Invitrogen) for 5 min on ice. Viable (calcein positive) cells were sorted using a 130 µm nozzle (SH800S SONY sorter) and collected in DPBS with 0.04% bovine serum albumin (BSA). scRNA-seq was performed on the Chromium instrument (10x Genomics) according to the 3′ RNA v3.1 user manual. In brief, FACS-sorted cells were washed once with DPBS containing 0.04% BSA and resuspended to a final concentration of 700–1,300 cells per microlitre. Cell viability was above 80%, as confirmed with 0.2% (w/v) Trypan Blue staining (Countess II). Cells were captured in droplets and subjected to reverse transcription and cell barcoding; emulsions were then broken and cDNA was purified using Dynabeads MyOne SILANE followed by PCR amplification according to the manual instructions. Up to 10,000 cells were targeted for each sample. Final libraries were sequenced on the Illumina NovaSeq S4 platform (R1, 28 cycles; i7, 8 cycles; R2, 90 cycles).

**Organoid generation and culture.** Primary and metastatic CRC and normal colon organoid lines were established as previously

described[10,43,54]. Cells processed as described above were centrifuged at 600*g* for 5 min at 4 °C and resuspended at 2,000 cells per 40 µl of Matrigel. After Matrigel domes solidified at 37 °C, HISC medium supplemented with Y-27632 was added to the wells. Organoids were passaged every 7–10 days, and were considered established after three passages. For non-tumour organoid culture, HISC medium was supplemented with human R-spondin 1 (1 µg ml⁻¹; Peprotech) and NGS-WNT (0.5 M, ImmunePrecise N000). The medium was changed every 3–4 days. Organoid lines were expanded and early-passage stock vials were cryo-preserved in liquid nitrogen.

For validation, organoids underwent targeted exome sequencing by MSK-IMPACT[53] and key oncogenic genomic alterations were identified by OncoKB[55] (see below). Diagnostic tissue from originating tumours was sequenced to confirm that these alterations were conserved in each derived organoid line. Organoids were verified on the basis of short tandem repeats at the time of establishment and before every experiment, and were routinely tested for mycoplasma contamination (MycoALERT PLUS detection kit, Lonza).

**MSK-IMPACT.** Tumour and organoid targeted exon sequencing was performed using MSK-IMPACT[53]. The OncoKB precision oncology knowledgebase, an FDA-recognized human genetic variant database curated by experts at MSK[55], was used to distinguish between oncogenic alterations (presumed drivers) and variants of unknown significance (presumed passengers). Only somatic alterations labelled as oncogenic, likely oncogenic or predicted oncogenic by OncoKB were included for analysis. The MSK-IMPACT data analysis pipeline is available at GitHub (https://github.com/rhshah/IMPACT-Pipeline). Genomic alterations were annotated with information from OncoKB using the OncoKB annotator tool (https://github.com/oncokb/oncokb-annotator).

**FACETS.** Copy-number alterations in solid tumours were computed from MSK-IMPACT using the FACETS (Fraction and Allele-Specific Copy Number Estimates from Tumour Sequencing) algorithm[56], which provides allele-specific copy-number estimates at the level of both gene and chromosome arm. FACETS was also used to generate purity-corrected segmentation files, for detection of whole-genome duplication events, to infer the clonality of somatic mutations, to assess arm-level copy-number changes and to generate mutant allele copy-number estimates.

## Computational data analysis

**scRNA-seq data pre-processing. Alignment of sequencing reads.** All scRNA-seq datasets were pre-processed as follows: FASTQ files from patient samples were processed with the SEQC (v.2.7.0) pipeline[57] using the hg38 human genome reference, default parameters and platform set to 10x Genomics v3 3′ scRNA-seq kit. The SEQC (v.2.7.0) pipeline performs read demultiplexing, alignment and unique molecular identifier (UMI) and cell barcode correction, producing a preliminary count matrix of cells by unique transcripts. By default, the pipeline will remove putative empty droplets and poor-quality cells based on (1) the total number of transcripts per cell (cell library size); (2) the average number of reads per molecule (cell coverage); (3) mitochondrial RNA content; and (4) the ratio of the number of unique genes to library size (cell library complexity). However, due to the sensitivity of the colorectal epithelium to dissociation, we observed increased indicators of cell stress, apoptosis and droplet contamination in many samples, including high mitochondrial and ambient RNA expression, which can obscure statistical inference from meaningful biological gene expression. As such, typical ad hoc cell filtering based on identifying a steep dropoff in the number of transcripts per droplet (that is, a deviation leading to a 'plateau' in ambient RNA levels), could impair the extraction of meaningful biology. We therefore sought to systematically evaluate and correct for ambient RNA expression and filter for real single cells using CellBender (v.0.1.0)[58] as described below.

**CellBender to subtract ambient RNA.** CellBender (v.0.1.0)[58] is an unsupervised method for removing ambient RNA from scRNA-seq data. It first infers levels of ambient RNA and rates of barcode swapping per gene and droplet, respectively, from an unfiltered cell-by-gene count matrix. This probabilistic model is then used to generate a denoised (that is, ambient RNA-corrected) count matrix, as well as the probability that each droplet contains a cell, which can be used for calling real cells. We ran CellBender (v.0.1.0) on the unfiltered count matrix of each sample produced by SEQC (v.2.7.0) with the following parameters: (1) set the expected number of cells as the number of cells loaded into each 10x Chromium lane per sample (typically 5,000–10,000 cells); (2) set total droplets used to estimate ambient background RNA to 30,000; and (3) set training epochs to 100. We used the denoised count matrix produced by CellBender (v.0.1.0) for all subsequent analyses.

**Removal of low-quality cells.** On the basis of CellBender-corrected expression counts, we sought to identify and filter out low-quality cells from downstream analysis. As our study focuses on epithelial cells, which are known to be more sensitive to single-cell dissociation than other cell types, we paid special attention to droplet quality by performing three filtering steps:

Step 1: remove all droplets with posterior probability of containing cells ≤0.5 according to CellBender (v.0.1.0). This lenient filtering ensured that no biologically relevant cells were removed, at the cost of retaining some cells with worse technical characteristics. Step 2: remove droplets with <200 total counts, <200 total genes expressed or of which the libraries comprised >50% mitochondrial RNA. Step 3: Iterative rounds of clustering and filtering to remove low-quality or apoptotic cells that group together to create unstructured 'junk' clusters. We carried out this filtration by combining count matrices from all patients for each sample type (non-tumour, primary tumour and metastasis), clustering cells using PhenoGraph[11] ($k = 20$), and studying the covariance structure of highly expressed genes within each cluster. We reasoned that highly expressed genes are not co-regulated in cells undergoing apoptosis, nor in droplets containing ambient RNA, motivating the removal of droplets lacking meaningful covariance structure. We repeated the process of clustering and filtering until only cells residing in structured clusters remained, after which all datasets were combined.

Cells passing all of the above criteria were retained for downstream analysis.

**scRNA-seq data analysis. Data normalization and dimensionality reduction.** Raw count matrices were normalized to the median library size and log-transformed with a base of e and pseudocount of 0.1. We then selected highly variable genes (HVGs) using the highly_variable_genes function in Scanpy (v.1.9.1) and flavour = seurat_v3 (we chose bins = 40). We kept the top 50 genes within each bin, for a total of 2,000 HVGs. Moreover, for all datasets (besides the fetal colon dataset), we included genes with known relevance to normal colonic cell types (41 genes) and cell states associated with inflammatory disease, injury of the colon, and regulation of REST and EMT (56 genes) (a complete list of genes is provided in Supplementary Table 7). Going forward, we will denote the 2,097 genes including both HVGs and manual additions as HVGs. Next, we performed PCA of log-normalized matrices using only HVGs and retained the number of principal components (PCs) that explain 75% of variance (112 PCs). For all datasets as well, we chose a number of PCs explaining 75% of variance.

**Data visualization.** For all two-dimensional embeddings, we used the Scanpy (v.1.9.1) neighbours function to compute a $k$-nearest-neighbour graph on the PCs based on Euclidean distance and $k = 30$. To visualize the global CRC cell atlas (Extended Data Fig. 2b,e,h,i), non-tumour epithelium (Extended Data Fig. 2f) and human fetal gut cell atlas (Fig. 2c), we generated projections using the UMAP implementation in Scanpy (v.1.9.1), with min_dist = 0.3–0.5 and init_pos = paga. To visualize epithelial cell subsets including all untreated epithelial cells (Extended Data Fig. 3a,c), all tumour cells (Extended Data Fig. 5e–j)

and patient KG146 cells (Fig. 2a and Extended Data Fig. 9b,c), we used force-directed layouts, which provide a more intuitive representation of cell state transitions and the local relationships between subpopulations, using Scanpy (v.1.9.1) with the ForceAtlas2 layout and init_pos = paga. The Python package matplotlib (v.3.6.0) was used to produce all plots.

**Gene expression denoising and imputation.** We applied MAGIC (v.3.0.0) imputation[59] to normalized, log-transformed count matrices to denoise and recover missing transcript counts due to dropout. Imputation was performed using conservative parameters ($t = 3$, $ka = 5$, $k = 15$). Imputed values are used for visualization of gene expression or gene signature expression (described in the main text and figure legends where used), as well as for analysing mixed-lineage gene correlations in untreated patient tumours (see the 'Gene correlations in normal intestine and untreated tumour' section).

**Gene signature scores.** To generate all gene signature scores in our study, we used the Scanpy (v.1.9.1) score_genes function, which calculates the mean expression of genes of interest subtracted by the mean expression of a random, expression-matched set of reference genes. To account for expression-level differences across genes within signatures, we provided $z$-normalized expression data as the input for this function.

**Cell annotation. Partitioning cells into epithelial, stromal and immune compartments.** We clustered the dataset of all cells using PhenoGraph (v.1.5.7) with the Louvain algorithm ($k = 45$) on the PCs obtained above. To ensure robustness to the choice of $k$, we repeated PhenoGraph (v.1.5.7) clustering for all values of $k$ between 20 and 100 in increments of 5 and calculated the adjusted Rand index between each pair of clusterings. We chose the value $k = 45$, which, within small variation, generated a Rand index > 0.9, indicating that cell assignments to clusters remain mostly unchanged and are therefore robust to the choice of $k$.

We next partitioned clusters into epithelial, stromal and immune compartments based on marker gene expression (Extended Data Fig. 3a,b). Specifically, we used the score_genes function in Scanpy (v.1.9.1)[60] to score expression of compartment-specific gene signatures from ref. 61, similar to the strategy used in that study (signatures for each compartment are shown in Supplementary Table 7). Each cluster was assigned to the compartment with the maximal score.

**Analysis of the epithelial compartment.** We filtered the epithelial compartment to discard any remaining low-quality cells, by removing the lowest modes in the distributions of log-transformed library-size, log-transformed number of genes expressed and the fraction of mitochondrial RNA, resulting in 67,534 epithelial cells. We chose to assign thresholds separately for each compartment due to their different sensitivities to dissociation and sample preparation, as well as inherent biological differences between compartments, for example, tumour cells often have very large library sizes compared to immune or stromal cells.

Within the epithelial compartment, we then recomputed HVGs (2,097 HVGs), re-performed PCA (210 PCs, 75% variance explained) and clustered cells with PhenoGraph (v.1.5.7) ($k = 30$) and removed the four remaining outlier clusters containing cells belonging to patients KG103, KG105 and KG66. These clusters were characterized by a very low library size; little block structure in their gene covariance matrices, primarily containing mitochondrial and ferroptosis-associated genes; and a strong overlap between cells originating from the patients' non-tumour, primary and metastatic samples. Consistent with our scRNA-seq data, we observed very aberrant mucosa in histological images of non-tumour samples for these patients, noting an association with previous disease conditions, which may explain the poor sample quality. Together, these observations suggested that the clusters probably represent highly stressed or dying, disease-associated cells that would not be informative to our study. After removing them, 47,437 cells remained; all downstream analysis of the epithelial compartment was performed on these cells.

**Tumour cell identification using single-cell CNA calls.** We identified cancer cells in the epithelial compartment (Extended Data Fig. 3c–e) using the following criteria: (1) evidence of copy-number alterations (CNAs) compared with cells originating from non-tumour colon samples; and (2) clustering that is distinct from non-tumour epithelial cells.

We identified CNAs at the single-cell level using infercnvpy (v.0.4.0), a Python implementation of InferCNV[12], using a sliding window of 200 genes and the default parameters. The mean expression of the reference diploid was determined using all available normal tumour-adjacent samples. We performed Leiden clustering on the inferred copy-number matrix and called cancer cell clusters if they had <25% normal tumour-adjacent cells and an average CNA score ≥1 s.d. from the diploid mean. As a result, 3,102 cells derived from tumour samples with no CNA were reclassified as non-tumour epithelial cells (we performed cell type annotation on these cells in the section below), and 26,145 cells derived from tumour samples with CNA remained classified as tumour cells. We also produced an independent CNA estimate using the FACETS pipeline[56] for samples from patients with targeted DNA panel sequencing data. In many cases, patient CNA estimates by FACETS were consistent with the most abundant single-cell CNA profiles computed with InferCNV for the same patients (an example is shown in Extended Data Fig. 3c,d).

**Cell type annotation in the non-tumour epithelial compartment.** To annotate epithelial cell types, we retained the subset of normal epithelial cells (21,297 cells) collected from the adjacent normal samples and those identified as normal from the tumour samples, computed a PC representation (249 PCs) of the resulting log-normalized count matrix, and clustered cells with PhenoGraph (v.1.5.7) using the Leiden option and $k = 15$ on the obtained PCs. We ensured robustness to the choice of $k$ as described above. This process resulted in 48 clusters of cells, which were annotated into cell type based on two criteria: (1) similarity of mean $z$-normalized expression to that of canonical marker genes for major colon epithelial cell types (Extended Data Fig. 3g); and (2) GSEA of relevant cell type gene sets from the literature (Supplementary Table 2), based on differentially expressed genes (DEGs) in each cluster compared with the rest, computed using the R package MAST (v.1.16.0)[62]. GSEA was performed using the Python package gseapy (v.0.14.0)[63] with 10,000 permutations and the default parameters. On the basis of both criteria, we manually annotated the clusters as ISC (2,580 cells), absorptive precursor (6,874 cells), enterocyte (2,115 cells), BEST4+ enterocyte (1,367 cells), secretory precursor (5,573 cells), goblet (1,751 cells), tuft (848 cells) and enteroendocrine (189 cells) (Extended Data Fig. 3f–j).

**Comparison of normal ISCs and treatment-naive tumours. Creation of an ISC-specific gene signature.** To determine ISC-specific marker genes, we performed differential expression analysis of ISC cells against all other non-tumour cells using MAST (v.1.16.0) on the normalized, log-transformed count matrix for non-tumour epithelial cells, and calculated gene rankings according to the $-\log[P] \times \log[\text{fold change}]$ value for each gene (Supplementary Table 2). The final ISC gene signature consisted of the top 100 ranked DEGs ($P < 0.01$ for all genes included; Supplementary Table 2). We calculated a gene signature score on the $z$-normalized expression of these genes using the score_genes function in Scanpy (v.1.9.1) (Fig. 3c).

**PCA and annotation of PC1.** We took a subset of the single-cell dataset containing all untreated normal and tumour epithelial cells (13,935 cells) and performed PCA of the log-normalized expression matrix. We focused on the first PC (PC1), which we note was responsible for 13.5% of the variance in the dataset (compared to 8.96% of variance described by PC2). To annotate PC1, we ordered all genes according to their feature loadings on PC1, excluding genes with zero-valued loadings which cannot be ordered. Using this ordering, we performed GSEA using the prerank function of the Python package gseapy (v.0.14.0) with

relevant cell type gene sets from the literature[13,21,61,64,65] (Supplementary Table 3) and the default parameters as inputs (Fig. 3d).

**DEG and GSEA analysis between untreated tumour and ISC cells.** We performed differential expression analysis of ISC cells against all untreated tumour cells using MAST (v.1.16.0) and GSEA using relevant cell type gene sets from the literature (Supplementary Table 3) as well as all Hallmark[66] and KEGG[67] gene sets (Extended Data Fig. 3f). GSEA was performed using the prerank function of the Python package gseapy (v.0.14.0) with 10,000 permutations and the default parameters (Supplementary Table 3).

**Identification of ISC phenotypic admixture in treatment-naive tumours.** To assess whether the cell type promiscuity observed at the cluster level (Extended Data Fig. 3e) is also present in individual cells, we used MAST (v.1.16.0)[62] to compute DEGs enriched in ISCs, enterocytes and goblet cells in the non-tumour epithelial dataset; ranked each DEG by $-\log[P] \times \log[\text{fold change}]$ value; and used the top 300 genes as lineage markers (Supplementary Table 3). For enterocytes and goblet cells, DEG analysis was restricted to differentiated cell types (that is, excluding precursor cell types). This allowed genes shared between precursor and differentiated cell types of the same lineage to be recovered by DEG analysis. For example, the expression of enterocyte marker *SLC26A3* in late-absorptive precursor cells lowered its observed differential expression in enterocytes compared with all other cells when absorptive precursors were included in DEG analysis, despite abundant *SLC26A3* expression among enterocytes.

As tumour cells do not emulate the complete phenotype of normal differentiated cell types[68], we restricted the 300 cell lineage markers from above to genes that are also abundantly expressed in more than 20% of primary tumour or metastasis cells. We consider a lineage marker to be abundantly expressed in a cell if its normalized expression in that cell was greater than or equal to the bottom quartile of expression from the lineage for which it is a marker. To ensure the specificity of markers, we also removed any gene that is abundantly expressed in other lineages. Finally, we determined the fraction of the remaining lineage markers that are abundantly expressed in each normal cell and treatment-naive tumour cell. The distributions for each cell type and tumour type are visualized using the sns.kdeplot function in Python Seaborn (v.0.11.2) (Extended Data Fig. 3h–j).

**Gene correlations in normal intestine and untreated tumour.** To understand whether the co-expression of conflicting cell type markers (Extended Data Fig. 3h–j) is associated with the loss of cell-type-specific gene regulation in tumours, we first computed Pearson correlations between all pairs of top ranking DEGs for ISCs, enterocytes and goblet cells as described above (300 genes total) using the imputed expression matrix for (1) all non-tumour epithelial cells and (2) all treatment-naive tumour cells (Extended Data Fig. 3g). To account for heterogeneity in gene regulation across tumours, correlations were first computed for each patient tumour separately, resulting in 12 correlation matrices, and averaged. In this way, the observed strong correlations correspond to those gene pairs with strong positive or negative correlations across multiple patient tumours, suggesting consistent gene dysregulation compared to the non-tumour setting.

**Identification of Hotspot gene modules in CRC tumour data.** We used Hotspot (v.0.9.1)[20], an algorithm for identifying context-specific gene modules within single-cell datasets given a user-provided metric for local cell–cell similarity, to identify shared context-specific gene modules in our patient tumour dataset. Hotspot evaluates the pairwise local correlation between genes in local cell neighbourhoods within the cell–cell similarity (k-nearest neighbours (k-NN)) graph, identifying genes with high local autocorrelation. Importantly, the way significant local autocorrelation is detected is suited to scRNA-seq, and is resilient to issues such as gene drop-out in individual cells. The computed gene–gene affinity matrix is clustered to output a set of gene modules.

In contrast to global measures of correlation (such as Pearson's correlation), which presume relationships between features are consistent across a single-cell dataset, Hotspot's local correlation measures are computed on the local cell neighbourhoods in the k-NN graph.

While non-negative matrix factorization has been used to identify cancer gene programs[21,22], it is a linear method that requires a consistent and complete decomposition of the entire dataset, and it can be sensitive to batch and other variation. By contrast, Hotspot is based on gene–gene covariance, which better represents sets of genes that work together towards common functions, and is robust to batch effects[57] (and is therefore likely to be more robust to variation between tumours). Importantly, Hotspot gene modules are based on covariance that can be localized to a cell subpopulation or exhibit potentially nonlinear graded expression over regions of the manifold. In cancer in particular, heterogeneous cell states adapted to different patient tumour contexts, such as metastatic sites, are likely to correspond to differences in gene covariance. Hotspot is also well suited to handle both gene pleiotropy and rare populations, which have important roles in tumour contexts. We therefore chose Hotspot to characterize sources of inter- and intra-tumour phenotypic heterogeneity in our dataset alongside global, manifold-defining modules of genes.

To apply Hotspot, we first partitioned the data to only include tumour cells (26,145 cells), and used their top 2,097 HVGs (see the 'Data normalization and dimensionality reduction' section above). After renormalization, we performed PCA and retained enough PCs to explain 75% of the variance (233 PCs). We then identified a subset of significantly autocorrelated features with respect to the PC latent space by running Hotspot using the depth-adjusted negative binomial (danb) observation model and 30 neighbours. The danb model[69] comprises the background null distribution against which expression counts are normalized, and is used to avoid flagging genes as significant due to local autocorrelation in the library size of cells.

We retained 2,003 genes with a FDR < 0.01 for calculating local correlations and downstream clustering (see below). We used the create_modules function in Hotspot (default parameters except for minimum_gene_threshold = 20 and core only = False) to obtain a preliminary set of 37 co-varying gene modules (gene assignments to modules are shown in Supplementary Table 4).

**Hotspot module clustering.** Hotspot clusters the gene–gene local correlation matrix into modules using an agglomerative hierarchical clustering procedure that, at each step, merges two genes/modules with the highest pairwise z-scored correlation. Once a merged module contains more genes than a minimum threshold, it is labelled and cannot be combined with other labelled modules. The process ends when the highest pairwise z-score between unmerged modules falls below a minimum value; at that point, all genes that are unlabelled are not assigned to any module.

In practice, we found that choosing a large value for the minimum gene threshold left too many genes unassigned and created modules that, due to their high numbers of genes, were difficult to interpret. Choosing too small of a threshold failed to merge some intercorrelated, biologically similar modules. We opted to use a lower threshold of 20 genes minimum—to err on the side of more manageable, smaller modules—and then manually group modules with similar biological interpretations, after ensuring their genes were also correlated. This process is described in detail below.

**Hotspot module grouping and annotation.** All original 37 Hotspot modules were annotated manually based on known canonical markers for intestinal and non-intestinal cell types and from the literature (a subset of annotation genes is shown in Fig. 1b and Extended Data Fig. 5b,c, and all annotation genes are shown in Supplementary Table 4). Gene set over-representation analyses using gseapy (v.0.14.0) and the Gene Ontology Biological Process gene sets provided supporting evidence for our initial annotations, or suggested possible directions

to investigate for modules that were difficult to annotate (module annotations and over-representation analysis for final module annotations are shown in Supplementary Table 4).

We focused on 23 out of the 37 modules (1,201 genes) representing meaningful biological gene programs and did not further explore 14 modules (722 genes) annotated as cell cycle/proliferation (2 modules), cell stress (4 modules), leukocyte (3 modules) or cilia (1 module), or else could not be interpreted (4 modules). We then manually grouped 19 modules into 6 groups after (1) arriving at the same biological interpretation for all modules within a group, and (2) ensuring the local correlations between genes of grouped modules were high on average (Extended Data Fig. 5a,b). These six grouped modules and four single modules resulted in ten final gene modules (Extended Data Fig. 5a,b and Supplementary Table 4).

Once the modules were annotated and grouped using the above-described strategies, we categorized them into two distinct categories:
(1) Canonical: modules describing canonical intestinal cell types and processes such as epithelial differentiation, mucus production and small-molecule transport, which are critical for maintaining normal intestinal function.
(2) Non-canonical: modules describing processes not typically seen in healthy intestine, such as keratinization, inflammatory response and wound healing.

**Hotspot gene module scores.** Hotspot modules scores were calculated for our tumour dataset using Hotspot's calculate_module_scores function. In brief, Hotspot evaluates per-cell scores for each module using the following procedure: for all genes in a module, expression counts are first mean-centred using the danb null model and smoothed using the weighted average of their nearest neighbours. The background null distribution factors cell library size differences into account; thus, centring based on the null ensures that correlations are not impacted by library size differences. PCA is then performed on the resulting centred counts and the first PC values are used as the module scores for each cell. We used Hotspot module scores to visualize and summarize the patterns of module expression within cell groupings (for example, clusters), and to relate gene modules to gene signature scores for pre-existing gene sets as well as to sample metadata.

To plot groups of cells with top-scoring expression for a given module, we used the scanpy dotplot function (Extended Data Fig. 5b,c). To assign cells as top-scoring for a Hotspot module, we $z$-normalized all Hotspot module scores across cells, and then required a top-scoring cell to very specifically express a given Hotspot module and not express any others. To ensure this, we required top-scoring cells to have (1) a score of >1 s.d. above its mean module score, and (2) a score of <1 s.d. above the mean for all other modules (cells are assigned to 1 out of the 23 ungrouped modules on which we focus in Extended Data Fig. 5b and to 1 out of the 10 grouped modules in Extended Data Fig. 5c). We found these criteria to be more selective than, for example, the percentile scores used in other sections when identifying cells that are high for a given module.

**Robustness of Hotspot modules.** We evaluated the robustness of the Hotspot analysis to the number of HVGs used as input features, based on the consistency of gene autocorrelation, and the consistency of obtained modules. We also evaluated the robustness of the Hotspot modules to bootstrapping of cells.

Consistency of gene autocorrelation to number of HVGs: we evaluated whether the input cell–cell similarity matrix faithfully captures the structure of the data across input features (highly variable genes). Given an input gene set, Hotspot removes genes with low autocorrelation along the $k$-NN graph, ensuring that only genes that vary along the manifold in an informative manner are selected for module detection. To determine whether genes passing this criterion are robust to the number of selected HVGs, we recomputed modules with 1,000–5,000 HVGs in increments of 500 genes, keeping all other parameters constant. For each combination, we calculated the difference in Hotspot local autocorrelations for each gene against its autocorrelation score

when we input 2,000 HVGs (the value used in this study) and visualized the average as a box plot (Supplementary Fig. 2). These differences are minimal across Hotspot runs for all HVG pairs (maximum difference of 0.07), suggesting that the cell similarity graph retains its structure regardless of how many HVGs are chosen.

Consistency of modules to number of HVGs: to verify that the set of genes comprising each module is robust to the number of features, we generated Hotspot modules for 1,500–2,500 HVGs in increments of 50 genes, with minimum_gene_threshold set to 20 and core_only set to false. We then calculated Pearson correlations between the module scores of each set of Hotspot modules and the set obtained with 2,000 HVGs, as used in this study. For each original Hotspot module obtained with 2,000 HVGs, we report the correlation of the module used in this study and the best-matching module—the module that is most highly correlated with the original module in this study (Supplementary Fig. 2). In general, every set of gene features identified a subset of modules showing close correspondence to our final set of modules based on max correlation.

Robustness of Hotspot gene modules to cell downsampling: to determine whether Hotspot modules depend on the exact cells used as input, we used a bootstrapping-like approach. We randomly resampled our tumour dataset ten times, with 1–10% of cells removed (in increments of 1%) and with 10%, 15% or 20% of cells removed. For each resampling, we calculated new PCs and a new $k$-NN graph, and reran Hotspot with the same genes and identical parameters as in our main analysis. We then evaluated the similarity of each Hotspot analysis with our original analysis as the maximal difference in local correlations (that is, the difference between the highest and lowest local correlation values across the matrices being compared) (Supplementary Fig. 2). We used this approach because cells cannot be resampled with replacement from a $k$-NN graph.

Relationship of Hotspot results to global correlation: while Hotspot gene–gene correlation is only evaluated on cell subsets, the subsets are highly constrained to the neighbourhood structure of the $k$-NN graph, which is a strong structural feature of the data, and makes it unlikely that Hotspot will find spurious modules. Nevertheless, we expect that these modules will also result in detectable levels of global Pearson correlation. For visual comparison, we plotted pairwise Pearson gene correlation against $z$-scored pairwise Hotspot local correlation on the log-normalized expression matrix (Supplementary Fig. 2), revealing that the ranking of weak and noisy global signals detected by global correlation largely align with the robust signals sensitively detected by Hotspot. Averaging all pairwise correlations within grouped modules likewise reveals that correlation and global correlation are qualitatively similar (Supplementary Fig. 2).

**Distribution of module expression among samples.** To determine the distribution of cells with high expression of the ten Hotspot modules on a per-patient basis, we first labelled a given cell with a gene module if it had >0.75 quantile expression score for that module, then plotted the cumulative fraction of labelled cells for all modules in each tumour sample (Fig. 1c), or in pooled primary and metastatic samples per patient (Extended Data Fig. 6g). Cumulative fractions can exceed 1 as a cell can exhibit high expression of more than a single module.

We also visualized non-canonical or canonical module prevalence (Fig. 1d,e) as the log-ratio of labelled cell fractions in metastatic to patient-matched primary tumours. Canonical and non-canonical module classifications are described in the 'Hotspot module grouping and annotation' section. Although cells can exhibit high expression of multiple canonical or multiple non-canonical modules, we found that they do not often express both canonical and non-canonical modules highly (Supplementary Fig. 2); thus, we labelled cells as non-canonical or canonical according to which classification they score highest (for example, a cell of which the maximum module score is squamous is labelled non-canonical).

We calculated significance values for these plots using the following strategy: For each binary labelling of cells (for example, non-canonical or not non-canonical), we randomly permuted the labels 1,000 times within each patient separately. For each random permutation, we calculated the log-ratios of positive cells (for example, non-canonical) in metastases to positive cells in primary tumours, as above. We then performed a rank-sum test comparing our original log-ratios to the combined log-ratios from the random permutations. The alternative hypothesis is that our sample distribution is greater than the null (for example, metastasis is higher).

**Interpatient entropy of gene modules.** We used entropy to evaluate the patient-specificity of each Hotspot module using the following procedure: (1) sample 357 cells per patient with replacement in our tumour dataset to ensure even cell distribution across patients. (2) For each module, determine the subset of high-scoring cells with module scores greater than 1 s.d. above the mean. (3) Compute the Shannon entropy of patient labels in the high-scoring subset of cells for each module using the SciPy (v.1.9.1) function scipy.stats.entropy. To calculate the Shannon entropy, we first built a $k$-NN graph with $k = 60$ on the multiscale space embedding of all epithelial cells; multiscale space was computed on the top 19 DCs (chosen by the knee-point of DC eigenvalues) using Palantir (v.1.2). Steps 1–3 were repeated 100 times before visualizing the entropy distribution with kernel density plots using the kdeplot function in Python Seaborn (v.0.11.2) (Extended Data Fig. 5d).

**Association of gene modules with clinical covariates in bulk RNA-seq data.** To test the association of each Hotspot modules with clinical covariates in bulk cohorts, we (1) ran single-sample GSEA on two bulk RNA-seq datasets, LARC and TCGA-COAD, collected from CRC patient primary tumours using the genes in our Hotspot modules as the input gene sets; and (2) for each dataset, tested the association of patients' clinical features with the enrichment scores of their tumour samples. **ssGSEA analysis.** For LARC, we analysed 108 LARC tumour samples with available RNA-seq data[32]. Genes were retained if they had >1 count per million across more than 50% of the samples. The edgeR v.3.40.2 package[70] was used for trimmed mean of $M$-values normalization and FPKM transformation and the org.Hs.eg.db v.3.16.0 package was used for gene annotation. Genes that mapped to multiple Ensembl IDs were removed. We then performed ssGSEA analysis using the hacksig v.0.1.2 R package[30] and all Hotspot modules. In this cohort, 0.4% of patients have a survival status of alive and an overall survival (OS) follow-up time of <12 months (0% with OS follow-up of <6 months); 1.7% of patients have no distal recurrence and a disease-free survival (DFS) follow-up time <12 months (0% with DFS follow-up of <6 months).

For TCGA, we downloaded and analysed RNA-seq data for 445 tumour samples from the TCGA-COAD study[31]. RNA raw counts were retrieved using TCGAbiolinks (v.2.26.0)[71] and genes with a count of 0 across all the samples were removed, as well as genes that had multiple associated gene symbols or no gene symbol. The VST transformation was performed using the DESeq2 (v.1.38.3) package[72]. Subsequently, ssGSEA analysis was conducted utilizing the R package GSVA (v.1.46.0)[73]. In this cohort, 13% of patients have a survival status of alive and an OS follow-up time of <12 months (7% with OS follow-up of <6 months); 0.9% of patients have no new tumour event and a DFS follow-up time of <12 months (0.7% with a DFS follow-up of <6 months). **Associations with clinical covariates.** For binary clinical features, we separated the enrichment scores into two groups based on the status of the patient from which the bulk RNA-seq data were collected and compared the enrichment scores for each Hotspot module between the two groups using the Mann–Whitney $U$-test (Fig. 1f,g and Extended Data Fig. 6h–j). **Survival analyses in TCGA-COAD cohort.** For each Hotspot module, we collected two groups of samples from the TCGA-COAD cohort: (1) highly enriched samples with ssGSEA enrichment scores greater than

1 s.d. above the mean enrichment score among all samples, and (2) low-enriched samples with enrichment scores less than 1 s.d. below the mean. We then performed a log-rank test on DFS between these groups using the lifelines (v.0.27.4) package in Python[74]. We generated survival curves for all modules with significant results ($P < 0.05$) (Extended Data Fig. 6k). Multivariate logistic regression models were used to evaluate associations between DFS, OS and module expression. Each sample was annotated as high or low for each signature based on an ssGSEA score >0.75 s.d. above or below the mean, respectively. Samples without the signature annotation were excluded from the analysis. Cox proportional hazards tests were used for multivariate analysis of DFS. We generated forest plots for all modules with significant results ($P < 0.05$) (Supplementary Fig. 3). R packages survival (v.3.6-4) and survminer (v.0.4.9) were used for the survival analysis.

**Delineation of canonical to non-canonical tumour axes across patients.** To characterize trends in cancer progression, we analysed the four patients with a sufficient number of cells in non-canonical states for robust characterization, namely KG146 (3,351 cells), KG182 (935 cells), KG183 (1,203 cells) and KG150 (2,574 cells). We reprocessed each patient individually to most faithfully capture trends within each individual patient; data from primary tumour, synchronous metastatic tumour and metachronous metastatic tumour samples were pooled for each patient and processed as described in the 'Data normalization and dimensionality reduction' section. We used DC analysis, which identifies the largest axes of nonlinear variation in the data and has been shown to effectively capture cell-state transitions in scRNA-seq data[75]. DCs were independently computed for each patient to separately compute potential per-patient routes of tumour progression and to avoid artificially imposing the trends from patients with large samples on patients with smaller samples.

We computed diffusion maps ($k = 30$ nearest neighbours) for each patient and retained a subset of DCs based on the eigengap of the ranked components' eigenvalues (KG146, 4; KG182, 6; KG183, 8; KG150, 4 DCs). The strongest DCs appear to define a continuum from canonical to non-canonical fates. Thus, we ranked the DCs of each patient by the difference between the average Spearman correlation of a given DC with (1) all non-canonical modules and (2) all canonical modules; the greatest difference between the two averages defines an axis for canonical to non-canonical transformation (Fig. 2b and Extended Data Fig. 7a). For KG146, KG182 and KG150, the first DC was selected, while the fourth DC was selected for patient KG183. The consistent and independent selection of the first DC as the canonical to non-canonical transition in 3 of 4 patients supports the importance of this axis as one of the strongest signals in the data. We note that KG183 had fewer non-canonical cells than the other patients, probably explaining why this transition was not the top DC for this patient. We used the progressive nature of our samples (normal to primary to metastasis) to reason that—as normal included only canonical, and metastasis contained the largest fraction of non-canonical cells—this axis indeed represents a cell-state progression. **Visualization of module trends.** We used generalized additive models (GAMs) with cubic splines as smoothing functions as in Palantir (v.1.2)[42] to analyse module score trends along DC axes (Fig. 2b and Extended Data Fig. 7a). GAMs increase robustness and reduce sensitivity to density differences, and cubic splines are effective in capturing non-linear relationships. We fitted trends for a module score using a regression model on the DC values ($x$ axis) and module score values ($y$ axis or colour intensity). The resulting smoothed trend was derived by dividing the data into 500 equally sized bins along the DCs and predicting the module score at each bin using the regression fit. We visualized module score trends from the 20th percentile value (white) to the maximum value (highest saturation) (Fig. 2b and Extended Data Fig. 7a).

**Derivation of a human fetal colon progenitor gene signature.** We used a fetal gut cell atlas containing scRNA-seq data from dissected human

embryos aged 6.1 to 17 weeks, which captures the development of human intestinal cells from a fetal progenitor state to differentiated crypts[40]. We downloaded a raw H5AD file from the authors containing all epithelial cells from fetal donors and limited our analysis to 8,408 cells originating from the large intestine of first and second trimester samples.

**Data reprocessing and DEG analysis.** As large intestine cells represent a minority of the fetal gut cell atlas (8,408 of 52,184 total cells), we partitioned and reprocessed the dataset to focus our analysis on these cells. We ran HVG selection (2,000 HVGs), PCA (167 PCs explaining 75% variance) and UMAP projection (min_dist = 0.5) using scanpy as described in the 'Data normalization and dimensionality reduction' and 'Data visualization' sections. UMAP projection revealed that the week 11.1 sample separated from all others. Cells from this sample were characterized by heat-shock genes *DNAJB1*, *HSP90AA1*, *HSPE1*, *HSPA8* and *HSPA1A* among the top 10 DEGs (compared to all other samples, by MAST analysis) indicating cell stress, so it was removed, resulting in 7,984 cells. In the remaining dataset, we retained the authors' original cell type annotations, but merged the enteroendocrine subtypes (M/X, D, β, L, N, K, I and enterochromaffin cells) into one group.

We found that the first trimester samples consist predominantly of progenitor cells; proximal progenitor, distal progenitor and stem cells comprise 88% of all cells. By contrast, second-trimester samples consist of mature colon mucosal cell types exclusively, and exhibit strong expression of *LGR5*, *TFF3*, *SLC26A3*, *NEUROD1* and *POU2F3* (corresponding to ISCs, goblet cells, mature enterocytes, enteroendocrine and tuft cells, respectively). We therefore concluded that the separation between the first- and second-trimester samples captures the distinction between progenitor-like cell types and colonic crypts.

To determine marker genes specific to the first trimester cell population, we performed a differential expression analysis of first versus second trimester cells using MAST (v.1.16.0) on the normalized, log-transformed count matrix and identified 173 DEGs with log[FC] > 2 and adjusted $P < 1 \times 10^{-5}$. Earlier cells are more proliferative, so we removed genes related to cell cycle or proliferation from our first-trimester gene list. Specifically, we calculated the Pearson correlation of all first trimester DEGs with 445 genes belonging to the Reactome 'Cell Cycle, Mitotic' and 'Cell Cycle, G1-G1/S Phase' gene sets and the Hallmark 'Cell Cycle, G2M Checkpoint' gene set[76]. We removed 60 genes with a correlation greater than 0.25 for at least one gene belonging to the gene sets. Our fetal gene signature comprises the remaining 113 genes (Supplementary Table 5).

**Comparison with existing dedifferentiation signatures.** We compared our 113-gene fetal signature to previously published dedifferentiation signatures[23,34,38,39]. For each pair of signatures, we calculated the Jaccard index (number of genes shared between signatures divided by total number of genes in both signatures), demonstrating that existing signatures are clearly distinct from our fetal signature and lack consensus (Extended Data Fig. 7b). We also determined how many of the 14 core fetal signature genes (see the next section) are present in each dedifferentiation signature, normalized to the total number of genes in that signature.

For each of the three major progenitor cell populations in first trimester dataset, and the five major populations in the second trimester dataset described above, we calculated the average score of the various signatures (see the 'Gene signature scores' section), finding that our fetal signature is clearly enriched in first trimester and depleted in second trimester populations, whereas other signatures lack coherent enrichment trends (Extended Data Fig. 7c).

**Mapping fetal signature along tumour progression axis.** For each patient, KG146, KG182, KG183 and KG150, we calculated gene set scores using the scanpy function score_genes and the list of 113 genes in our fetal signature. We determined which genes in our fetal signature are correlated (Pearson $r > 0.5$) with the fetal signature score trend along the major diffusion component, as in the 'Identification of fetal-state-associated transcription factors' section below (Extended

Data Fig. 7d). Among the 113 genes of the fetal signature, 88 genes are strongly correlated in at least one patient; 59 in at least two patients; and 37 in at least three patients, with a large number of genes correlated for each patient (56 in KG146, 51 in KG182, 29 in KG150 and 62 in KG183). Moreover, 14 are strongly correlated with the signature score in every one of the four patients, forming a 'core signature' of candidates for driving fetal state reversion in patient tumours. We calculated a gene set score using these 14 core genes, as well and a significance value was calculated for its distribution among samples as in 'Distribution of module expression among samples' above (Extended Data Fig. 7e).

**Kaplan–Meier analyses of fetal signature in bulk data.** For all of the samples in the LARC and TCGA-COAD cohorts, we calculated ssGSEA enrichment scores as described in the 'ssGSEA analysis' section using our fetal gene signature as input. For each cohort, we then collected (1) highly enriched samples with ssGSEA enrichment scores >1 s.d. above the mean enrichment score among all samples; and (2) lowly enriched samples with enrichment scores <1 s.d. below the mean. We performed a log-rank test on DFS between these groups using the Python lifelines (v.0.27.4) package (Extended Data Fig. 7g,h).

**Pseudo-ordering of cell states by module overlap.** Although diffusion component analysis orders patient cells along transitions from canonical to non-canonical fates in a reproducible and unbiased way, it is less effective in patients whose tumours have few cells in non-canonical states. As an alternative, we consider the observation that in most cell-state trajectories, gene modules tend to be co-expressed in the same cells if they define pairs of sequential cell states. Using this logic, the existence of a substantial fraction of cells co-expressing two distinct gene modules can be used to suggest a pseudo-ordering of these states, and a transition between them. A key feature of Hotspot is that a cell can co-express multiple modules (Fig. 1b and Supplementary Fig. 2), making it possible to examine cells occupying mixed states. To ensure that we only consider robust module expression, we assigned a cell to a module if it expresses module genes above the 75th percentile. For all pairs of Hotspot modules, we computed the fraction of modules that are co-expressed in the same cell, and we aggregated across patients due to the sparsity of some cell states in any given patient. This analysis reveals a progression (block diagonal) that agrees with DC analysis of the four patients with non-canonical states (KG146, KG182, KG183 and KG150) and is consistent across these four patients (Fig. 2d), the full cohort with these four patients removed (Fig. 2e), and five patients from an independent dataset (Fig. 2f).

**Replication in an independent CRC scRNA-seq dataset.** The scRNA-seq dataset from ref. 28 consists of matched primary tumour and liver metastasis samples for each of five patients who received multiple cycles of chemotherapy. We downloaded GEO accession GSM7058755 (colorectal cancer, nonimmune cells) comprising all tumour and stromal cells from these patients. We first filtered all cells with >40% mitochondrial UMIs or <1,000 UMIs, leaving 23,341 cells. We normalized data and classified epithelial cells using the same methods described for our data in the 'scRNA-seq data analysis' and 'Cell annotation' sections; in brief, each PhenoGraph (v.1.5.7) ($k$ = 30) cell cluster was classified according to the highest average score across all cells for a broad panel of stromal and epithelial genes. Only these filtered epithelial cells were used in downstream analyses.

Hotspot module analyses differ between our dataset and the ref. 28 dataset in two ways. First, gene module scores in our dataset are calculated using the $k$-NN graph that was generated to run Hotspot (see the 'Hotspot gene module scores' section), whereas the ref. 28 dataset requires its own $k$-NN graph, rendering the results impossible to compare. Instead, we calculated gene set scores using the scanpy function score_genes and the list of genes for each Hotspot module. Second, in our dataset, we used 0.75 quantile score thresholds to quantify

per-patient cell abundances of Hotspot modules (Fig. 1b). As these are relative thresholds, repeating the analysis in the ref. 28 dataset would not account for the possibility that module genes are expressed at lower levels in ref. 28 compared with our tumour dataset. For this reason, we chose to construct thresholds which specifically reflect the level of expression in our dataset where we know these modules are expressed. To do so, we combined the two datasets and renormalized (median library size normalization and log-scale expression) to the same level of expression. Gene set scores were calculated on the joined, renormalized datasets using score_genes and the 0.75 quantile cut-offs were based only on the cells from our dataset, so that thresholds would reflect the level of expression in our dataset and be applicable to the data from ref. 28. These thresholds were used in Extended Data Fig. 6g.

**Palantir pseudotime and branch calculations.** The four patients whose tumours include the most cells with non-canonical fates (KG146, KG182, KG150 and KG183) span the spectrum of progression, and all contain both squamous and neuroendocrine states. We used Palantir (v.1.2)[42] to investigate these two fates and the genes underlying their respective fate transitions. As an input, Palantir requires an initial state, and as the output, it computes terminal fates and provides a cell-fate map that assigns a probability for each cell to differentiate into each terminal fate. Palantir also outputs a pseudotime alignment of cells from the initial to each of the terminal states and, therefore, by combining pseudotime and fate probability for each cell, it can provide branching gene trends leading to each terminal fate (by weighing the contribution of each cell to the gene trend based on the fate probability). Palantir was run separately on the tumour datasets of patients KG146, KG182 and KG150, excluding KG183 because of an insufficient number of non-canonical cells in this patient. We selected cells with the highest imputed expression of *LGR5* as the initial state, motivated by their identification as cells of origin in CRC studies. Notably, Palantir has been shown to be robust to the exact choice of starting cell. Running Palantir with 500 waypoints and an eigengap-based number of DCs (6 for KG146, 4 for KG182 and 8 for KG150) yielded distinct branching trajectories from the *LGR5*[+] state to two *CDX2*[−] (non-intestinal) terminal cells in all three patients. We disregarded three additional branches in KG146 and KG150 that probably represent canonical-state trajectories, as the terminal cells express *CDX2* and differentiated intestine markers including *FABP1* and *TFF3*.

To annotate the two non-canonical terminal states, we compiled known squamous and neuroendocrine cell markers observed in healthy cells or non-CRC cancers, and calculated the Pearson correlations between their imputed expression and non-intestinal branch probabilities (Extended Data Fig. 8a,b). We excluded all cells with a probability of <0.5 for a given branch when calculating correlations, to avoid interference by cell states outside that branch. This analysis identified branches to neuroendocrine-like and squamous-like states in each patient (a representative example for KG146 is shown in Extended Data Fig. 8a). To further support our annotations, we calculated Pearson correlations between the non-intestinal branch probabilities in each patient and imputed expression for all genes observed in all three patients. We found that the top 5 genes ordered by average correlation among the three squamous branches in KG146, KG182 and KG150 are associated with squamous epithelium and keratinization (*DMRTA1*, *NECTIN4*, *DLX3*, *CXCL14*, *LYPD3*), and 4 out of the top 5 genes among the three neuroendocrine branches are associated with glial and neural cells (*TRPM3*, *ITPR2*, *PLPPR1*, *PPFIA2*) (Supplementary Table 5).

Palantir gene trends were visualized as described in the 'Visualization of module trends' section using generalized additive models to fit gene expression along Palantir-computed pseudotime (Extended Data Fig. 8c). All expression trends for individual genes were calculated on MAGIC-imputed data (see the 'Gene denoising and imputation'

section), and the s.d. of each expression bin was represented by the s.d. of the residuals of the fit.

**Cell state classifications in KG146 patient tumours.** To annotate KG146 tumour cells, we first analysed primary tumour data (880 cells) by determining PCs that explain 75% of the variance (119 PCs) and using PhenoGraph (v.1.5.7) ($k = 30$) to identify six clusters. Repeating this process for the liver metastasis (1,279 cells, 200 PCs) yielded nine clusters. We directly transferred all Hotspot gene module scores from our complete tumour dataset to the KG146 tumour cells, calculated the average Hotspot gene module scores for each cluster and annotated the clusters as ISC-like, absorptive-like, secretory-like, fetal, injury repair, neuroendocrine-like and squamous-like based on high scores for specific gene modules associated with respective labels (Extended Data Fig. 9b,c). This resulted in four ISC-like clusters, one fetal/injury repair cluster and one secretory-like cluster in the primary tumour data, and two ISC-like clusters, one absorptive-like cluster, one secretory-like cluster, two fetal/injury repair clusters, one neuroendocrine cluster and two squamous clusters in the liver metastasis data. In both datasets, we further reclassified one cluster of ISC-like cells as TA/Proliferative-like based on their unique expression of the proliferation markers such as *MKI67* and *PCNA*.

We plotted groups of cells by PhenoGraph (v.1.5.7) cluster (column) and Hotspot module scores (row) using the scanpy dotplot function, with cluster assignments generated within each sample using $k = 30$ (Extended Data Fig. 9d).

**Normalizing and scoring gene sets in organoid data.** To ensure module and fetal signature scores (Extended Data Figs. 9d and 10a,c,d) were comparable across datasets, we joined our original HISC, IGFF and irinotecan-treated organoid and KG146 patient tumour datasets and normalized the combined data as described in the 'Data normalization and dimensionality reduction' section. We then *z*-scored the log-normalized gene expression matrix across cells and calculated gene set scores on this matrix using the score_genes function in scanpy using lists of genes for each module and the list of fetal signature genes. The shPROX1 organoid datasets were normalized as described in the 'Data normalization and dimensionality reduction' section independently of this dataset.

**Mapping organoid data to patient tumour.** To map cells of each organoid sample to the phenotypically closest tumour cell state from the full KG146 patient dataset, we developed a manifold-based classifier that combines Harmony[77], a framework for connecting scRNA-seq data using an affinity matrix augmented by mutual nearest neighbours between datasets, and PhenoGraph[11] to transfer labels between datasets. We performed the same analysis on each organoid sample individually along with the relevant matched primary or metastatic patient sample, including original (Fig. 3b), irinotecan-treated (Extended Data Fig. 5b) and shPROX1 (Fig. 4a) samples. The process involves three distinct steps—feature selection, co-embedding and classification. The resulting numpy matrices were concatenated to obtain an 'augmented' cell–cell affinity matrix that consists of three main components: (1) similarity between in vivo cells; (2) similarity between in vitro cells; (3) similarity between in vitro and in vivo cells.

**Feature selection.** We identified the top 100 DEGs for each annotated cell state in the KG146 patient tumour dataset, resulting in a total of 800 genes. We then partitioned the resulting lists to retain the subset of highly significant genes with log[FC] > 3 and Benjamini–Hochberg-adjusted $P < 0.001$, resulting in 753 genes total. These genes were used for PCA and to produce neighbour graphs in the following steps.

**Co-embedding.** As expected given the differences between in vivo and in vitro data, using a standard co-embedding approach consisting of a joint PCA and UMAP, we observed extreme batch effects between the

two datasets, making label transfer between similar cells ineffective. We therefore followed the approach outlined previously[45] to bridge between datasets. We first computed the nearest neighbour graph (Scanpy neighbours function with $k = 30$) in each dataset separately, then computed mutual nearest neighbours (MNNs) between the samples using Harmony[77]. Importantly, we used the cosine metric to quantify the distance between cells across samples, as this metric is less sensitive to technical artifacts and better reflects conserved biological states in both in vivo and in vitro samples[78]. We chose a higher number of mutual neighbours ($k = 60$) because it is more robust to sparsity in the MNN graph.

Next, the within-sample nearest neighbour and between-sample MNN graphs were converted to within-sample and between-sample affinity matrices, respectively, using the adaptive Gaussian kernel (default parameters) as implemented in Harmony[77]. The resulting matrices were concatenated to obtain an augmented cell–cell affinity matrix that consists of three main components: (1) similarity between in vivo cells; (2) similarity between in vitro cells; and (3) similarity between in vitro and in vivo cells. This matrix was input to PhenoGraph (v.1.5.7) classification (see below) to propagate labels from the reference (KG146) dataset to the unlabelled dataset (organoid) and generate UMAP co-embeddings of patient and organoid datasets.

**Classification.** Finally, we supplied the augmented affinity matrix from step 2 to the PhenoGraph (v.1.5.7) classify function[11] with the default parameters. This function converts the affinity matrix into a row-normalized Markov matrix and computes the probability of random walks starting from unlabelled cells from the in vitro samples, and reaching a class of labelled cells in the in vivo sample. Finally, each unlabelled cell is assigned the cell-state label with the maximum probability.

Given the large differences between the in vitro and in vivo samples, we wanted to summarize our classification using coarser cell typing. Thus, to summarize our organoid sample classifications, we aggregated the probabilities of different cell states into three generalized categories; ISC-like, combining TA/Proliferative for ISC/TA, absorptive-like and secretory-like for differentiated intestine, and combining fetal/injury repair, neuroendocrine and squamous for non-canonical. These groupings can be interpreted as the likelihood of a cell belonging to any of the several cell states which were combined. The probabilities for each resulting categories are plotted using the python-ternary (v.1.0.8)[79] package (Fig. 3b and Extended Data Fig. 10b).

**Identification of fetal-state-associated transcription factors.** We aimed to generate a ranked list of human transcription factors that are associated with the fetal progenitor state in a conserved manner across non-canonical patient tumours. Starting with a list of 1,665 human transcription factors[80], we restricted potential targets to transcription factors for which we observe (1) greater than 10 UMIs total in all four patient datasets (leaving 1,099 transcription factors); (2) at least 5 UMIs in any cell in all four patient datasets (leaving 527 transcription factors); and (3) at least 50 transcription factor-expressing cells in all four patient datasets (leaving 508 transcription factors). This filtering was intended to remove sparsely expressed transcription factors that may not be as easily targeted, and to restrict our analysis to transcription factors that are more reliably correlated with our fetal signature.

Next, we used Palantir to compute expression trends for all the transcription factors and the fetal progenitor signature score along the canonical-to-non-canonical DC (see the 'Delineation of canonical to non-canonical tumour axes across patients' section). As we are interested in transcription factors that drive the transition from canonical to fetal cell states, we focused on those with peak expression just prior to entering non-canonical states along the DC of each patient. For patients KG146, KG182, KG150 and KG183, we first identified the position along their DC of the first maxima of the trend in the fetal

progenitor signature calculated along this DC (indicated by the vertical dashed lines in Extended Data Fig. 7a). Maxima were identified as the first inflection point along the first derivative of the trend (that is, when the derivative first changes from positive to negative). Trends for KG150 and KG183 lack a first-derivative inflection point, so we used the position of the maximum value for these patients. We then calculated the Pearson correlation between the expression of each transcription factor and the fetal progenitor gene signature score using only cells at positions along a patient's DC which precede the signature score peak. This yielded four correlation values total for each transcription factor, one for each patient. We focused only on transcription factors with a minimum correlation of $r = 0.5$ in patient KG146 (leaving 14 transcription factors) and $r = 0.2$ in all four patients (leaving 5 transcription factors).

For the remaining six transcription factors, we determined their treatment response in HISC-grown organoids by computing the log-transformed fold change between irinotecan-treated and untreated conditions. log-transformed fold changes were calculated from single-cell data only using cells classified as non-canonical (see the 'Mapping organoid data to patient tumour' section) (Extended Data Fig. 11b).

## Multiplexed immunofluorescence

**Multiplexed tissue staining and imaging.** To maximize capture of all areas of the tumour including the invasion front, full clinical pathology sections prepared for clinical diagnosis were used for imaging (not core punches as typically used for tissue microarrays). Primary antibody staining conditions were optimized using standard immunohistochemical staining on the Leica Bond RX automated research stainer with DAB detection (Leica Bond Polymer Refine Detection, DS9800). Using 4 μm FFPE tissue sections and serial antibody titrations, the optimal antibody concentration was determined followed by transition to a seven-colour multiplex assay with equivalency. Optimal primary antibody stripping conditions between rounds in the seven-colour assay were performed following one cycle of tyramide deposition followed by heat-induced stripping (see below) and subsequent chromogenic development (Leica Bond Polymer Regine Detection, DS9800) with visual inspection for chromogenic product with a light microscope (T.J.H.). Multiplex assay antibodies and conditions are described in Supplementary Table 8.

**Seven-colour imaging assay.** FFPE 4 μm tissue sections of were baked for 2.5 h at 63 °C in vertical slide orientation with subsequent deparaffinization performed on the Leica Bond RX followed by 30 min of antigen retrieval with Leica Bond ER2 followed by six sequential cycles of staining with each round including a 30 min combined block and primary antibody incubation (Akoya antibody diluent/block ARD1001EA), except for HER2, which required a 1 h incubation. From each tissue section, we captured around nine fields of view (FOVs) on average, each 1.34 mm$^2$ in size.

For chromogranin A and OLFM4, detection was performed using a secondary horseradish peroxidase (HRP)-conjugated polymer (Akoya Opal polymer HRP Ms + Rb ARH1001EA; 1:5, 10 min incubation). Detection of all of the other primary antibodies was performed using goat anti-rabbit Poly HRP secondary antibody (Invitrogen, B40962, 1:100, 10 min incubation). The HRP-conjugated secondary antibody polymer was detected using fluorescent tyramide signal amplification using Opal dyes 520, 540, 570, 620, 650 and 690 (Akoya, FP1487001KT, FP1494001KT, FP1488001KT, FP1495001KT, FP1496001KT, FP1497001KT). The covalent tyramide reaction was followed by heat-induced stripping of the primary–secondary antibody complex using Akoya AR9 buffer (AR900250ML) and Leica Bond ER2 (90% AR9 and 10% ER2) at 100 °C for 20 min preceding the next cycle (1 cycle of stripping for HER2 (1 μg ml$^{-1}$, CST, D8F12), CDX2 (0.46 μg ml$^{-1}$, CST, D11D10), TP63 (0.15 μg ml$^{-1}$, CST, D9L7L), PLCG2 (0.2 μg ml$^{-1}$, CST,

E5U4T) and chromogranin A (2.705 µg ml⁻¹, Abcam, EP1030Y); 2 cycles for SOX2 (1.26 µg ml⁻¹, Abcam, SP76), CK20 (0.208 µg ml⁻¹, CST, D9Z1Z), VIM (0.0375 µg ml⁻¹, CST, D21H3), TROP2 (3.29 µg ml⁻¹, Abcam, SP294), CK5 (0.142 µg ml⁻¹, CST, E2T4B) and OLFM4 (0.27 µg ml⁻¹, CST, D1E4M); 2.5 cycles of stripping for Ki-67 (1:100, Biocare, SP6). After six sequential rounds of staining, the sections were stained with Hoechst (Invitrogen, 33342) to visualize nuclei and mounted with ProLong Gold antifade reagent mounting medium (Invitrogen, P36930).

**Multispectral imaging and spectral unmixing.** Seven-colour multiplex-stained slides were imaged using the Vectra Multispectral Imaging System version 3 (Akoya). Scanning was performed at ×20 (×200 final magnification). Filter cubes used for multispectral imaging were DAPI, FITC, Cy3, Texas Red and Cy5. A spectral library containing the emitted spectral peaks of the fluorophores in this study was created using the Vectra image analysis software (Akoya). Using multispectral images from single-stained slides for each marker, the spectral library was used to separate each multispectral cube into individual components (spectral unmixing) allowing for identification of the seven marker channels of interest using Inform v.2.4 image analysis software.

**Twelve-colour imaging assay.** FFPE 4 µm tissue sections of were baked for 1 h at 62 °C in vertical slide orientation with subsequent prestaining of deparaffinization (Leica, Bond Dewax Solution, AR9222) and 35 min of Bond ER2 (Leica, AR9640) antigen retrieval at 100 °C was performed on the Leica Bond RX automated research strainer. The Lunaphore COMET imaging chip (MK03) was placed over the acquisition region based on alignment with regions selected on a haematoxylin and eosin stained image, then loaded into the COMET device and subjected to a first cycle of autofluorescence image acquisition, followed by nine cycles of staining, imaging and elution. Each cycle contained one rabbit primary antibody and/or one mouse primary antibody (HER2 (1 µg ml⁻¹, CST, D8F12); CDX2 (0.46 µg ml⁻¹, CST, D11D10); PROX1 (1.5 µg ml⁻¹, CST, D2J6J); Ki-67 (1 µg ml⁻¹, Abcam, EPR3610); vimentin (0.33 µg ml⁻¹, Thermo Fisher Scientific, V9); PLCG2 (1 µg ml⁻¹, CST, E5U4T); PLCG2 (1 µg ml⁻¹, CST, E5U4T); CHGA (0.13 µg ml⁻¹, CST, 5H7); CK5 (0.213 µg ml⁻¹, CST, E2T4B); SYNC (2 µg ml⁻¹, Thermo Fisher Scientific, Poly); CK20 (0.21 µg ml⁻¹, CST, D9Z1Z); GPC1 (6.47 µg ml⁻¹, Abcam, EPR22580-72)) detected with an Alexa Flour conjugated species-specific secondary antibody (Invitrogen, 10 µg ml⁻¹; Supplementary Table 8). The cyclic staining, imaging and elution was automatically performed by COMET. The optimal concentration, position in the panel and incubation time (4 min for each marker per cycle, but 8 min for HER2) of each marker in the panel was optimized to be concordant with its corresponding single diaminobenzidine immunohistochemistry stain. Optimal immunohistochemistry staining for each marker was determined by an MSK pathologist (T.H.) according to the DAB staining result, as performed on the Leica BondRX Stainer using the BOND Polymer Refine Detection Kit (Leica, DS9800). All antibodies were pre-diluted for COMET analysis with Intercept Antibody Diluent (LI-COR, 927-65001). After each staining cycle, elution of primary and secondary antibodies was performed using elution buffer solution (Lunaphore, BU07-L). For each sample, a 82.5 mm² region of interest was imaged under the Lunaphore slide cover chip at ×20 magnification, and a full channel stacked OME.tif file was generated automatically once all cycles were completed.

**Single-cell segmentation.** We used Mesmer (v.0.12)[81], a deep-learning cell segmentation algorithm, to identify cell boundaries in all COMET and Vectra images. The input to Mesmer (v.0.12) is a single nucleus-stained image and a single membrane or cytoplasm-stained image to define the extent of each nucleus and cell. We used DAPI as a nuclear marker for all COMET and Vectra images. To create an image that will define the boundaries of multiple cell types, we combined the channels

for several cell-type-specific membrane or cytoplasmic markers into a single image by min–max scaling each channel (using the MinMax-Scaler function in the sklearn.preprocessing (v.1.4.2) package with the default parameters) and summing them. For COMET, we combined CK20, HER2, CK5, SYNC (normal and tumour epithelial cells) and VIM (stromal cells). For Vectra panel 1, we used HER2, SOX2, CK20, CDX2 and CHGA (tumour cells) and VIM (stromal cells). For Vectra panel 2, we used TP63, OLFM4, TROP2 and CK5 (tumour cells) and PLCG2 (stromal cells).

We ran Mesmer (v.0.12) on these images with the default parameters to predict cell boundaries, and calculated the cell size, eccentricity and centroid of each cell boundary using the regionprops function (default parameters) in the Python skimage (v.0.23.2) package. When running Mesmer (v.0.12), we first subsampled COMET images by half so that they would fit in system memory (128 GB of memory). For both cell size and DAPI expression, we found the distributions of segmented cells produced by Mesmer (v.0.12) were bimodal and the lower mode contained primarily empty regions and not real cells. We therefore filtered out all predicted cell boundaries below threshold values of 30 px² cell size (estimated from the distribution) and a log₂-normalized DAPI intensity of 11 (COMET), 1 (Vectra panel 1) and 1 (Vectra panel 2) (estimated from the distribution). This resulted in a COMET dataset of 6,852,690 cells across 18 FOVs, a Vectra panel 1 dataset of 6,090,968 cells across 664 FOVs; and a Vectra panel 2 dataset of 5,213,051 cells across 602 FOVs.

**Normalization, background removal and thresholding.** Raw per-cell marker expression levels were first determined by summing the brightness values over all pixels within each cell boundary. To ensure that our downstream analysis was not influenced by cell size, we then divided the per-cell expression for each channel by the cell boundary sizes determined by the regionprops function, as described above. Once normalized, all cells were pooled into cell-by-expression matrices within the same imaging technology and panel (Vectra panel 1, Vectra panel 2 and COMET) for downstream analyses and annotated with patient- and sample-level metadata.

**CK5 background in Vectra panel 2.** To address the high levels of background in CK5 staining, an experienced physician manually annotated 167 FOVs out of 602 to denote whether they have high background signal. To identify high-background images in the remaining 435 FOVs, we first found the highest level of background CK5 signal (10th percentile of expression across all cells in the FOV) within FOVs labelled low-background (around 0.0068). We then classified all unlabelled FOVs with a background CK5 signal of >0.0068 as high background. As a result, 327 unlabelled FOVs with low background were used in later analyses, and 108 with high background were removed. In total, we used 456 FOVs with a low level of CK5 background expression in all figures involving Vectra panel 2.

**Thresholding for tumour markers in Vectra panels 1 and 2.** To call cells as 'positive' for the tumour markers in each panel, we calculated the quantile values of expression across all cells in small increments from 0.0 to 1.0 and found the knee-point of these values (typically between 0.8 and 0.9 for all markers). This method enabled us to avoid using a single quantile threshold for all markers, which would be unreasonable, as markers have different expression distributions across samples. The intuition behind our approach is that the knee-point at which quantile values start increasing rapidly reflects where the population changes from negative to positive. This can also be seen as a way of finding the knee-point in the distribution without fitting an estimated distribution to the data. We validated our labels by comparing the expression of tumour markers with tumour regions that were manually annotated by a physician in a subset of images. We then took all cells called as positive for a marker and repeated the knee-point analysis to identify the subset of cells with high marker expression, which we report as a percentage of all cells positive for any tumour

marker, listed above in the 'Single-cell segmentation' section, from the same panel; we quantify CK20 in panel 1 and OLFM4, TROP2 and CK5 in panel 2 this manner (Extended Data Figs. 3f and 5a–d).

**PROX1+ and CDX2+ cell identification in COMET.** For each COMET image, regions corresponding to normal epithelium and tumour tissue were manually demarcated by an experienced physician based on histology, and we retained only cells belonging to these masked regions for downstream analyses. We included this preliminary filtering step to (1) remove most stromal cells, therefore reducing the computational cost downstream; and (2) distinguish tumour cells from normal epithelium using histological evidence, as this was not possible from marker expression alone (for example, due to expression of stromal marker genes in basal-like tumour cells). As we use only COMET data to compare CDX2+ intestinal and PROX1+ fetal-like cells, we first restricted our analysis to cells expressing either PROX1 or CDX2 above a stringent level of background (99th percentile of expression in unmasked regions calculated separately for each image). We then removed 6 samples with fewer than 1% PROX1-expressing cells, leaving 7 samples. For each of these samples, we further labelled cells as CDX2+, PROX1+ or double+ if their expression of one or both genes was >0.5 s.d. above the mean.

### Organoid experimental methods

**Organoid and cell culture.** For experiments, organoids were collected from Matrigel with 3 mM EDTA in DPBS and, where indicated, were treated with TrypLE (Thermo Fisher Scientific) for 5–10 min at 37 °C and filtered through a 40-µm cell strainer to generate single cells. Cells were plated at a density of 2,000 cells per 40 µl of Matrigel and were incubated at 37 °C for 30 min until the domes solidified. Organoids were cultured in HISC (Advanced DMEM/F12 (AdDF12; Thermo Fisher Scientific), GlutaMAX (2 mM, Thermo Fisher Scientific), HEPES (10 mM, Thermo Fisher Scientific), $N$-acetyl-L-cysteine (1 mM, Sigma-Aldrich), B27 supplement with vitamin A (Thermo Fisher Scientific), primocin (100 µg ml$^{-1}$, InvivoGen), EGF (50 ng ml$^{-1}$, Peprotech), Noggin (100 ng ml$^{-1}$, Peprotech), A8301 (500 nM, Sigma-Aldrich), FGF2 (50 ng ml$^{-1}$, Peprotech), IGF-I (100 ng ml$^{-1}$, Peprotech)) or IGFF (HISC without EGF, Noggin, A-8301, FGF2, IGF-I) medium supplemented with Y-27632. The medium was changed every 3–4 days. Organoids were collected at 7 days for downstream assays. Where indicated, small intact organoids (2–3 days after passage) were treated with 250 nM irinotecan added to HISC medium for 7 days and then collected for downstream assays. Lentiviral production and transduction of organoids was performed as described previously[82]. In brief, HEK293T cells (ATCC) were cultured in DMEM supplemented with 10% FBS, GlutaMAX (2 mM, Thermo Fisher Scientific) and penicillin–streptomycin (100 IU ml$^{-1}$, 0.1 mg ml$^{-1}$, Thermo Fisher Scientific). All cells tested negative for mycoplasma. For scRNA-seq, organoids were collected from Matrigel with 3 mM EDTA in DPBS, dissociated to single cells with Accutase (Sigma-Aldrich) for 30–45 min at room temperature, washed with IGFF medium and processed as described above.

**Inducible knockdown.** For doxycycline inducible *PROX1* knockdown experiments, de novo 97-mer mirE shRNA sequences were synthesized (IDT Ultramers) and PCR amplified using the primers miRE-Xho-fw (5′-TGAACTCGAGAAGGTATATTGCTGTTGACAGTGAGCG-3′) and miRE-EcoOligo-rev (5′-TCTCGAATTCTAGCCCCTTGAAGTCCGAG GCAGTAGGC-3′) as described previously[83]. Sequences (shPROX1-1: TGCTGTTGACAGTGAGCGCGAGGACCAAGATGTCATCTCATAGTGAAG CCACAGATGTATGAGATGACATCTTGGTCCTCATGCCTACTGCCTCGGA; shPROX1-2: TGCTGTTGACAGTGAGCGCCCCCGAGAAAGTTACAGAG AATAGTGAAGCCACAGATGTATTCTCTGTAACTTTCTCGGGGATGCCT ACTGCCTCGGA) were cloned into the LT3GEPIR backbone[83] (Addgene, 111177) and used to generate lentiviral particles to transduce into organoids as described previously[82]. The original plasmid containing

an shRNA sequence to *Renilla* 306 luciferase (shRen.713) was used as a control. Transduced organoids were selected using HISC medium supplemented with 2 µg ml$^{-1}$ puromycin for 7 days. For inducible knockdown experiments, organoids were dissociated into single cells, plated at a density of 2,000 cells per 40 µl of Matrigel and maintained in HISC or IGFF medium supplemented with 2 µg ml$^{-1}$ doxycycline (Thermo Fisher Scientific) unless otherwise specified for 7 days before downstream assays. For organoid initiation and outgrowth assays, organoids containing inducible *PROX1* or control shRNA cultured in HISC medium supplemented with 2 µg ml$^{-1}$ doxycycline for 7 days were dissociated into single cells, stained with DAPI (1 µg ml$^{-1}$, Thermo Fisher Scientific), and live cell (DAPI$^-$) and GFP$^+$ sorted to select for healthy cells expressing the shRNA construct, and plated at a density of 750 cells per 15 µl Matrigel in HISC medium without Y-27632 and supplemented with 2 µg ml$^{-1}$ doxycycline, and imaged at 7 days (BioTek).

**Western blotting.** Approximately 4,000 organoids (3–4 million cells) were recovered from Matrigel using 3 mM EDTA in DPBS, washed, centrifuged (200$g$, 5 min, 4 °C) and lysed with 1× RIPA buffer supplemented with PPI (1:100, Sigma-Aldrich, 04693132001) and benzonase (1:100, Thermo Fisher Scientific, 70-664-3) on ice for 30 min. The protein concentration was determined using Pierce BCA assay (Thermo Fisher Scientific, 23227). A total of 10 µg protein per sample was separated by SDS–PAGE on Bis-Tris polyacrylamide gels (Thermo Fisher Scientific, NW04120BOX), transferred to activated PVDF membranes (Millipore, IPFL00010) and blocked in 3% BSA-TBST solution for 30 min. The membranes were incubated overnight at 4 °C with the following antibodies: mouse anti-β-actin (1:1,000, Thermo Fisher Scientific, AM4302) and rabbit anti-PROX1 (1:1,000, Abcam, ab199359), followed by secondary antibody incubation with 488 anti-mouse and 680 anti-rabbit secondary antibodies (1:5,000, LI-COR Biosciences, 1 h, room temperature) before imaging (Odyssey CLx). Western blots were quantified using ImageJ (v.1.53t)[84].

**Organoid whole-mount immunofluorescence.** Whole organoids were plated in 10% Matrigel and HISC medium on chamber slides with removable wells (Thermo Fisher Scientific, 177380), and incubated at 37 °C for 1 h. The organoids were then fixed in 4% paraformaldehyde in PME solution for 10 min, washed twice with IF buffer (0.2% Triton X-100, 0.05% Tween-20 in DPBS), permeabilized (0.5% Triton X-100 in DPBS for 10 min, room temperature), blocked in 10% normal goat serum (Thermo Fisher Scientific, 50062Z) for 30 min before overnight primary antibody incubation with 1:500 rabbit anti-PROX1 (Abcam, ab199359) in 10% normal goat serum at 4 °C. Organoids were incubated with 1:400 of 594 anti-rabbit antibody (Invitrogen, A11012) for 1 h at room temperature, washed three times with IF buffer and one time with cold DPBS. The wells were removed and the samples mounted using medium containing DAPI (Novus Biologics, H-1200-NB). Imaging was performed on the Zeiss Axio Imager 2 with Apotome structured illumination microscope. PROX1 expression was quantified using the CellProfiler (v.4.2.5) software[85].

**RT–qPCR.** Total RNA was extracted from organoids using the RNeasy Mini kit (Qiagen). Total RNA (3–4 µg) was used to prepare cDNA using the Transcriptor First-Strand cDNA synthesis kit (Roche). qPCR was performed with TaqMan gene expression assay primers (Thermo Fisher Scientific; *PROX1*, Hs00896293_m1; *GAPDH*, Hs02758991_g1; *ELF5*, Hs01063022_m1; *KRT23*, Hs00210096_m1; *TFF3*, Hs00902278_m1; *FABP1*, Hs00155026_m1; *TRPS1*, Hs00936363_m1; *GNAI1*, Hs01053355_m1; *FGF20*, Hs00173929_m1; *NELL2*, Hs00196254_m1; *NRXN3*, Hs01028186_m1; *POMC*, Hs01596743_m1; *KRT20*, Hs00300643_m1; *GPC1*, Hs00892476_m1; *LGR5*, Hs00173664_m1; *TMEM132A*, Hs01096434_m1; *NEUROD1*, Hs01922995_s1; *CHGB*, Hs01084631_m1), the Relative expression was quantified using the $\Delta\Delta C_t$ method,

normalized to the expression of *GAPDH* on the QuantStudio 6 and 7 Pro real-time PCR system (Applied Biosystems).

**Orthotopic xenograft experiments.** All animal experiments were performed in accordance with protocols approved by the Memorial Sloan Kettering Cancer Center Institutional Animal Care and Use Committee (IACUC). NSG (*NOD.Cg-Prkdc^scid^Il2rg^tm1Wjl^/SzJ*, 005557) mice were obtained from the Jackson Laboratory and were transplanted at 6 weeks of age[86]. Mice were maintained in a specific-pathogen-free (SPF) facility under a 12 h–12 h light–dark cycle under controlled temperature and humidity, and given ad libitum access to standard diet or irradiated diet supplemented with 2,500 ppm doxycycline (Modified LabDiet 5053) and water. For orthotopic caecal and intrahepatic injections, OKG146P, OKG146Li, OKG136P or OKG136Li organoids were transduced with pLenti-PGK-Akaluc (AkaLuc) or pLenti-PGK-tdTomato-Akaluc (TdT-AkaLuc) virus (subcloned from pLenti-PGK-Venus-Akaluc(neo)) and HR180-LGR5-iCT plasmids through Gibson assembly (Addgene, 124701, 129094). For intrahepatic injections assessing *PROX1* knockdown, OKG146P and OKG136P lines expressing shRNAs to *PROX1* or *Renilla* control were transduced with pLenti-PGK-Akaluc (AkaLuc). For caecal injections, 200,000 cells from each organoid line were mixed with 50% Matrigel in 10 µl of HISC and injected into the caecal submucosa of NSG mice. For hepatic injections, 500,000 cells from each organoid line were mixed with 50% Matrigel in 10 µl of HISC and injected under the liver capsule of NSG mice. Weekly bioluminescence imaging was performed on the IVIS Spectrum Xenogen instrument (Caliper Life Sciences) and analysed using Living Image software v.2.50. Experimental group sizes were practically determined on the basis of five mice per cage, with $n \geq 5$ age- and sex-matched mice per group. Animals were randomly assigned to experimental groups where relevant. Experimental end point was reached when tumour size exceeded >10% of body mass, or when animals showed any signs of respiratory distress or illness, such as hunched posture, failure to groom or greater than 10–15% weight loss. These limits were not exceeded in any of the experiments. The animals were euthanized at the humane end point, and tissues were collected for downstream assays. Where needed, tissues were fixed in 4% paraformaldehyde. Investigators were blinded to mouse group when scoring histopathological samples.

**Statistics and reproducibility.** In vitro experiments were repeated a minimum of three times independently with similar results.

### Reporting summary

Further information on research design is available in the Nature Portfolio Reporting Summary linked to this article.

## Data availability

All raw and processed sequencing and imaging data, including cell-by-expression matrices, are available through the HTAN Data Portal (http://humantumoratlas.org/publications/hta8_crc_moorman_2024) and links therein. Raw sequencing data have been deposited at dbGaP (phs002371.v6.p1) and linked from the HTAN data portal (https://data.humantumoratlas.org/data-access). The following publicly available datasets were used in this study: Human Space-Time Gut Cell Atlas (https://www.gutcellatlas.org/#datasets) and CRC consensus molecular subtypes centroid data (https://doi.org/10.7303/syn2623706). Source data are provided with this paper.

## Code availability

Python notebooks with markup and code to replicate figures are available at GitHub (https://github.com/dpeerlab/progressive-plasticity-crc-metastasis.git).

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

**Acknowledgements** This work was supported by National Institutes of Health (NIH) grants U2CCA233284 (D.P. and K.G.), U54CA209975 (D.P. and K.G.), R37CA266185 (K.G. and J.S.), K08CA230213 (K.G.), T32GM007739 (E.K.B. and K.L.), U01CA23844401A1 (W.J.), R01EB027498A1 (W.J.), P30CA008748 (MSKCC), Howard Hughes Medical Institute Investigator (D.P.), Gilliam Fellowship (A.M.), NSF GRFP (A.M.), Damon Runyon Clinical Investigator Award (K.G.), Burroughs Wellcome Career Award for Medical Scientists (K.G.), AACR NextGen Grant

for Transformative Cancer Research (K.G.), Stand Up to Cancer Convergence 3.1416 Award (K.G.), Pershing Square Sohn Prize (K.G.), Starr Cancer Consortium (K.G.), Josie Robertson Investigator Award (K.G.), Barbara and Stephen Friedman Predoctoral Fellowship (E.K.B.) and Gerry Metastasis and Tumour Ecosystems Center grants (A.M., Q.J., D.P. and K.G.).

**Author contributions** K.G. and D.P. conceived the study, and designed and supervised the experiments. K.G. conceived and established the trio cohort and matched organoid resources for CRC. A.R.M., R.S. and D.P. designed and developed computational methods. F.C., Q.J., S.H., S.B., J.B., S.A., V.S. and K.L. processed clinical samples. E.K.B., F.C., Q.J., A.M., S.H., S.B. and J.B. designed and performed organoid and mouse experiments. A.R.M., F.W., A.L., C.B., Y.X., F.S.-V. and R.S. performed computational analysis. F.C., M.L., A.S. and K.G. curated patient data. I.M., L.M. and R.C. processed single-cell samples. M.L., A.S., C.F., K.S., E.P., R.Y., T.P.K., W.J., P.P., M.R.W., M.D., J.S., J.G.-A. and K.G. enabled human biospecimen procurement. Y.L. and Z.Y. performed and T.J.H. supervised multiplex immunofluorescence data acquisition. A.R.M., E.K.B., T.N., D.P. and K.G. wrote the manuscript. All of the authors reviewed and edited the manuscript.

**Competing interests** K.G. is listed as an inventor on US patent 11,464,874, and US provisional patent applications 63/478,809 and 63/478,829 on targeting L1CAM to treat cancer, submitted by MSKCC. D.P. is on the scientific advisory board of Insitro. J.S. is a consultant for Paige AI. R.Y. has served on the advisory board for Pfizer, Mirati Therapeutics, Revolution Medicine, Loxo@Lilly and Amgen, received a speaker's honorarium from Zai Lab, and has received research support from Pfizer, Boehringer Ingelheim, Mirati Therapeutics, Daiichi Sankyo, FogPharma and Boundless Bio. J.G.-A. owns stock in Intuitive Surgical. R.C. is a consultant for Sanavia Oncology, S2 Genomics and LevitasBio. The other authors declare no competing interests.

**Additional information**
**Correspondence and requests for materials** should be addressed to D. Pe'er or K. Ganesh.

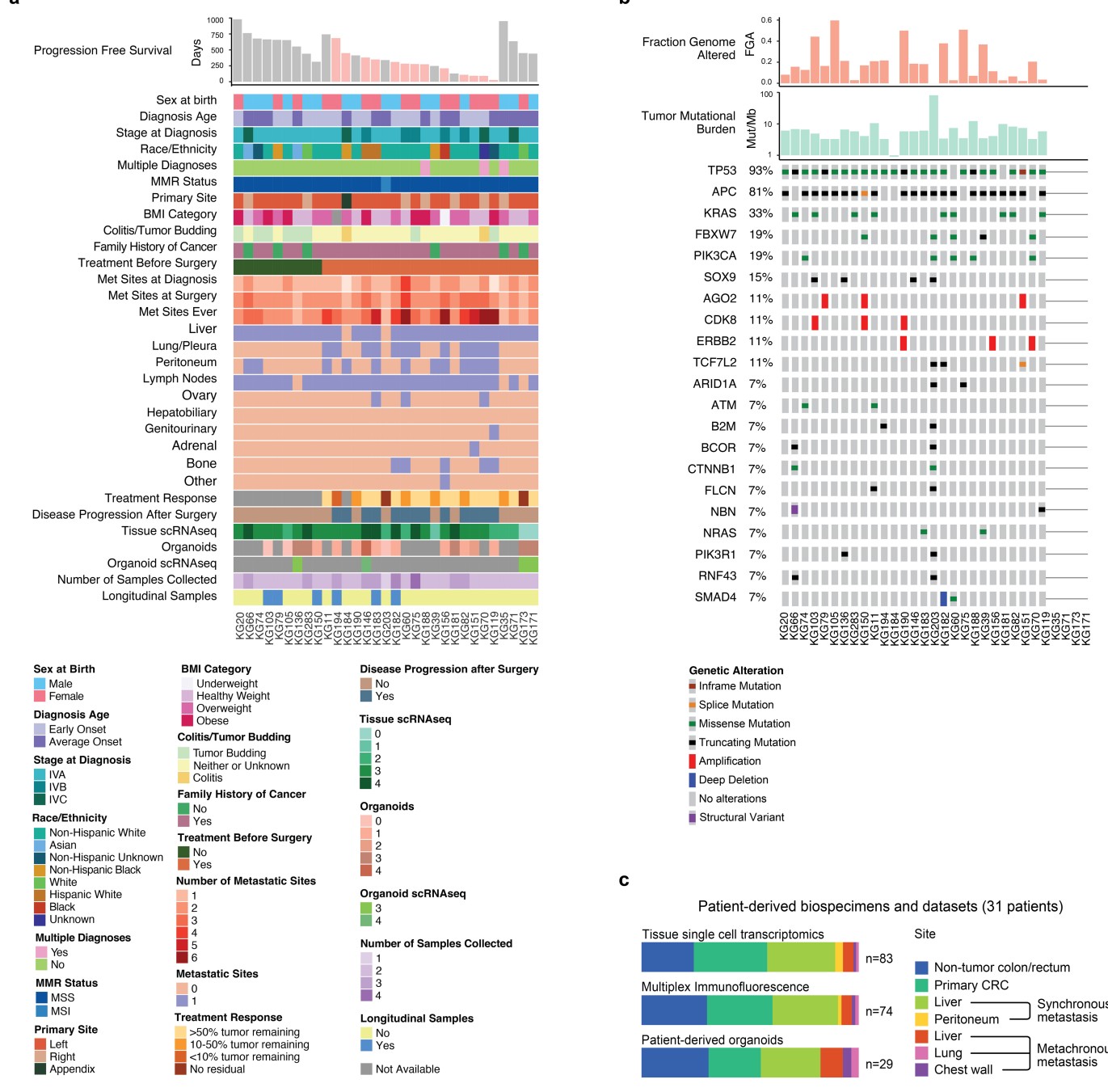

**Extended Data Fig. 1 | Clinical and genomic characteristics of metastatic colorectal cancer biospecimen cohort. a**, Clinical characteristics of patients in the study cohort, including key demographic, clinical, metastatic site, treatment and outcome variables, as well as whether scRNA-seq data was collected from each biospecimen or organoid (also see Supplementary Table 1). Longitudinal samples refer to additional metachronous tumour samples collected at time of progression, subsequent to initial synchronous tissue collection. No more than one biospecimen was sequenced per site. **b**, Genomic features of tumours sequenced using the MSK-IMPACT platform[53]. **c**, Summary of biospecimens and datasets collected.

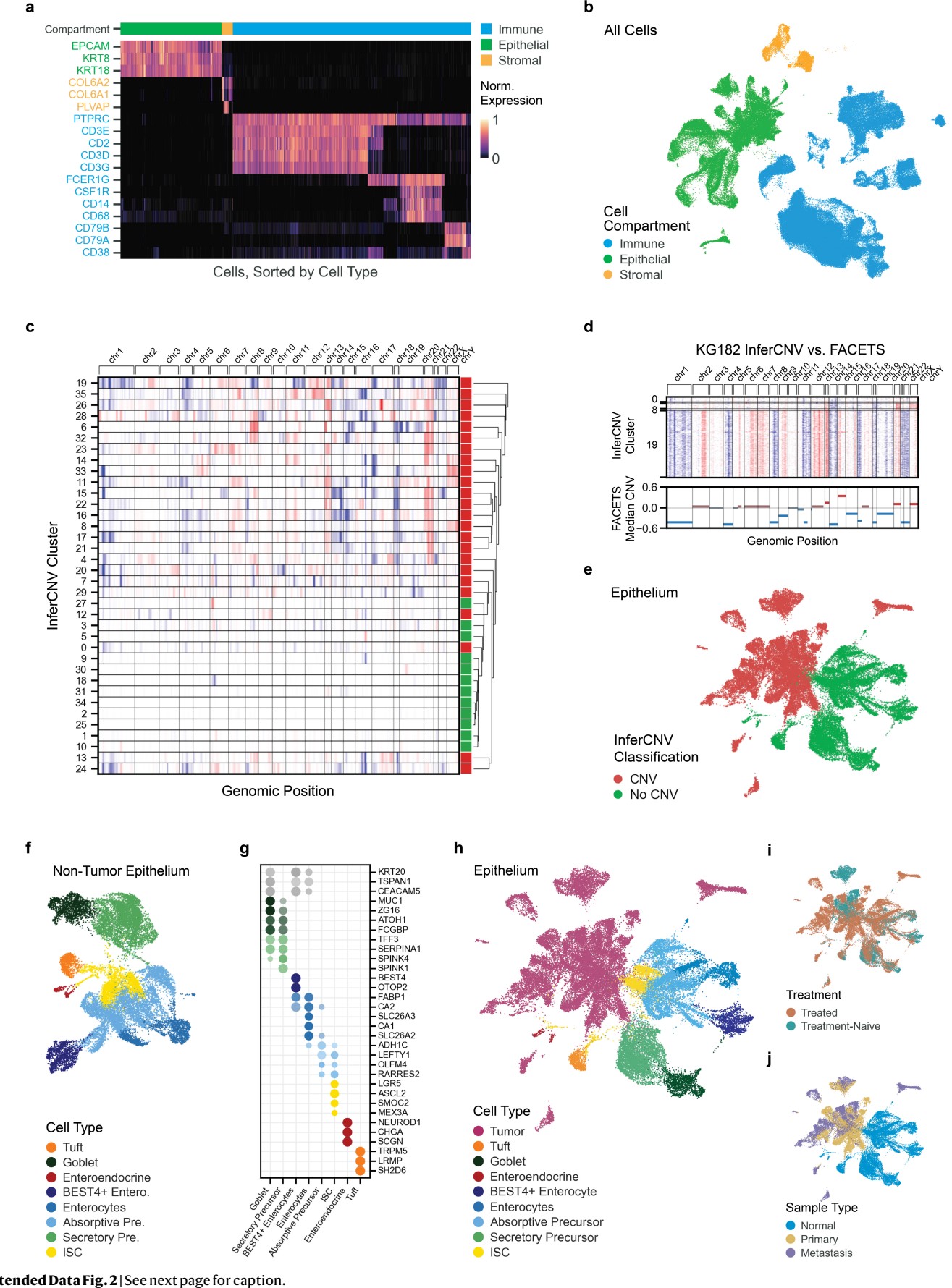

**Extended Data Fig. 2** | See next page for caption.

**Extended Data Fig. 2 | Clinical and genomic features of patient cancers and collected biospecimens. a**, Expression of canonical marker genes (rows) across all cells (columns) grouped by compartment, cell type and cluster. Colour values represent normalized transcript counts (Methods). **b**, Uniform manifold approximation and projection (UMAP) embedding of all cells coloured by immune (111,609 cells), epithelial (47,437 cells) or stromal (5,258 cells) compartment. **c**, Copy number changes calculated with InferCNV[12] and binned by genomic region, per epithelial cell cluster (cells clustered based on inferred copy number matrix; Methods). Rows represent mean inferred copy number relative to diploid reference population of non-tumour epithelium, for each cluster of cells. Cancer cells are called on a per-cluster basis according to their mean copy number profile; right column, green (no CNV) or red (CNV). **d**, Inferred copy number of tumour cells for a representative patient (top, mapping to three inferCNV clusters, including a clearly dominant cluster with

82.3% of cells) compared to copy number values estimated by FACETS[56] using bulk sequencing for a targeted gene panel (bottom; see Methods). **e**, UMAP of all epithelial cells as in Fig. 1d, coloured by CNV classification. **f**, UMAP of all non-tumour epithelial cells (21,297 cells), coloured by cell type annotation (Methods). **g**, Expression of colon epithelial cell type markers across annotated cell types. Dot size scales with proportion of cells in a cell type that express each gene; colour intensity indicates mean z-scored, log-normalized expression of each gene. Genes (rows) are coloured by the cell type (as in **f**) they mark. **h**, UMAP of all epithelial cell scRNA-seq profiles collected from non-tumour colon (20,817 cells), CRC primary tumour (14,402 cells), and CRC metastatic tumour (12,218 cells) samples, coloured by cell type. ISC, Intestinal stem cell. **i**, UMAP of epithelial cells, coloured by tissue origin. **j**, UMAP of epithelial cells, coloured by pre-surgical treatment status (see Supplementary Fig. 1 for patient treatment details).

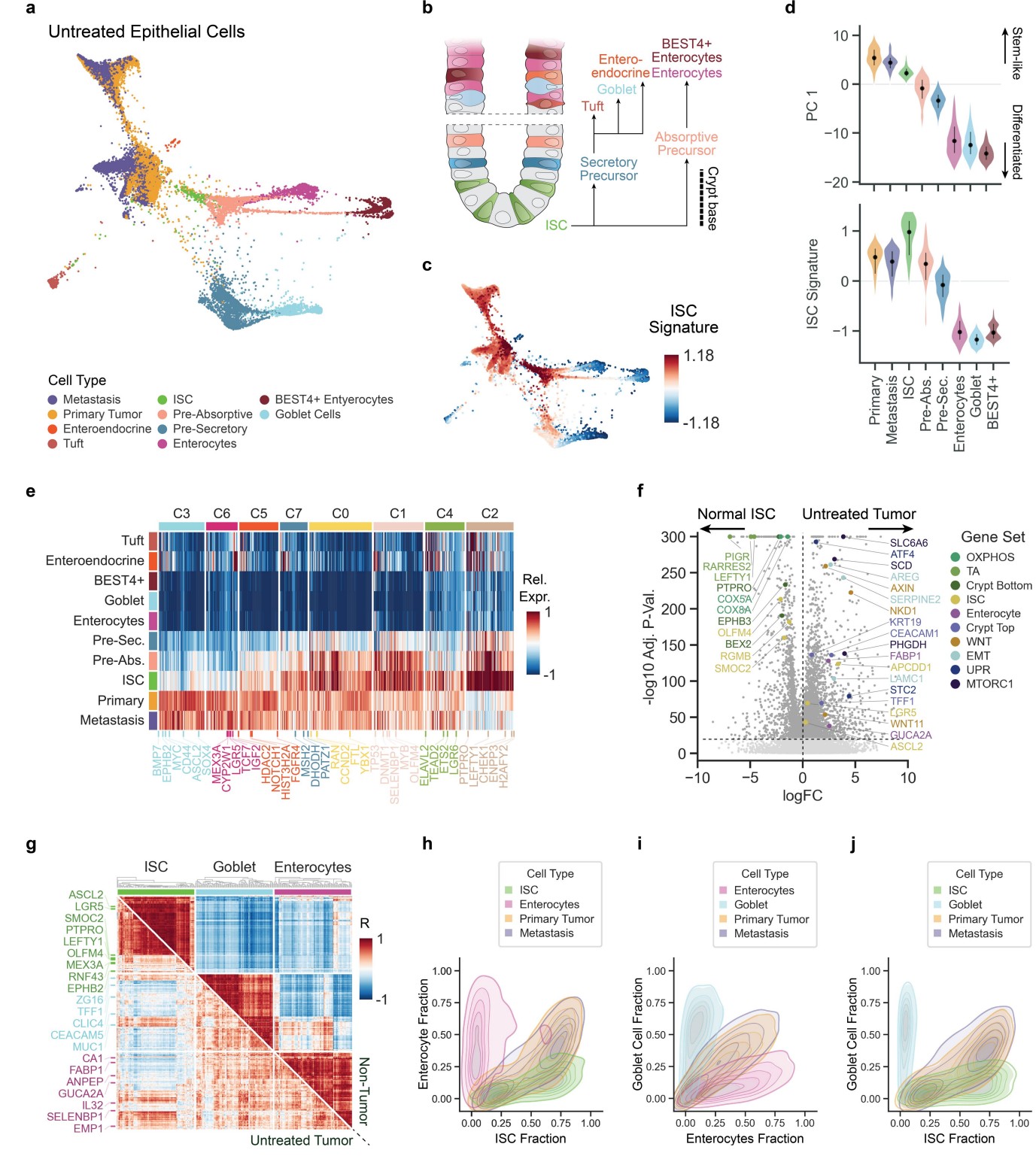

**Extended Data Fig. 3** | See next page for caption.

**Extended Data Fig. 3 | Untreated CRC tumours express early stem cell and mixed lineage programs. a**, Force-directed layout (FDL) of all treatment-naive epithelial cells (13,935 cells), coloured by tumour type (primary, metastasis) for tumour cells or cell type for all other cells. **b**, Major cell types of the colon crypt. **c**, FDL of treatment-naive epithelial cells as in **a**, coloured by our de novo ISC gene signature score (Methods). Scale indicates average, $z$-normalized gene expression (Methods). **d**, Distributions of first principal component (PC1) values (top) or relative ISC signature expression (bottom) for all treatment-naive epithelial cell types (13,935 cells) ordered by median PC1 value and coloured as in **a** and **b**. Gene set enrichment analysis (GSEA) performed on genes ranked by their PC1 loadings was used to determine 'differentiated' versus 'stem-like' directions (Methods; see Supplementary Table 2 for all gene sets and enrichment scores). Dots, median values; lines, interquartile range. **e**, Mean expression of ISC gene signature genes, $z$-normalized per gene across cells, within all treatment-naive epithelial cell types (Supplementary Table 2 and Methods). Rows organized by cell type and columns organized by clustering on the genes. **f**, Genes differentially expressed between ISCs and all treatment-naive tumour cells. Top-enriched gene sets according to GSEA performed on genes ranked by differential expression in tumour cells (Methods); highlighted genes are from leading edge subsets. OXPHOS, oxidative phosphorylation; TA, transit amplifying; ISC, intestinal stem cell; WNT, WNT-beta-catenin signalling; EMT, epithelial-mesenchymal transition; UPR, unfolded protein response; MTORC1, MTORC1 signalling. logFC, log fold-change. -log10 Adj. P-Val., two-sided t-test, $-\log_{10}$-adjusted $p$ value with Benjamini-Hochberg correction. **g**, Gene-gene correlations in cells of normal intestine (upper diagonal), and untreated tumours (lower diagonal) showing expression of both ISC and mixed lineage markers in treatment-naive tumour cells. Genes (some highlighted) consist of the top 100 differentially expressed genes (DEGs) in ISCs, enterocytes, and goblet cells compared to all other non-tumour epithelial cells. Colour values represent average gene-gene correlations calculated for each sample (Methods). **h**, Abundance of ISC- and enterocyte-specific DEGs in normal ISCs and enterocytes, and primary tumour and metastasis cells from untreated patients (Methods). Density plot only includes markers with top quartile expression per gene across all tumour cells. **i**, Same as **h**, for enterocyte and goblet cell markers. **j**, Same as **i**, for ISC and goblet cell markers.

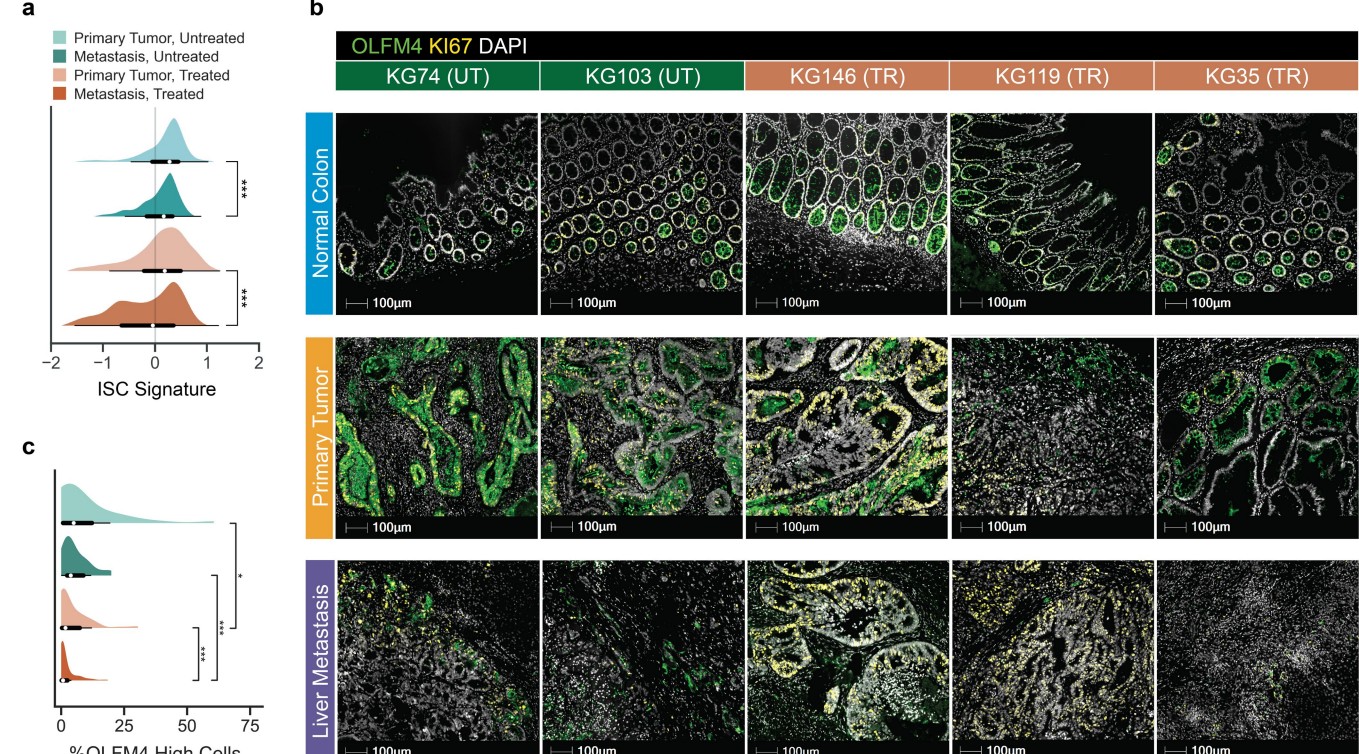

**a**

Primary Tumor, Untreated
Metastasis, Untreated
Primary Tumor, Treated
Metastasis, Treated

ISC Signature

**c**

%OLFM4 High Cells

**b**

OLFM4 KI67 DAPI

KG74 (UT) | KG103 (UT) | KG146 (TR) | KG119 (TR) | KG35 (TR)

Normal Colon

Primary Tumor

Liver Metastasis

**Extended Data Fig. 4 | Decreased expression of ISC programs in CRC metastasis. a**, Histograms of ISC gene signature scores for all tumour cells (26,145 cells). Dots, median; interquartile range (IQR), black bars; 1.5x IQR, black lines. ***, untreated $p = 9.75e-15$; ***, treated $p = 1.11e-157$ two-sided Wilcoxon rank-sum test. **b**, Vectra multiplex immunofluorescence from matched trios of normal colon, primary tumour and liver metastasis, showing OLFM4 (ISC marker), Ki67 (proliferation marker) and DAPI in representative samples from previously untreated (green) and treated (brown) patients. **c**, Histograms quantifying OLFM4-high expressing cells (Methods; untreated primary = 3,61,532, treated primary = 1,180,237, untreated metastasis = 565,014, treated metastasis = 150,1084). Dots, median; IQR, black bars; black lines, 1.5x IQR; *, primary $p = 0.042$; ***, treated $p = 0.003$; ***, metastasis $p = 9.50e-9$; two-sided Wilcoxon rank-sum test.

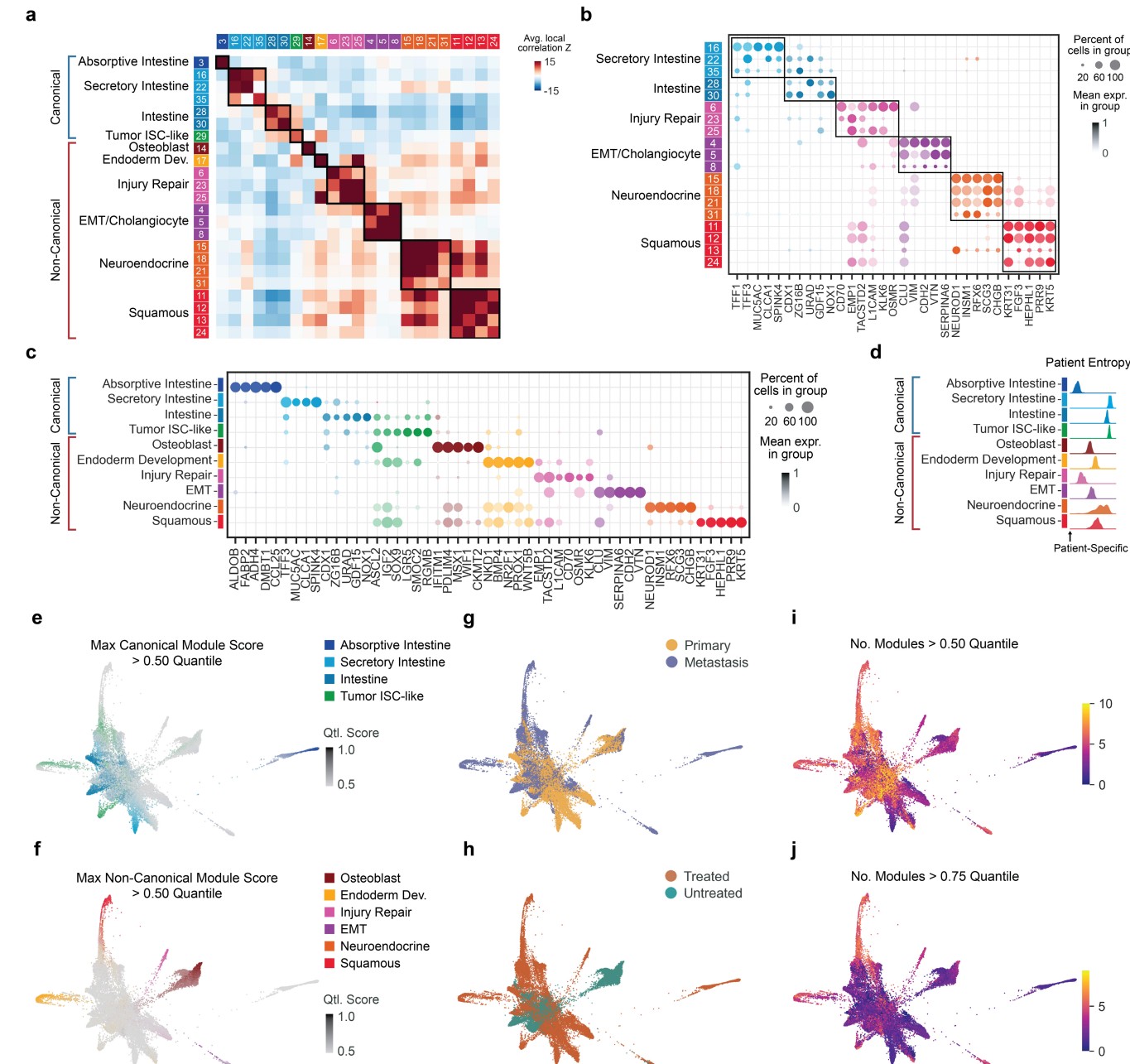

**Extended Data Fig. 5 | Hotspot identifies canonical intestinal and non-canonical gene expression modules in tumour cells from the full metastatic CRC cohort. a**, Average pairwise local correlation score (Methods) between Hotspot modules before grouping. Each entry is the average local correlation between all pairs of genes assigned to the corresponding module pair (row and column). Numbered squares are coloured according to final module annotation after grouping. **b**, z-normalized gene expression within the top-scoring cells for each ungrouped Hotspot module (rows) (Methods). Columns are coloured according to the module assigned to the gene on the x-axis. Dot size indicates proportion of cells in a cluster that score for that gene module; colour intensity indicates mean module score of that cluster. Numbered squares are coloured as in **a**. **c**, z-normalized gene expression within groups of top-scoring cells for

each grouped Hotspot module as in **b**. **d**, Kernel density plots depicting entropy across patients within high-scoring tumour cells for each module (Methods). High entropy indicates that cells with high score for a module (>1 s.d. above mean, calculated across all tumour cells) come from a diverse set of patient samples; low entropy indicates that cells primarily originate from a single patient. **e**–**j**, FDL of all tumour cells (26,145 cells). Each cell is coloured by gene module score for 4 canonical (**e**) or 6 non-canonical (**f**) CRC gene modules, tumour site (**g**), treatment status (**h**), according to its maximum module score, or grey if no score exceeds the module's 50th percentile across all cells, coloured by the number of Hotspot modules for which it scores higher than the 0.5 (**i**) or 0.75 (**j**)[46] quantile module score across all cells.

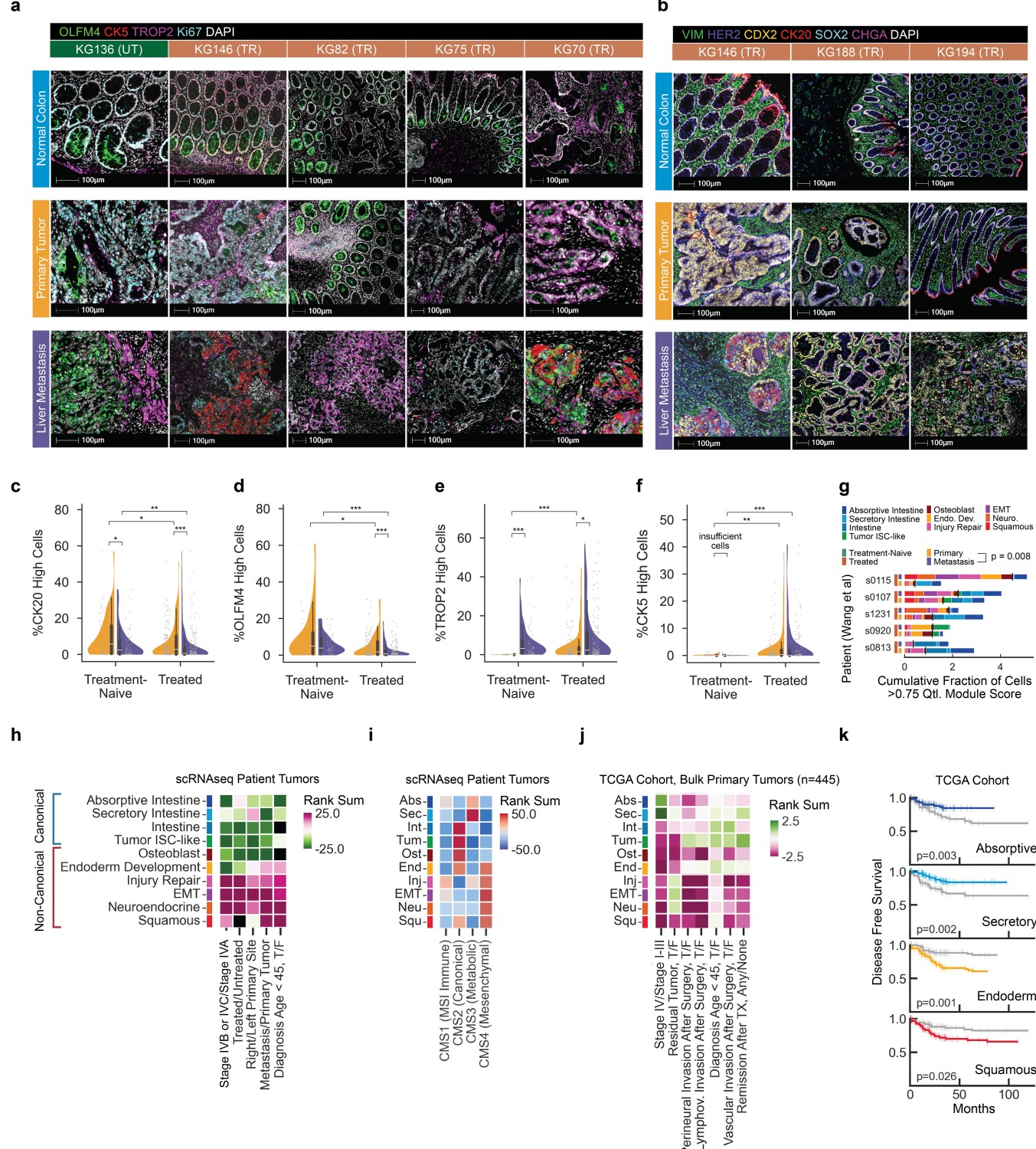

**Extended Data Fig. 6** | See next page for caption.

**Extended Data Fig. 6 | Shift from canonical intestinal marker expression in metastases and therapy-treated samples is associated with poor clinical outcomes. a,b**, Vectra multiplex immunofluorescence from additional representative matched trios stained for canonical intestinal and non-canonical markers, **a**) OLFM4 (ISC marker), CK5 (squamous cell marker), TROP2 (injury repair state marker), Ki67 (proliferation marker) and DAPI (nuclei), and **b**) HER2 (pan-epithelial), CDX2 (intestinal lineage), CK20 (differentiated intestinal), SOX2 (multipotential stem cell) and CHGA (neuroendocrine marker), demonstrating increasingly disordered morphology and increasing expression of non-canonical markers during tumour progression. Patient identifiers (columns) indicate treated (TR, red) or untreated (UT, green) status. **c–f**, Fraction of tumour cells per field of view with high expression ('high expression' for each marker based on minimal expression threshold determined by knee-point decile value from all marker-positive cells) of **c**) CK20 (differentiation; *, treatment-naïve $p = 0.025$; ***, treated $p = 2.92e-4$; *, primary $p = 0.0286$; **, metastasis $p = 0.002$), **d**) OLFM4 (intestinal stem cell; ***, treated $p = 3.24e-4$; *, primary $p = 0.042$; ***, metastasis $p = 8.50e-9$), **e**) TROP2 (injury repair; ***, treatment-naïve $p = 4.65e-15$; *, treated $p = 0.017$;

*** , primary $p = 1.06e-9$) and **f**) CK5 (squamous; **, primary $p = 0.002$; ***, metastasis $p = 7.80e-6$). Orange, primary tumours; purple, metastases; white line, median; black bars, IQR; black lines, 1.5x IQR. $n = 2,096,785$ cells (**c**), $n = 1,408,818$ cells (**d–f**), rank-sum test. **g**, Cumulative fraction of cells expressing >0.75 quantile score for a given module in 5 patients from ref. 28 ($p = 0.008$). One sided rank-sum test (Methods). **h,i**, Enrichment of gene module scores in scRNA-seq data with respect to **h**) patient baseline clinical, pathological and treatment attributes and **i**), CRC consensus molecular subtype (CMS) classifications[29]; Mann–Whitney U rank-sum test. **j**, ssGSEA gene module score enrichment with respect to patient baseline clinical, pathological and treatment attributes for 445 patients with colon (COAD) adenocarcinoma from the TCGA cohort[31]; Mann–Whitney U test. **k**, Kaplan-Meier plots showing disease-free survival for patients in the TCGA cohort with high or low ssGSEA enrichment for the indicated signatures. A patient is signature-high if the enrichment score of their tumour is >1 s.d. above the mean, calculated across all patients, and signature-low if it is <1 s.d. below the mean (log-rank; $n = 147, 158$, 144, and 123 for Absorptive $p = 0.003$, Secretory $p = 0.003$, Endoderm $p = 0.001$, and Squamous $p = 0.026$, respectively).

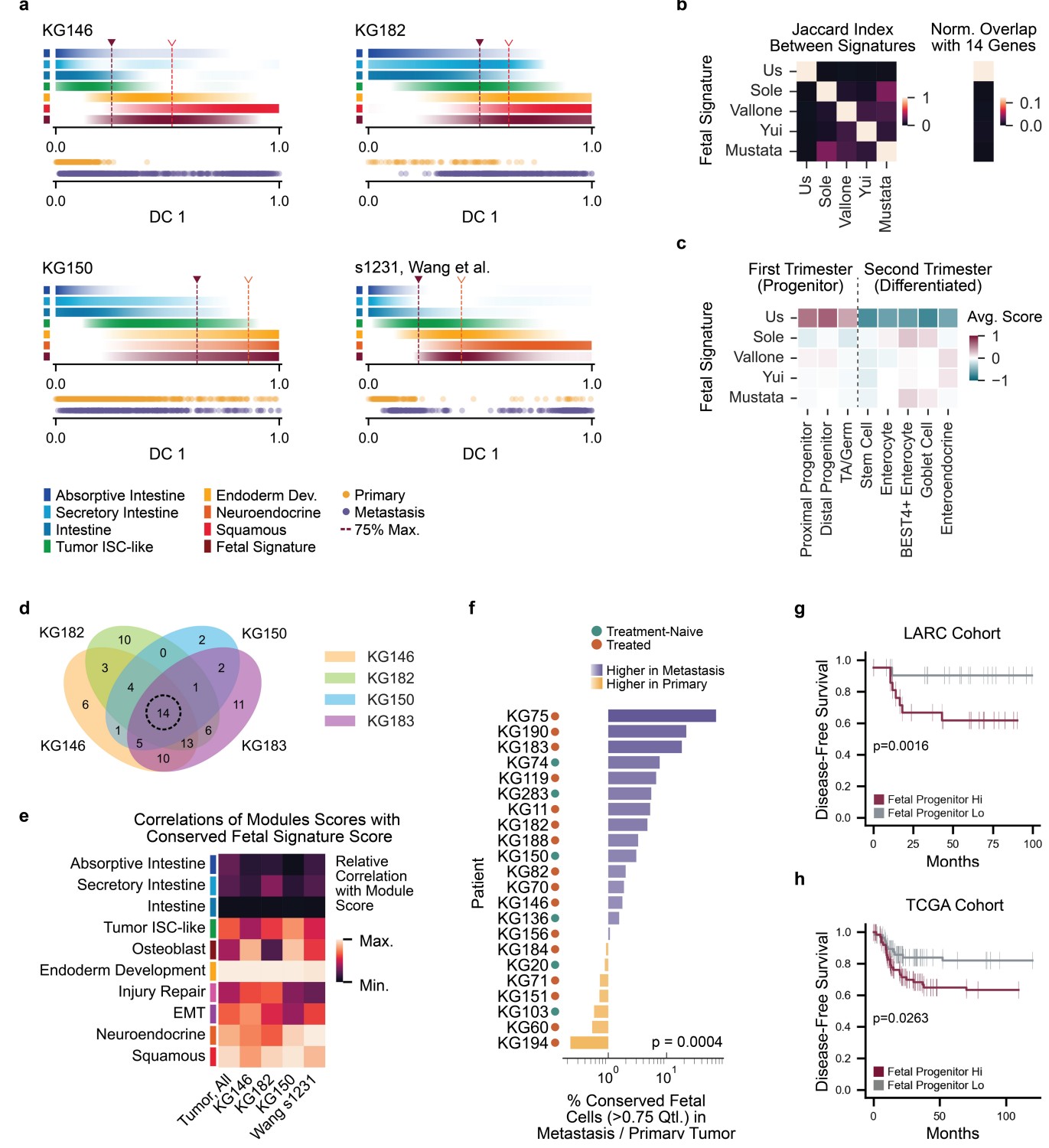

**Extended Data Fig. 7 | Cell state progression is conserved across patients.**
**a**, Top, trends in gene module scores along DC1, observed in all tumour cells from patients KG152, KG180, KG183, and patient s1321 from ref. 28 (Methods). Rows depict module score along DC1 from 20th percentile (white) to maximum value (highest saturation). Fetal signature expression peaks between canonical and NC trends; the closed and open arrowheads correspond to the 75th percentile of fetal and predominant terminal non-canonical module scores, respectively. Bottom, positions of tumour cells along DC1. **b**, Jaccard index (similarity metric defined as no. shared genes / total no. genes in both sets) between the fetal signature and published signatures from mouse (left), and normalized overlap with the 14 shared genes from (**d**) (right). **c**, Average *z*-normalized gene set scores in the cell types of the human fetal dataset, demonstrating overlap of our data with the human fetal signature, compared to previous mouse

signatures. **d**, Overlap of fetal signature genes with Pearson correlation >0.5 across the four patient samples harbouring a substantial number of cells in non-canonical differentiated states. **e**, Relative correlation of core fetal signature with different modules, showing highest correlation for Endoderm Development module across all tumours. **f**, Log-ratio of metastasis- to primary-derived tumour cells that exhibit >0.75 quantile score for the 14-gene conserved fetal signature, for each patient sample ($p = 0.0004$, one-sided rank sum test; Methods). **g**, Kaplan–Meier plots showing disease-free survival for patients in the LARC cohort with high or low ssGSEA enrichment for the fetal progenitor signature. A patient is fetal-progenitor-high if their enrichment score is >1 s.d. above the mean, calculated across all patients, and fetal-progenitor-low if it is <1 s.d. below the mean (log-rank, $n = 42$, $p = 0.0350$). **h**, Same as **g**, for patients in the TCGA cohort (log-rank, $n = 134$, $p = 0.0016$).

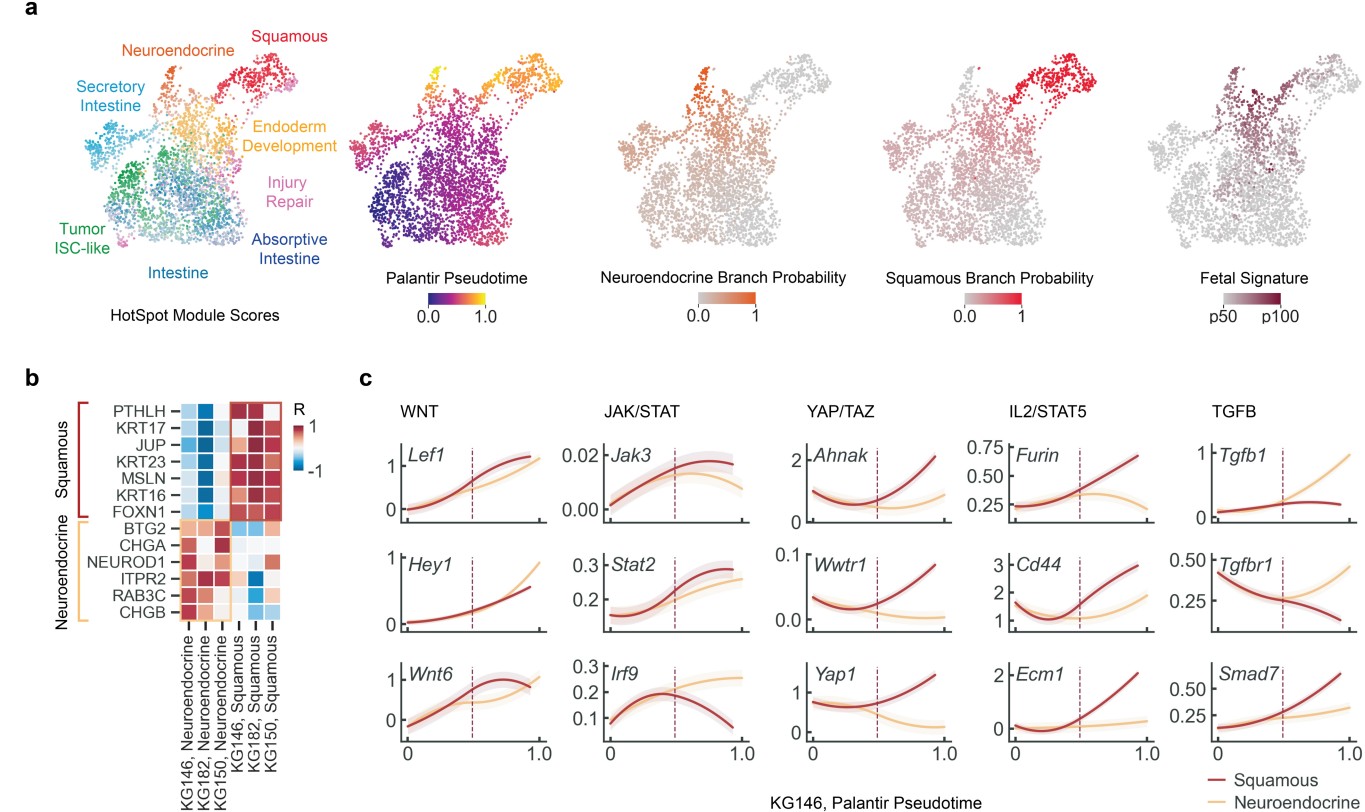

**Extended Data Fig. 8 | Gene expression changes along Palantir pseudotime for squamous and neuroendocrine trajectories. a**, UMAP of 3351 cells from KG146 primary tumour, synchronous liver metastasis, and metachronous lung metastasis. Cell are coloured by maximum module score, or grey if no module score exceeds its 25th percentile (top-left); by Palantir pseudotime (top-middle); by Palantir branch probability for the squamous-annotated (top-right) and neuroendocrine-annotated (bottom-left) branches; or by the fetal signature (bottom-left). **b**, Pearson correlations of Palantir[42] branch probabilities with the expression of genes associated with squamous (top) and neuroendocrine (bottom) transdifferentiation, across tumours from patients KG146, KG182 and KG150. **c**, Gene expression trends along Palantir pseudotime for selected signalling pathways during trans-differentiation of KG146 tumour cells toward squamous or neuroendocrine-like terminal states. Trends are computed using generalized additive models (GAMs) in Palantir. Solid lines represent mean expression, shaded regions represent 1 s.d.

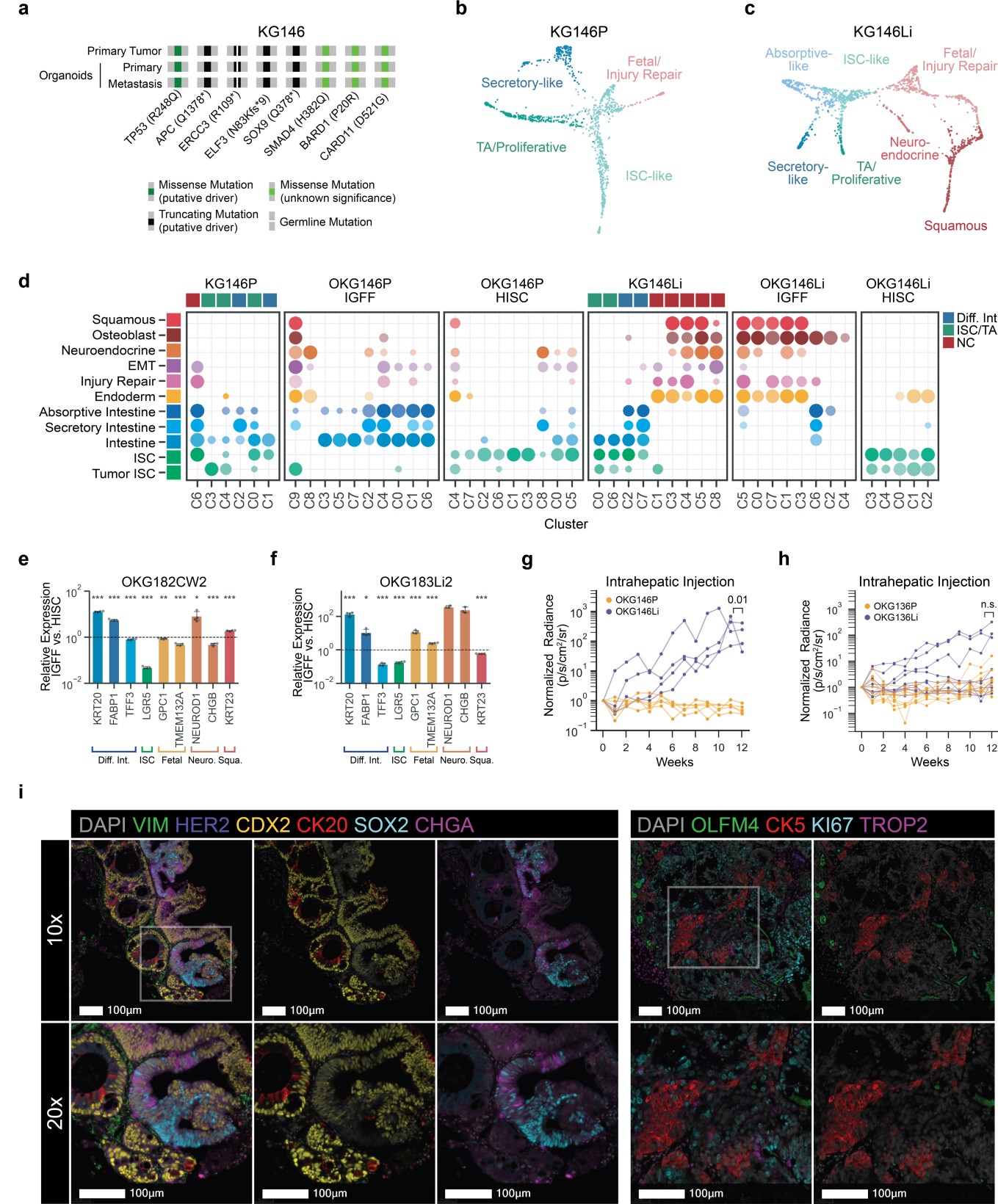

**Extended Data Fig. 9** | See next page for caption.

**Extended Data Fig. 9 | Transcriptomic comparison of patient tumours and organoids deconvolves cell-autonomous and microenvironmental contributions to cell state. a**, MSK-IMPACT results showing concordant mutation profiles of primary tumour and patient-derived organoids for patient KG146. **b**, FDL of KG146 primary CRC tumour cells (*n* = 880) coloured by cluster-level cell state annotations (Methods). **c**, Same as **b**, for KG146Li patient tumour cells (*n* = 1,279). **d**, CRC gene module scores within cell clusters for patient tumour and organoid scRNA-seq samples (Methods). Dot size indicates proportion of cells in a cluster that score for that gene module; colour intensity indicates mean module score of that cluster. Cluster assignments are made within each sample, and scores are calculated separately for all patient tumour samples and all organoid samples. Squares above patient tumour samples indicate the cluster classification (blue, differentiated intestine-like states; green, ISC-like and TA-like states; red, non-canonical states). **e**,**f**, Relative expression of differentiated intestine, ISC, fetal and NC differentiated states in patient-derived organoid line OKG182CW2 (secondary metastasis from chest wall) (**e**) or OKG183Li2 (secondary metastasis from liver) (**f**) cultured for 7 d in IGFF compared to HISC media. RT-qPCR normalized to *GAPDH* mRNA expression level (*n* = 4 technical replicates, Mann–Whitney rank-sum test, \*, *p* < 0.05, \*\*, *p* < 0.01, \*\*\*, *p* < 0.001\*\*\*\*, *p* < 0.0001; KG182CW2 *p*-values, KRT20: 8.09e-09,

FABP1: 6.82e-07, TFF3: 0.001, LGR5: 2.51e-14, GPC1: 0.003, TMEM132A: 1.36e-06, NEUROD1: 0.02, CHGB: 3.05e-06, KRT23: 1.24e-06; KG183Li2 *p*-values, KRT20: 1.59e-4, FABP1: 0.012, TFF3: 5.120e-10, LGR5: 1.22e-10, GPC1: 1.82e-4, TMEM132A: 5.43e-06, KRT23: 1.22e-09. **g**,**h**, Normalized average radiance measured by weekly ex vivo bioluminescence imaging following intrahepatic injection of 500,000 cells in NSG mice, normalized to signal immediately following injection (week 0), for **g**) OKG146P (primary tumour) and OKG146Li (metastasis) lines (*n* = 5 and 4, respectively; *p* = 0.0143, Mann–Whitney two-sided rank-sum test at week 12; *p* = 1.458e-13, normalized radiance > 10, Fisher's exact test, all weeks), and **h**) OKG136P (primary tumour) and OKG136Li (metastasis) lines (*n* = 10 and 10; *p* = 0.39, Mann–Whitney two-sided rank-sum test at week 12; *p* = 2.35e-6, normalized radiance > 10, Fisher's exact test, all weeks). Error bars, s.e.m. **i** Representative Vectra images of OKG146Li intrahepatic xenografts, harvested at 13 weeks post injection and stained for (left) DAPI (nuclei), VIM (vimentin, stroma), HER2 (epithelial differentiation), CDX2 (intestine), CK20 (differentiated intestine), SOX2 (stem cell) and CHGA (neuroendocrine) markers, and for (right) DAPI (nuclei), OLFM4 (ISC), CK5 (squamous), Ki67 (proliferation) and TROP2 (injury repair state) markers. Channels are shown for all markers (left), intestinal markers (centre) and SOX2 and CHGA (right); 20x magnification corresponds to 10x inset.

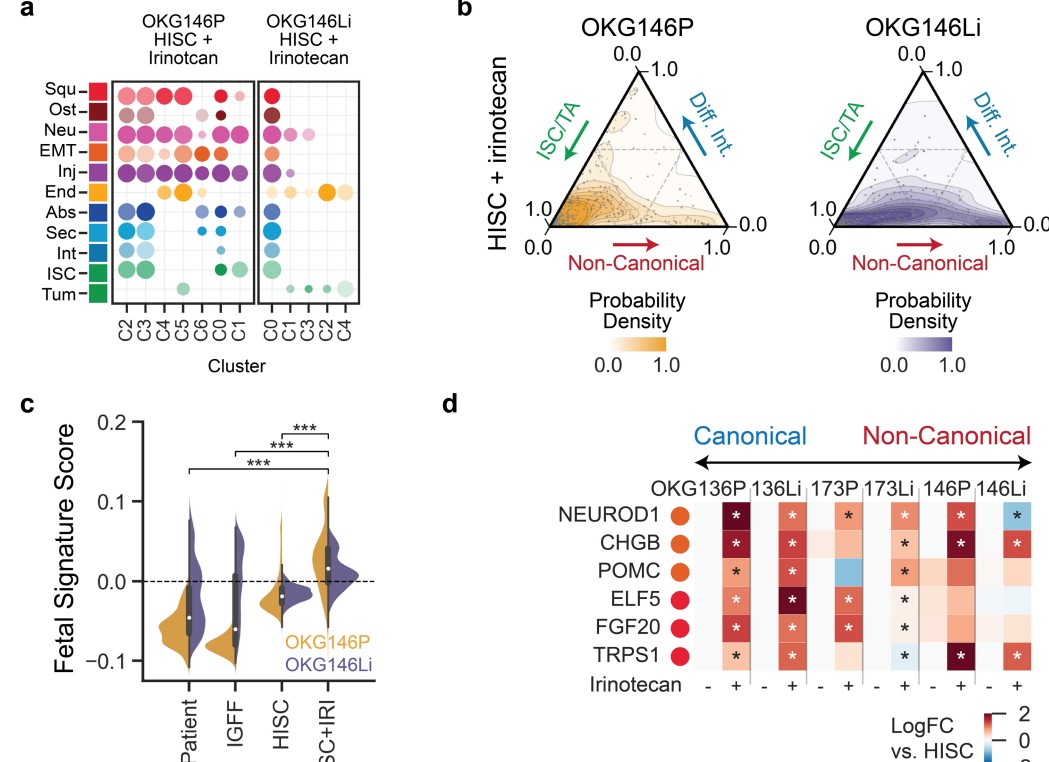

**Extended Data Fig. 10 | Chemotherapy induces expression of non-canonical programs. a**, Gene module scores within cell clusters for patient tumour and organoids cultured in HISC media and treated with irinotecan, as in Extended Data Fig. 9d. **b**, Cell state assignment probabilities per cell for live cells from OKG146 organoids grown in HISC media and treated with 250 nM irinotecan chemotherapy for 7 d. Lines indicate density contours within 5th and 95th percentiles of each distribution, dots indicate individual cells. **c**, Fetal signature score distributions in primary (orange) and metastatic (purple) cancer cells from patient KG146 in patient tumour (left) and patient-derived organoids cultured in IGFF, HISC or HISC media containing 250 nM irinotecan.

Fetal signature is the same as in Fig. 4 and calculated on z-normalized, imputed expression. Dots, median; IQR, black bars; black lines, 1.5x IQR; ***, $p < 0.001$, two-sided t-tests comparing primary and metastasis samples between conditions. $p = 3.48e-26, 2.82e-06, 7.40e-147$ for patient, IGFF: HISC respectively ($n = 12,016$ cells). **d**, Relative expression (log fold-change) of NC differentiation markers in primary tumour (OKG136P, OKG173P, OKG146P) and metastasis (OKG136Li, OKG173Li, OKG146Li) organoids cultured for 7 d in HISC media supplemented with 250 nM irinotecan compared to HISC media alone. RT-qPCR normalized to *GAPDH* expression ($n = 4$ technical replicates, two-sided t-test; *, $p < 0.05$).

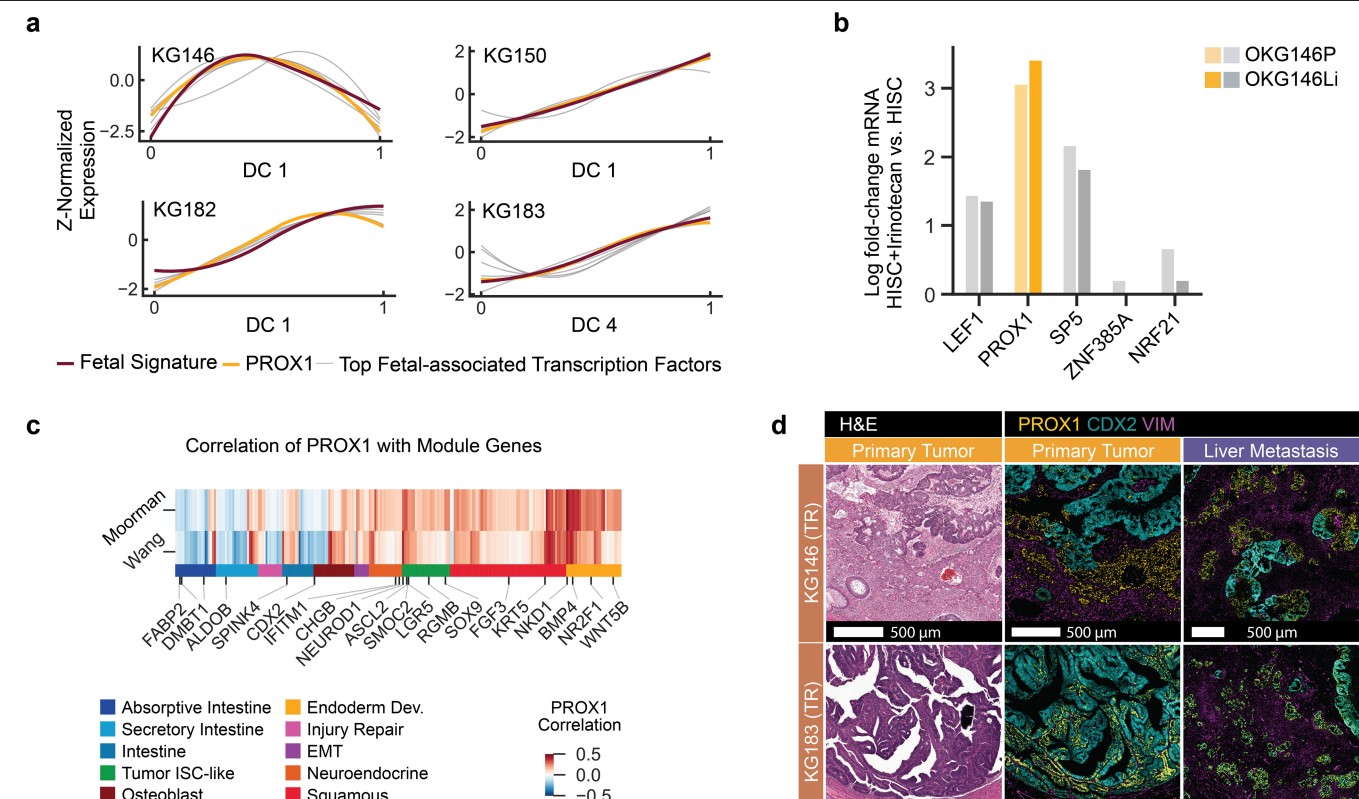

**Extended Data Fig. 11 | Identification of *PROX1* as a fetal-state associated transcription factor. a**, Gene trends along aligned canonical–non-canonical axes, of the 6 top-ranked fetal-associated human TFs in four patients. TFs are ranked according to their mean correlations of imputed gene expression with the fetal signature score in these patients. **b**, Relative expression, from scRNA-seq data, of top 6 fetal-associated TFs in OKG146P (light bars) and OKG146Li (dark bars) organoids cultured for 7 d in HISC media containing 250 nM irinotecan, compared to HISC media alone. **c**, Pearson correlation of

*PROX1* expression with module gene expression across this cohort or the ref. 28 cohort. Genes are grouped by module and ordered within each module by hierarchical clustering; modules are ordered by average correlation with *PROX1* expression. **d**, Hematoxylin and eosin (H&E) and immunofluorescence imaging of PROX1, CDX2 (intestinal lineage) and VIM (vimentin, stroma), showing PROX1 expression in poorly differentiated cells along the invasion fronts of two patient primary tumors.

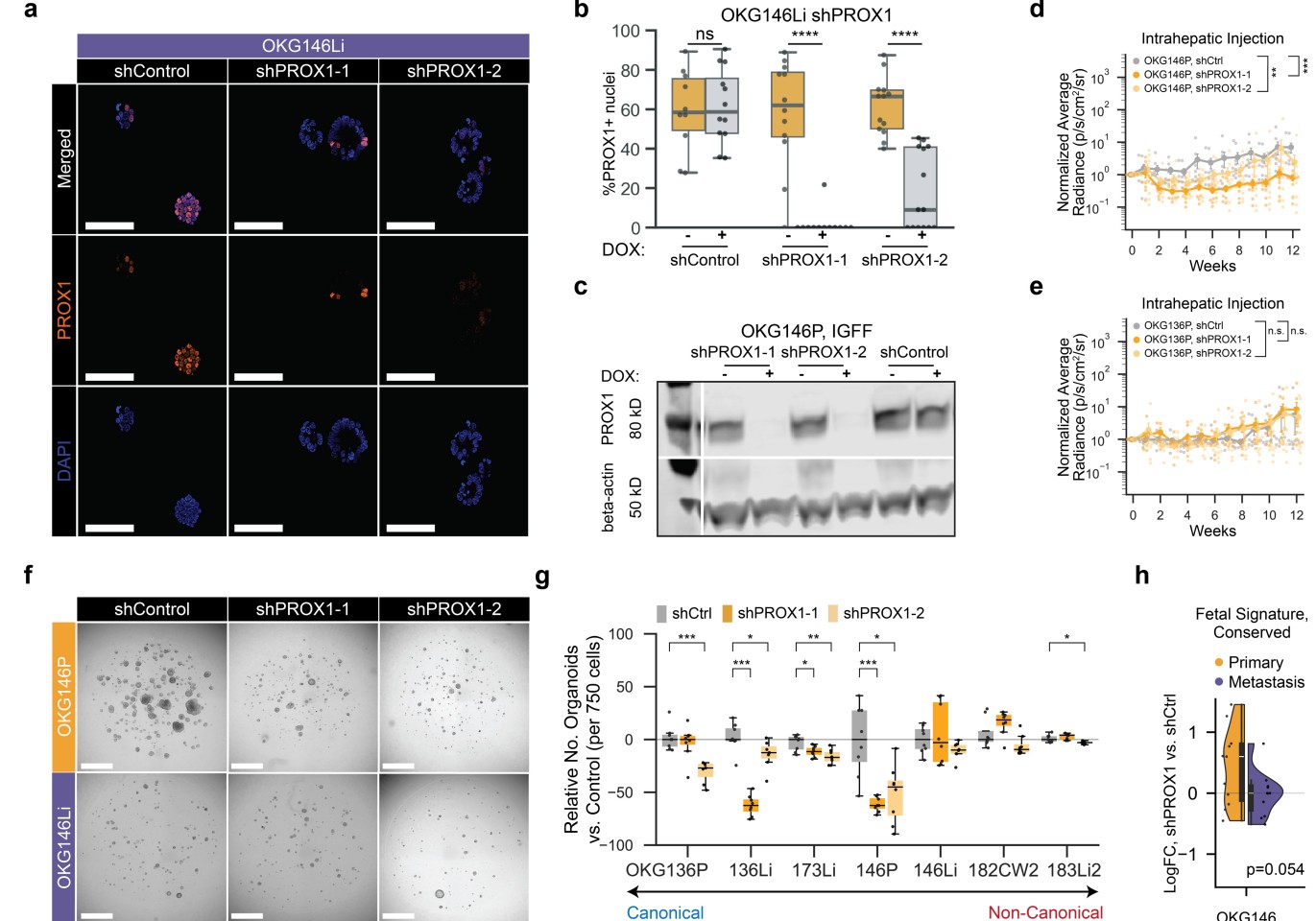

**Extended Data Fig. 12 | *PROX1* is a context-dependent stabilizer of intestinal identity in an epithelial injury-associated, fetal-like state. a**, Representative images and **b**) quantification of whole mount immunofluorescence of OKG146Li organoids expressing shRNAs targeting *PROX1* or control, cultured in HISC media with 2 μg/ml doxycycline for 7 d, stained for PROX1 (red) and DAPI (blue). Scale bar, 100 μm. Boxes, IQR; whiskers, 1.5*IQR. *n* = 10, 12 fields of view for shCtrl; 12, 12 for shPROX1-4; 12, 13 for shPROX1-5, quantified per HISC or HISC + 2 μg/ml doxycycline condition, respectively. ****, *p* < 0.0001, ns = not significant. shControl, *p* = 0.79, shPROX1-1, *p* = 5.62e-07, shPROX1-2, *p* = 1.20e-06, two-sided t-test. **c**, PROX1 protein expression in OK146P organoids expressing doxycycline-inducible shRNAs targeting *PROX1* or control, cultured in IGFF medium with 2 μg/ml doxycycline for 7 d. GAPDH, loading control. See Supplementary Fig. 4 for original gel image. **d**,**e**, Normalized average radiance measured by weekly ex vivo bioluminescence imaging following intrahepatic injection of 500,000 organoids from OKG146P (**d**) or OKG136P (**e**) lines bearing shCtrl, shPROX1-1 and shPROX1-2 (*n* = 9, 10 and 10, respectively, for each organoid line from each patient) in NSG mice, normalized to signal immediately following injection (week 0). Error bars, s.e.m. ** *p* = 0.008, ***, *p* = 0.001, two-sided Mann–Whitney rank-sum test. *p* = 0.0069 for OKG146P shPROX1-1 and *p* > 0.05 for all others, normalized radiance > 10, Fisher's exact test, all weeks. **f**, Representative images of OKG146P and OKG146Li organoids expressing doxycycline-inducible shRNAs targeting *PROX1* or control shRNA, 7 d after plating 750 single cells/15 μl Matrigel in HISC + 2 μg/ml doxycycline (scale bar = 1000 μm). **g**, Organoid initiation capacity (number of organoids formed per 750 single cells/15 μl Matrigel) of organoids in **f**). *n* = 8 independent replicates. Boxes, IQR; whiskers, 1.5*IQR. *, *p* < 0.05, ****, *p* < 0.0001. shPROX1-1 *p* = 0.26, 2.9e-08, 0.010, 0.00018, 0.68, 0.93, 0.90; shPROX1-2 *p* = 9.1e-05, 0.018, 0.0023, 0.0025, 0.069, 0.073, 0.02 for: KG136P, KG136Li, KG173Li, KG146P, KG146Li, KG182CW2, KG183Li2. One-sided t-test, Benjamini-Hochberg correction. **h**, Relative expression, from scRNA-seq data, of the conserved fetal signature in OKG146P (*n* = 3098), OKG146Li shControl (*n* = 2840) and OKG146P (*n* = 3504), OKG146Li (*n* = 2794) shPROX1 organoid lines, 7 d after induction with dox. Dots, median; IQR, black bars; black lines, 1.5x IQR, *p* = 0.054, two-sided *t*-test.

Karuna Ganesh, MD, PhD

# Reporting Summary

## Statistics

For all statistical analyses, confirm that the following items are present in the figure legend, table legend, main text, or Methods section.

| n/a | Confirmed | |
|---|---|---|
| ☐ | ☒ | The exact sample size (*n*) for each experimental group/condition, given as a discrete number and unit of measurement |
| ☐ | ☒ | A statement on whether measurements were taken from distinct samples or whether the same sample was measured repeatedly |
| ☐ | ☒ | The statistical test(s) used AND whether they are one- or two-sided<br>*Only common tests should be described solely by name; describe more complex techniques in the Methods section.* |
| ☐ | ☒ | A description of all covariates tested |
| ☐ | ☒ | A description of any assumptions or corrections, such as tests of normality and adjustment for multiple comparisons |
| ☐ | ☒ | A full description of the statistical parameters including central tendency (e.g. means) or other basic estimates (e.g. regression coefficient) AND variation (e.g. standard deviation) or associated estimates of uncertainty (e.g. confidence intervals) |
| ☐ | ☒ | For null hypothesis testing, the test statistic (e.g. *F*, *t*, *r*) with confidence intervals, effect sizes, degrees of freedom and *P* value noted<br>*Give P values as exact values whenever suitable.* |
| ☒ | ☐ | For Bayesian analysis, information on the choice of priors and Markov chain Monte Carlo settings |
| ☒ | ☐ | For hierarchical and complex designs, identification of the appropriate level for tests and full reporting of outcomes |
| ☐ | ☒ | Estimates of effect sizes (e.g. Cohen's *d*, Pearson's *r*), indicating how they were calculated |

*Our web collection on statistics for biologists contains articles on many of the points above.*

## Software and code

Policy information about availability of computer code

| Data collection | No software was used for data collection |
|---|---|
| Data analysis | The following tools and versions were used for analysis of scRNAseq data: SEQC (v2.7.0), CellBender (v0.1.0), scanpy (v1.9.1), PhenoGraph (v1.5.7), MAGIC (v3.0.0), infercnvpy (v0.4.0), MAST (v1.16.0), gseapy (v0.14.0), matplotlib (v3.6.0), seaborn (0.11.2), numpy (1.22.1), scipy (1.9.1), HotSpot (v0.9.1), lifelines (v0.27.4), Palantir (v1.2), python-ternary (v1.0.8). Mesmer (v0.12), python skimage (0.23.2), sklearn.preprocessing (1.4.2), were used for cell segmentation of multiplex immunofluorescence imaging of patient samples. Software and tools code used for data analysis are provided open source at http://github.com/dpeerlab.<br><br>The following tools were used for ssGSEA analysis of bulk RNAseq data: edgeR (v3.40.2), org.Hs.eg.db (v3.16.0), hacksig (v0.1.2), TCGAbiolinks (v2.26.0), DESeq2 (v1.38.3), GSVA (v1.46.0).<br><br>The following tools were used for multivariate logistic regression models to evaluate associations between DFS, OS and module expression: R packages survival_3.6-4 and survminer_0.4.9<br><br>Multispectral imaging, spectral unmixing, and cell segmentation was performed using Inform 2.4 image analysis software.<br>Quantitative PCR relative expression was quantified using the ddCT method, normalized to the expression of GAPDH on a QuantStudio 6 and 7 Pro real-time PCR system (Applied Biosystems). Western blot quantification was performed using ImageJ (v1.53t). Organoid whole mount immunofluorescence imaging was quantified using CellProfiler (v4.2.5). Bioluminescence images of orthotopic xenograft experiments were analyzed with Living Image (v2.50). |

Python notebooks with markup and code to replicate figures is available at [https://github.com/dpeerlab/progressive-plasticity-crc-metastasis].

For manuscripts utilizing custom algorithms or software that are central to the research but not yet described in published literature, software must be made available to editors and reviewers. We strongly encourage code deposition in a community repository (e.g. GitHub). See the Nature Portfolio guidelines for submitting code & software for further information.

## Data

Policy information about availability of data

All manuscripts must include a data availability statement. This statement should provide the following information, where applicable:

- Accession codes, unique identifiers, or web links for publicly available datasets
- A description of any restrictions on data availability
- For clinical datasets or third party data, please ensure that the statement adheres to our policy

All raw and processed sequencing and imaging data, including cell-by-expression matrices, is available via the HTAN Data Portal (http://humantumoratlas.org/publications/hta8_crc_moorman_2024) and links therein. Raw sequencing data are deposited in dbGaP study accession phs002371.v6.p1 and linked from the HTAN data portal (https://data.humantumoratlas.org/data-access). The following publicly available datasets were used in the manuscript: Human Space-Time Gut Cell Atlas (https://www.gutcellatlas.org/#datasets) and CRC consensus molecular subtypes centroid data (https://doi.org/10.7303/syn2623706).

## Research involving human participants, their data, or biological material

Policy information about studies with human participants or human data. See also policy information about sex, gender (identity/presentation), and sexual orientation and race, ethnicity and racism.

| | |
|---|---|
| Reporting on sex and gender | Clinical data, including baseline demographic data such as sex at birth were abstracted via manual review of patient electronic medical records by board certified medical oncologists (M.L.; K.G.), collected as part of institutional review board approved protocols (MSK IRB #14-244 and #22-404). |
| Reporting on race, ethnicity, or other socially relevant groupings | Clinical data, including baseline demographic data such as race and ethnicity were abstracted via manual review of patient electronic medical records by board certified medical oncologists (M.L.; K.G.), collected as part of institutional review board approved protocols (MSK IRB #14-244 and #22-404). |
| Population characteristics | This study included male (16/31) and female patients with MSS/pMMR metastatic colorectal cancer. Time to each treatment event was calculated from the date of diagnosis to allow for comparison across patients. 17/31 patients had multiple metastatic sites at the time of surgery, and had >50% of tumor sites remaining following surgery. 17/31 patients had early onset colorectal cancer (age of diagnosis <50). 22/31 patients received 5-fluorouracil-based chemotherapy prior to surgery. Clinical MSK-IMPACT targeted exon sequencing was performed on tumor/normal tissue from 27/31 patients and revealed expected mutations. Clinical and genetic characteristics of all patients are listed in Extended Data Fig 1, Supplementary Table 1 and Supplementary Data Fig 1. |
| Recruitment | Patients undergoing synchronous colorectal resection and metastasectomy at MSKCC were identified by chart review, and those who had signed pre-procedure informed consent to MSK IRB protocols #06-107, #12-245, #14-244 and #22-404 for biospecimen collection were selected for this study form February 2019 to September 2021. |
| Ethics oversight | The study protocols were approved by the MSK IRB |

Note that full information on the approval of the study protocol must also be provided in the manuscript.

# Field-specific reporting

Please select the one below that is the best fit for your research. If you are not sure, read the appropriate sections before making your selection.

☒ Life sciences　　☐ Behavioural & social sciences　　☐ Ecological, evolutionary & environmental sciences

For a reference copy of the document with all sections, see nature.com/documents/nr-reporting-summary-flat.pdf

# Life sciences study design

All studies must disclose on these points even when the disclosure is negative.

| | |
|---|---|
| Sample size | For clinical samples, minimum numbers of samples required for statistical power were used in accordance with MSKCC IRB guidelines. For experiments involving analysis of in vivo xenograft tumors, experimental group sizes were practically determined on the basis of cages (five mice per cage), with n≥5 mice per group. Age and sex matched animals were used for all experiments. Minimum number of animals for statistical significance were used. No statistical method was used to predetermine sample size. |
| Data exclusions | Animals that died within 24 hours intrahepatic or cecal injection were assumed to have died due to procedure-related complications and were excluded from further analysis |
| Replication | All attempts at replication were successful. Multiple biological independent experiments were performed wherever feasible as described in |

| Replication | the methods and figure legends. For representative images, each experiment was successfully repeated at least three times under similar conditions. |
|---|---|
| Randomization | Animals were randomly assigned to experimental groups for relevant animal experiments. For other experiments, randomization was not required. |
| Blinding | Investigators were blinded to patient identity or mouse group when scoring histopathological samples. For other experiments, blinding was not required. |

# Reporting for specific materials, systems and methods

We require information from authors about some types of materials, experimental systems and methods used in many studies. Here, indicate whether each material, system or method listed is relevant to your study. If you are not sure if a list item applies to your research, read the appropriate section before selecting a response.

## Materials & experimental systems

| n/a | Involved in the study |
|---|---|
| ☐ | ☒ Antibodies |
| ☐ | ☒ Eukaryotic cell lines |
| ☒ | ☐ Palaeontology and archaeology |
| ☐ | ☒ Animals and other organisms |
| ☒ | ☐ Clinical data |
| ☒ | ☐ Dual use research of concern |
| ☒ | ☐ Plants |

## Methods

| n/a | Involved in the study |
|---|---|
| ☒ | ☐ ChIP-seq |
| ☒ | ☐ Flow cytometry |
| ☒ | ☐ MRI-based neuroimaging |

## Antibodies

| Antibodies used | Multiplexed Immunofluorescence:<br>Primary antibody staining conditions were optimized using standard immunohistochemical staining on the Leica Bond RX automated research stainer with DAB detection (Leica Bond Polymer Refine Detection DS9800)<br><br>SOX2 (1.26 µg/mL , Abcam, SP76)<br>CK20 (0.208 µg/mL, CST, D9Z1Z)<br>CDX2 (0.0575 µg/mL, CST, D11D10)<br>VIM (0.0375 µg/mL, CST, D21H3)<br>Chromogranin A (2.705 µg/mL, Abcam, EP1030Y)<br>TROP2 (3.29 µg/mL, Abcam, SP294)<br>Ki67(Culture supernatant) (1:100, Biocare, SP6)<br>CK5 (0.142 µg/mL, CST, E2T4B)<br>TP63 (0.15ug/mL, CST, D9L7L)<br>OLFM4 (0.27ug/mL, CST, D1E4M)<br>PLCG2 (0.2 ug/mL, CST, E5U4T)<br>For OLMF4 and CHGA, a secondary horseradish peroxidase (HRP)-conjugated polymer was used (Akoya Opal polymer HRP Ms + Rb, 1:5, ARH1001EA)<br>Detection of all other primary antibodies was performed using goat anti-rabbit Poly HRP secondary antibody (1:100, Invitrogen, B40962)<br><br>Lunaphore COMET:<br>HER2 (1 µg/mL, CST, D8F12)<br>CDX2 (0.46 µg/mL, CST, D11D10)<br>PROX1 (1.5 µgmL, CST, D2J6J)<br>Ki67 (1 µg/mL, Abcam, EPR3610)<br>Vimentin (0.33 µg/mL, Thermofisher, V9)<br>PLCG2 (1 µg/mL, CST, E5U4T)<br>CHGA (0.13 µg/mL, CST, 5H7)<br>CK5 (0.213 µg/mL, CST, E2T4B)<br>SYNC (2 µg/mL, ThermoFisher, Poly)<br>CK20 (0.21 µg/mL, CST, D9Z1Z)<br>GPC1 (6.47 µg/mL, Abcam, EPR22580-72)<br>Secondary antibodies are as follows from Invitrogen: D anti-R AF647 (A32795, 10 µg/mL); D anti-M AF647 (A32787, 10 µg/mL); D anti-R AF555 (A32794, 10 µg/mL); D anti-M AF647 (A32787, 10 µg/mL)<br><br>Organoid immunofluorescence imaging:<br> rabbit anti-PROX1 (1:500, Abcam, ab199359)<br> 594 anti-Rabbit antibody (1:400, Invitrogen, A11012)<br><br>Western Blot:<br>mouse anti-Beta-actin (1:1000, Thermo Fisher Scientific, AM4302)<br>rabbit anti-PROX1 (1:1000, Abcam, ab199359) |
|---|---|

488 anti-mouse secondary antibody  (1:5000, LI-COR Biosciences)
680 anti-rabbit secondary antibody (1:5000, LI-COR Biosciences)

| | |
|---|---|
| Validation | Pre-validated antibodies were purchased from reputable commercial sources. All primary antibodies were tested in the relevant application to detect protein without cross-reactivity to non-specific proteins. |

# Eukaryotic cell lines

Policy information about cell lines and Sex and Gender in Research

| | |
|---|---|
| Cell line source(s) | Organoid lines were generated from freshly resected surgical tissue from male and female patients enrolled in MSK approved protocols (MSK IRB #14-244 and #22-404).<br>HEK293T cells used for lentivirus production were purchased from ATCC |
| Authentication | For validation, organoids underwent targeted exome sequencing via MSK-IMPACT. Organoids were STR-verified at the time of establishment and before every experiment. HEK23T cells were not subject to further authentication. |
| Mycoplasma contamination | All  cell and  organoid lines used in this study were routinely tested for mycoplasma contamination (MycoALERT PLUS detection kit, Lonza). All organoid and cell  lines used in this study tested negative for mycoplasma. |
| Commonly misidentified lines<br>(See ICLAC register) | None. |

# Animals and other research organisms

Policy information about studies involving animals; ARRIVE guidelines recommended for reporting animal research, and Sex and Gender in Research

| | |
|---|---|
| Laboratory animals | NSG (NOD.Cg-PrkdcscidIL2rgtm1Wjl/SzJ; stock 005557) mice were obtained from the Jackson Laboratory and were transplanted at 6 weeks of age. Mice were maintained in a specific-pathogen-free (SPF) facility with 12h light-dark cycle under controlled temperature and humidity, and given ad libitum access to standard diet and water |
| Wild animals | No wild animals were used in this study. |
| Reporting on sex | Studies were carried out with female mice. |
| Field-collected samples | No field collected samples were used in this study. |
| Ethics oversight | All animal experiments were performed in accordance with protocols approved by the Memorial Sloan Kettering Cancer Center Institutional Animal Care and Use Committee (IACUC). |

Note that full information on the approval of the study protocol must also be provided in the manuscript.

