## [Peer Review File · Nature]

Manuscript Title: Progressive plasticity during colorectal cancer metastasis

Redactions – unpublished data

Reviewer Comments & Author Rebuttals

Reviewer Reports on the Initial Version:

Referees' comments:

Referee #1 (Remarks to the Author):

This manuscript describes a cohort comprising 31 patients that includes biospecimens of normal colon, primary and metastatic colorectal cancer. The authors employed scRNAseq analysis and a wide array of informatic tools, validation in independent data sets, matching primary tumor and metastatic organoids, and multiplex spatial profiling to show that primary CRC cells are enriched for ISC-like transcriptional programs whereas metastases are associated with so-called 'non-canonical' cell types including non-intestinal lineages, tissue regeneration, drug-tolerance and tumor progression. Importantly non-canonical modules were also associated with poor overall survival. Trajectory analysis also demonstrated that cancer progression involved a transition from intestinal-like states to a fetal progenitor state, before further transitioning into non-canonical squamous-like and neuroendocrine-like states. Finally, the authors identified the transcription factor PROX1 as a potential suppressor of non-canonical plasticity.

The authors have established an impressive resource for CRC research community. Furthermore, the manuscript makes insightful observations that highlight the importance of cellular plasticity in driving tumorigenesis. The manuscript builds on previous studies indicating fetal reprogramming is a key process implicated in gut regeneration and tumorigenesis. Interesting insights include the demonstration that metastatic progression in human CRC is associated with fetal cell states that have the capacity to differentiate into non intestinal states. Although not entirely novel conceptually, these findings represent an important advance in the field that will receive significant attention. However, unlike typical Nature papers, this manuscript is extremely dense and difficult to read through. For the most part the data is descriptive in nature with little functional data providing mechanistic insight. The data is generally of high quality but appears anecdotal at times. Indeed, conclusions are often drawn through analysis of single patient samples and lack validation over multiple patient samples. For these reasons the manuscript may be better suited for a more specialized journal.

Specific points:

1) Previous studies (Vasquez et al. 2022, Cell Stem Cell) have suggested that fetal reprogramming acts as an adaptive process in response to therapy that may promote CRC progression. Although this manuscript shows clear evidence that non-canonical features augment in metastasis relative to primary tumors, it's not clear whether these changes are the result of the natural evolution of tumors or the

impact of therapy. The authors conclude from Figure 3e that patients exhibit different ranges of canonical and non-canonical states. But one could also state that therapy does not on its own correlate with non-canonical gene expression, given that treated patients appear to exhibit both high and low expression of non-canonical genes. Is there any evidence to suggest that therapy is one of the determinants of the non-canonical response? To this point the authors describe on page 9 data from a single patient (KG185, KG182, and KG 146) showing increased non-canonical expression in mets but is it possible to conclude anything from these samples with regards to therapy?

2) In figure 3, the authors state that “multiplex immunofluorescence on 73 tissue sections, including derived from part of the same tissue samples used for scRNA-seq (Fig. 3c,d)”.

Only representative images are shown. These data need to be quantified if they were indeed performed on 73 sections.

3) The authors use multiplex imaging to show correlations between Prox1 expression and canonical and non-canonical markers. Although convincing in the single example shown, the data is not quantified across multiple pairs of primary and metastatic tumors. How consistent are these observations across more cases?

4) Extended data Fig 8c shows gene expression trends in tumor cells from one patient. From this data it appears that all combinations of PROX/CDX2 are possible across the tumor cell landscape. How is it possible to infer any clear function for Prox1 with this level a variability across tumor cells?

5) The authors present data from one primary tumor organoid line showing that PROX1 knock down impairs organoid forming capacity but not in its metastatic counterpart. For reasons stated elsewhere it would be important to validate this across more than one patient to be able to draw meaningful conclusions regarding PROX1 function. In Fig 6f, the authors go on to show that Prox1 knockdown augments 2 non-canonical differentiation markers. A more thorough analysis of intestinal and non-canonical genes should be performed to rule out random effects on these particular genes. Are these the only affected genes that support the conclusion? What can be said about the suppression of non-canonical genes and organoid forming capacity? Does Prox1 maintain the ISC state in primary tumors? This is suggested in the discussion based on murine models. Is there evidence for this in the patient organoid lines? Finally, if Prox1 reinforces the “intestinal” canonical state, one would expect overexpression of Prox1 in metastatic organoids to drive a non-canonical to canonical transition. This should be tested.

Referee #2 (Remarks to the Author):

The authors present a comprehensive biospecimen resource consisting of matched normal tissue, primary tumour, and metastasis. Using single-cell analysis from fresh human samples, multiplex imaging, and organoids derived from patients, the authors studied CRC progression through sequential cell states

focusing on a group of 4 patients displaying high phenotypic plasticity. They described a series of phenotypic transitions, starting with differentiated normal epithelia acquiring a WNT-signalling enriched state (LRG5+ ISCs). They show that the primary tumour program diverges from the normal ISC due to the enrichment of embryonic developmental genes and the presence of lineage promiscuity. Yet, this foetal-like phenotype also transitions to a developmentally primitive state that leads into non-intestinal transdifferentiation in advanced metastatic disease. Mechanistically, the authors identified PROX1 as a transcriptional regulator driving tumour promotion and restricting non-canonical differentiation in advanced disease. Overall, the manuscript describes an extensive analysis of a unique cohort of patients using state-of-the-art technologies, suggesting a multi-step trajectory across different plasticity states during CRC progression. This is a potentially interesting paper providing new insight into the mechanism of CRC development and metastasis. However, there are some major concerns that need to be addressed to strengthen the conclusion.

Specific comments:

Apart from the initial single-cell analysis, the rest of the results come from mostly only one of the patients. For instance, the only multiplex immunofluorescence images presented in the figures comes from one patient (KG146), although the authors state that they performed 72 samples. Similarly for organoids, it was mentioned that 29 samples were obtained from the single cell suspension used for RNA-Seq, while only 3 PDOs were used for functional studies. It'd be important to include more replicates from various patients to strengthen their conclusion.

The data presentation is a bit confusing, as the referencing for the figures does not seem to follow a logical order. There are many instances of mistakes in figure references in the text and several typos throughout the manuscript.

Throughout the study, the authors used their own samples, models and even signatures to generate the conclusions. While the single cell analysis appears to be comprehensive, the analysis is largely based on a few hundreds of cells per patient. Providing evidence that the proposed pattern can be observed in other dataset would greatly enhance the manuscript.

In line 101 page 4 of the manuscript, it is stated that the primary tumour cells are more closely clustered to normal epithelial cells while metastasis are divergent. In Figure 1e it appears that the primary cells overlap partly to the metastatic cells while both are separate to normal cells. Is there any measurement to support the statement in the text regarding the similarity between primary tumour and normal cells?

Could the authors elaborate on how Figure 1d and 1e support the inter and intra-patient tumour phenotypic heterogeneity? (Line 144, page 5)

Figure 2b is referenced in the text in line 165 page 6, to highlight the normal intestinal programs. This is confusing as figure 2b is only a diagram of the cells in the colon crypts. The mentioned programs are

shown in Figure 3a, and the extended data figures 4b, h. Additionally, the genes included in the text (FABP1, KRT20, TFF1 and LYZ) are not highlighted in any of these panels.

In line 119 page 4 the text mentions that genes associated to DNA synthesis and cell cycle progression are upregulated in normal ISCs. The genes mentioned in the text (RFC4, PRIM1, MCM2) are not shown in the extended data figure 3a, where the text refers to. The text continues to highlight the WNT-signalling associated genes and the embryonic developmental genes, these are all contained in the extended data figure 3a except for CYP2W1. Although there is no reference to extended data figure 3 in the text.

Could the authors comment on how extended data figure 4i supports the statement that non-canonical modules occur repeatedly in metastatic cancers? Extended data figure 4i, does not seem to include information from the tumour site. Instead Figure 3f seems to contain that information.

In lines 278-280, it says “We used dynamic time warping to align trajectories from each of the four patients whose tumors exhibit non-canonical differentiation to that of KG146 (possessing the broadest range of cell states)...”. KG146 is one of the four patients, not sure how the authors align the trajectory?

Why do the authors not include all the four patients described in the text in the panels in Figure 4g and extended data figure 6d?

In extended data figure 8a, the expression of the top 6 foetal-associated transcription factors in the organoid systems while exposed to irinotecan was obtained as a relative expression from a scRNA-Seq experiment. The expression of these genes, especially due to their relevance to the conclusions to the manuscript should have been validated independently, e.g. by qPCR.

The extended data figure 8 is not correctly referenced in the text.

In line 361 it is written OKG46Li organoids instead of OKG146Li.

Could the authors elaborate what is the behaviour with the rest of the patients, besides from the 4 highlighted ones? Would the authors consider that all of them would show a similar pattern to patient 146, or is there any unique intrinsic factor making these patients able to reach this enhanced plasticity in their advanced disease. Along this line, could the authors elaborate in the genetic background of the 4 patients that show the non-canonical behaviour?

In line 375, please provide the 6 shortlisted transcription factors.

shPROX1 data organoids in Fig.6 and Extended Fig.8 are interesting. However, is there any functional relevance of the data? Do the PROX1 KD primary organoids show more metastatic potential in vivo? In other words, does PROX1-mediated non-canonical differentiation promote metastasis?

Referee #3 (Remarks to the Author):

This study leverages single cell RNA-sequencing of matched sample trios (normal, primary tumor, metastasis) of treated and untreated CRC patients to describe the plasticity of cancer cell states along an axis of canonical (intestine related) to non-canonical (intestine unrelated) states. The latter states are enriched in metastasis, involve downregulation of classical programs including the LGR5+ intestinal stem cell-like program, are associated with poor clinical outcome and induced by cell-intrinsic (PROX1 loss) as well as extrinsic factors (i.e. therapy). Trajectory analysis suggest that canonical cancer cell states first transition into an ISC-like state and then de-differentiate into an intermediate highly plastic fetal progenitor-like state. This state could serve as a bridge to more differentiated non-canonical states (neuroendocrine and squamous). Lastly, organoid PDX models derived from metastatic tissue containing non-canonical programs can grow in metastatic locations (liver) in contrast to organoids derived from primary locations (lacking non-classical programs).

This is an important and highly interesting study that reflects a major advance in understanding CRC metastasis and the associated cell plasticity. It provides a roadmap for comparison of primary vs. metastasis samples and convincingly shows how important it is to do this in a human setting in comparison to mouse models (CRC liver metastasis in mouse models are largely LGR5+). It is based on profiling of clinical samples that are especially hard to collect (synchronous and metachronous metastasis together with primary and normal), therefore representing a valuable resource. The analysis is rigorous and sound, the manuscript is well written and the figures are clear and intuitive to follow. Overall, I support the publication of this work and expect it to have a significant impact. I have quite a few comments but most are minor and reflect suggestions for the authors discretion.

Major comments:

1. Much of the analysis is based on 4 patients and some of the observations are derived from just one patient, or from experiments with organoids derived from one patient. Yet, the discussion highlights consistency among patients and argues that the transition from canonical to non-canonical is a recurring feature of most patients. Therefore, it would be good to soften this claim and to further evaluate the degree of consistency across all patients and not only the 4 that are highlighted in the analysis.
2. An impressive cohort of multiplex images with relevant markers was created (n=73 samples) but not quantitatively used to back up the claims found by scRNA-seq (see below). Is it possible to utilize that to support at least some of the claims?

Minor comments/suggestions:

1. Given the complexity of Hotspot and the fact that this is not a widely used or easily understood method, it would be good to demonstrate that the main results can be reproduced with an alternative approach.
2. PC1 is initially described as being tightly linked to the ISC signature and then later it is described as a

measure of canonical vs. Non-canonical modules, e.g. in Fig. 3e-f. Both of these claims do not seem to be demonstrated directly and it is confusing that the same PC is interpreted quite differently in the different sections. In the first section this is less of an issue because the ISC signature is also shown directly, but in the later section there does not seem to be a direct analysis of canonical vs. non-canonical modules. It seems to me that adding a more direct measure - such as those in ED fig. 4g - would be helpful, and could fully replace the use of PC1 in fig. 3.

3. Fig. 3C-D and Fig. 5E-F:

a. Not clear which channels are turned on and off in first vs. second row of panel.

b. Higher magnification inlets should be provided to better judge staining quality.

c. It is unclear how large these sections are. Please provide an overview of the sample areas in mm².

4. Fig 2g: it would be better to quantify the exact fraction of cancer cells expressing OLFM4 in primary vs metastasis.

5. Extended Data Fig. 5c: This heatmap does not seem to support the claim that non-canonical programs are correlated with OS from surgery or Diagnosis.

6. Extended Data Figure 5G: It would be better to control for other prognostic factors that might be associated with the programs and therefore confound the results (age, stage, performance status). This could be implemented using a multivariable regression model, such as Cox proportional hazards regression.

7. Line 262 and Figure 4b: It would be helpful to show this plot (or a variant of it) for all 31 patients.

Similarly, Line 309-311: The authors generalize that the fetal progenitor state is an intermediate that bridges to more differentiated cells. However, this is only shown for n=1 patient (KG146). Also: How does the fetal progenitor figure behave if plotted as in extended figure 6A + B?

8. Line 291-293: This claim seems highly speculative. I suggest to systematically compare the abundance of canonical and non-canonical programs of cells from all metastasis locations (i. synchronous liver vs peritoneum, ii. metachronous liver vs lung vs chest wall). Does this result in the same findings as shown in extended data figure 6d?

9. Figure 5d: This finding is very interesting and important. However, again n=1. What about the organoids models representing the other three patients that show high state diversity (KG182, KG150 and KG183)?

10. Line 356-358: "canonical and non-canonical markers appears to be mutually exclusive". Can this be quantified using image analysis?

11. Fig 6C: This shows CDX2^{high} to CDX2^{low}PROX^{high} transition. However, CK5^{high} cells seem to be intermingled with PROX1^{high} cells without obvious spatial segregation. Therefore, this representative picture does not support the claim of transitioning from CDX2^{high} to CDX2^{low}PROX1^{high} to PROX1^{low}CK5^{high} cells.

12. In line 85 + 86 not only the number of samples but also from how many patients they are derived from should be stated

13. Lines 100-101: metastasis cells are more diverged - this seems to be driven by a small subset of tumors rather than a global tendency of all metastases so it might be better to rephrase this sentence.

14. Extended Data Fig 3a: Color code and column clusters unclear

15. Line 104: "enrichment" implies higher epithelial fraction than in the real tissue, which is not shown, so perhaps better to replace with "abundance" or something similar

16. Line 139: it is not clear from the data that selection is involved in the lower ISC of metastatic cells - this could also reflect a dynamic response of metastatic cells to their environment. Similarly in lines 236-237 and lines 437-439 – unclear if selection should be invoked.
17. Extended Data Figure 3b: Almost all of the depicted genes have a $\log_{2}FC < 1$. Some of them (LGR5) are almost 0. With such low FC, it is difficult to ascribe biological significance to those changes. Also, it would be interesting to see how this compares to the treated tumors. What are the pathways up- and downregulated after chemotherapy stress and how do these compare to the untreated pathways?
18. Extended Data Figure 3b: Fig 3a: What genes/modules are behind the non-annotated clusters?
19. Extended Fig 4i: It is not clear what "well-mixed set of patients" (= high entropy, line 659) means concretely. I.e. the secretory intestine program has the highest entropy: What's the fraction of patients that have it? Overall I believe a better way to show this might be with a heatmap that shows per patient the presence and abundance of each of the 10 programs. Sorted by the most redundant programs to the patient specific ones (i.e. from secretory intestine to absorptive intestine.)
20. Fig. 3E: Not clear if this refers to cancer cells from primary only or combined with metastasis.
21. Lines 219-220: this is an important claim, but it is not clear to me how it is demonstrated in the cited figures. In Fig. 3e i don't see a clear association and if there is it should be explained. In Fig. 3f i do see the association but only in a subset of patients and it would be good to spell out more clearly the fraction of patients with such effect. Also ED fig. 5 does not seem to clearly support the claims as the associations vary between modules and do not highlight a clear pattern as seems to be implied in the text.
22. Fig. 3F: Unclear why these samples were chosen. This plot should be shown in the supplement for every pair. Also, it is unclear to me what defines the separation of patients into three groups, from top to bottom.
23. Lines 256-260: it is unclear from fig. 4b (and in general) how high is the consistency of the fetal signature with the changes that occur in the tumor along DC1. The average profile shown at the bottom of this panel implies that there is some consistency but it could still be reflecting a minority of the genes in the signature and accordingly might not fully justify the claims. It would be good to clarify what fraction of genes from the fetal signature are individually consistent with this pattern of highest value in the intermediate state defined by DC1.
24. Lines 272-285: an overall similarity between the trajectories of different tumors is easily apparent from fig. 4. Yet there are some differences that likely reflect real biology. The attempt to align all of the distinct trajectories using a time-warping analysis seems unnecessary to me and the argument for a single stereotypical pattern (as opposed to a consistent overall trend but with distinct tumor-specific variants) seems somewhat exaggerated to me. I would suggest to shorten this part and soften the claim.
25. Extended Fig 5C-D: The Osteoblast program seems to behave more in line with the canonical than the non-canonical programs. Can the authors comment on that?
26. Line 266-267: This claim could be backed up with a volcano plot.
27. Lines 328-333: a mutual nearest neighbor approach is used to map organoid cells to states of the primary tumor. This approach makes it difficult to evaluate how similar the organoid states are to the primary tumor states - even if the similarity is extremely low the mapping might still be defined. Therefore, it would be good to directly demonstrate the similarity, for example by plotting the organoid expression of sets of genes (not individual markers) that define the tumor states. If the similarity is

limited then that might justify mapping only the subset of cells with high similarity.

28. Fig. 6f-i: the inclusion of only two genes as a proxy for each state gives an impression of cherry picking, and accordingly it would be good to also show all signature genes or at least their average.

29. Line 455-456: I agree that this is an important question. I wonder if this can be addressed here:

Given that all patients were profiled by MSK-IMPACT it would be desirable to have an analysis investigating correlations between genomic alterations and cell state abundancies. This should also be done using TCGA: Deconvolution of bulk primaries and correlate cell-state abundancies with genomic alterations (of course possible that the primaries don't show enough signal for the non-canonical states).

Author Rebuttals to Initial Comments:

Reviewer #1

This manuscript describes a cohort comprising 31 patients that includes biospecimens of normal colon, primary and metastatic colorectal cancer. The authors employed scRNAseq analysis and a wide array of informatic tools, validation in independent data sets, matching primary tumor and metastatic organoids, and multiplex spatial profiling to show that primary CRC cells are enriched for ISC-like transcriptional programs whereas metastases are associated with so-called 'non-canonical' cell types including non-intestinal lineages, tissue regeneration, drug-tolerance and tumor progression. Importantly non-canonical modules were also associated with poor overall survival. Trajectory analysis also demonstrated that cancer progression involved a transition from intestinal-like states to a fetal progenitor state, before further transitioning into non-canonical squamous-like and neuroendocrine-like states. Finally, the authors identified the transcription factor PROX1 as a potential suppressor of non-canonical plasticity.

The authors have established an impressive resource for the CRC research community. Furthermore, the manuscript makes insightful observations that highlight the importance of cellular plasticity in driving tumorigenesis. The manuscript builds on previous studies indicating fetal reprogramming is a key process implicated in gut regeneration and tumorigenesis. Interesting insights include the demonstration that metastatic progression in human CRC is associated with fetal cell states that have the capacity to differentiate into non intestinal states.

Comment 1.1. Although not entirely novel conceptually, these findings represent an important advance in the field that will receive significant attention. However, unlike typical Nature papers, this manuscript is extremely dense and difficult to read through. For the most part the data is descriptive in nature with little functional data providing mechanistic insight. The data is generally of high quality but appears anecdotal at times. Indeed, conclusions are often drawn through analysis of single patient samples and lack validation over multiple patient samples. For these reasons the manuscript may be better suited for a more specialized journal.

We thank the reviewer for their positive comments, and appreciate the feedback that the writing is dense and that the manuscript needs additional quantification and validation in more patient samples, cohorts or organoids. We have substantially edited the manuscript to improve clarity, and performed extensive new computational analyses and experiments in additional organoid lines to address these criticisms.

However, we respectfully disagree about conceptual novelty. This study presents the first substantial dataset of human metastases comparing matched normal, primary and metastatic samples from the same patients. Most studies use only mouse models, focus on colorectal tumor initiation (only a handful of papers from the de Sauvage, Batlle and van Rheenen labs examine progression to metastasis), and compare to signatures induced during mouse small intestinal regeneration after wound healing. Beyond the novelty of examining the metastatic transition in human colorectal cancer samples, ours is the only study to rigorously define the fetal signature in patients and the relationship of the fetal state to advanced tumors. By interrogating a human first and second trimester dataset, we have delineated the first human fetal large intestine signature, which corresponds well with our tumor samples. This signature has very limited similarity with the small intestinal regenerative signatures defined in mouse models by others

(new **Extended Data Fig. 7b–d**). Moreover, we find that a core set of 14 genes from the fetal signature are significantly associated with the fetal signature in all four patients harboring a substantial number of cells in differentiated non-canonical states, suggesting that these genes may play a role in facilitating this transformation (Pearson $r > 0.5$ for gene expression and fetal signature trends over DC 1). Notably, the overlap of the mouse signatures with these 14 key genes is even smaller (Sole 0/123, Vallone 2/670, Yui 7/1258 and Mustata 0/247, denoting # shared genes / total # mouse signature genes).

Extended Data Figure 7b–d | Comparison of fetal signature to published small intestine regenerative signatures. **b** Jaccard index (similarity metric defined as # shared genes / total # genes in both sets) between the fetal signature and published signatures from mouse (left), and normalized overlap with the 14 shared genes from **(b)** (right). **c** Average z-normalized gene set scores (rows) in the different cell types (columns) of the human fetal dataset, demonstrating the overlap of our data with the human fetal signature, compared to previous mouse signatures. **d** Overlap of fetal signature genes with Pearson correlation > 0.5 across the four patient samples harboring a substantial number of cells in non-canonical differentiated states.

As highlighted by Reviewer 3, the fetal signature is particularly important, since a primary novel contribution of our work is in demonstrating that the human fetal colonic program serves as a conduit to non-canonical differentiation into squamous and neuroendocrine states in advanced metastases. This transition has not been observed in mouse models of tumorigenesis or metastasis and suggests that the human metastatic fetal-like program has functional properties that are distinct from those acquired during tumor initiation. We further validate our hypothesis using functional experiments with patient-derived organoids. We had included functional data from 6 organoid lines from 4 patients in the first submission, and now add validation in an independent human dataset, as well as many additional experiments: orthotopic growth *in vivo* (testing 2 additional lines), gene induction upon chemotherapy (4 additional lines) or shPROX1 knockdown (3 additional lines), organoid formation efficiency upon shPROX1 knockdown (5 additional lines), as well as entirely new experiments in shPROX1 knockdown lines for scRNA-seq (2 lines) and orthotopic growth *in vivo* (4 lines) (see **General Response** for full list). Our experiments include a total of 8 organoid models from 5 patients.

Comment 1.2a. Previous studies (Vasquez et al. 2022, Cell Stem Cell) have suggested that fetal reprogramming acts as an adaptive process in response to therapy that may promote CRC progression.

Work in mouse models of intestinal regeneration (after colitis, helminth infection, radiation injury, etc.) has shown that small intestinal epithelial cells transiently re-express gene programs derived from differential expression in fetal relative to adult small intestinal organoids^{18,31}. Vasquez and colleagues²⁷ compiled these signatures to define a regenerative stem cell signature (RSC), which is expressed in a subset of TCGA primary tumors and upregulated in primary colon tumors that survive FOLFOX chemotherapy in FOxTROT neoadjuvant study samples²⁷. While this manuscript was in revision, Christopher Tape and colleagues showed that oncogenic driver mutations inhibit the expression of a “RevCSC” signature closely related to RSC, while chemotherapy treatment of organoid:CAF co-cultures promotes re-entry into the RevCSC state^{22,23}. These studies leave significant unanswered questions about the functional significance of these signatures in tumor progression, their relevance to the metastatic transition and outcomes, and their fidelity to human fetal states, which our work addresses:

1. We and others have shown that the revival stem-cell/fetal-like state from mouse small intestinal regeneration is transient, appears before differentiation along an intestinal lineage, and predominates in curable primary tumors. Why this state matters for tumor progression is thus not clear. Our study is the first to demonstrate that a fetal state is neither an endpoint, nor does it precede a relatively benign return to ISC and normal differentiated intestine; rather, it plays an important role in tumor progression, as a conduit to non-canonical differentiation. This suggests that stress-induced adaptive lineage switching (which we demonstrate experimentally in multiple metastasis-derived lines) is the reason advanced metastatic colorectal cancers become resistant to all therapies and ultimately cause death.
2. While prior work focused on mouse small intestinal regeneration signatures or their extrapolation to human primary tumors, our study uniquely addresses how cancer cells change within individual patients as they progress from primary to metastatic tumors. In addition to the fetal state, we also observe the enrichment of non-canonical (e.g. neuroendocrine, squamous) states in metastasis, which are associated with poor outcomes across multiple independent cohorts. Furthermore, we find that the fetal signature is enriched in metastases relative to synchronously resected primary tumors ($p = 0.0004$), though both tumors have experienced the same amount of chemotherapy. These data demonstrate for the first time that while therapy can induce the fetal signature, metastasis *per se*, rather than therapy, can contribute to fetal signature expression across patients.
3. Prior studies derived fetal-like signatures from organoids generated from second trimester (week 2) mouse small intestine. While some genes related to this state were shown to be expressed in primary CRC tumors and organoids in patients^{22,23,27}, the extent to which the mouse small intestinal fetal program recapitulates human fetal large intestinal cell programs was unclear. Our study defines human first and second trimester colon fetal signatures *de novo*, using Human Gut Atlas data⁷, thus establishing ground truth. Notably, we find that the previous mouse signatures correlate poorly with each other or with the human first trimester colon signature, which is a far more primitive state devoid of ISC gene expression (**Extended Data Fig. 7b,c**). We hypothesize that the murine regenerative signatures capture an injury repair state that is highly manifest in fetal organoids grown from disrupted crypts *in vitro*, and that these signatures may be co-expressed with, but are distinct from the ground state fetal program we describe. We characterized distinct 150-gene injury and 113-

gene fetal signatures, which only overlap by 5 genes, and demonstrated that the human colon first trimester signature is expressed at the primary tumor invasion front; it is enriched in matched, synchronously resected metastases; and it can serve as a conduit for both intestinal regeneration and non-canonical lineage plasticity, underpinning the stress-induced adaptability and ultimate lethality of advanced, metastatic CRC.

Comment 1.2b. Although this manuscript shows clear evidence that non-canonical features augment in metastasis relative to primary tumors, it's not clear whether these changes are the result of the natural evolution of tumors or the impact of therapy. The authors conclude from Figure 3e that patients exhibit different ranges of canonical and non-canonical states. But one could also state that therapy does not on its own correlate with non-canonical gene expression, given that treated patients appear to exhibit both high and low expression of non-canonical genes. Is there any evidence to suggest that therapy is one of the determinants of the non-canonical response? To this point the authors describe on page 9 data from a single patient (KG185, KG182, and KG 146) showing increased non-canonical expression in mets but is it possible to conclude anything from these samples with regards to therapy?

We observe fetal reprogramming and the acquisition of non-canonical fates in multiple therapy-naive patients, indicating that this progression mechanism can be associated with metastasis without selective pressure from therapeutics (new **Fig. 3b**). The discovery that metastasis alone is sufficient to enrich for non-canonical differentiation is a critical novel finding. We now include a visualization that we believe better demonstrates that primary tumors tend to harbor more differentiated canonical states (compared to their matched metastasis samples), and that metastases include more differentiated non-canonical states (compared to their primaries, $p = 0.001$) (new **Fig. 3f,g**).

Figure 3b,f,g | Non-canonical modules are present in untreated patients and are enriched in metastases. **b**, Distribution of module labels in cells from combined primary and metastatic tumors from treated and untreated patients; module labels are based on >0.75 quantile score in a given cell for the gene module. Vertical black line divides canonical intestinal cell types (right) from other cell types (left). **f,g**, Log-ratio of metastasis- to primary-derived tumor module proportions in each patient sample, based on accumulation of non-canonical (**f**) or canonical (**g**) modules. Metastatic tumors are significantly enriched for cells expressing non-canonical modules ($p = 0.001$, rank sum test; Methods).

Importantly, we see that the core fetal signature we characterized increases most in metastases relative to their matched primaries—both in untreated patients and in patients who received chemotherapy prior to surgery (new **Extended Data Fig. 7e**).

Extended Data Figure 7e | Cells in the fetal state are more prevalent in metastases. Log-ratio of metastasis- to primary-derived tumor cells that exhibit >0.75 quantile score for the 14-gene conserved fetal signature, for each patient sample ($p = 0.0004$, rank sum test; Methods).

We do observe that tumors from patients selected to receive pre-surgery chemotherapy are more likely to express non-canonical states (though metastases have more non-canonical expression than their matched primaries in these cases, too) (new **Fig. 3f**). Upfront chemotherapy is only administered to those patients whose metastases are too extensive to resect at diagnosis and must be shrunk first⁹. Since we do not have pre- and post-treatment metastatic tumors, we cannot deconvolve whether the greater fraction of non-canonical states in these tumors is due to treatment, or to more aggressive tumor biology that caused patients to present to the oncologist with advanced, unresectable liver metastases.

However, as we showed in the initial submission, treating both primary (146P) and metastasis (146Li) organoids with chemotherapy can induce non-canonical states (original **Fig. 5g,h** and **Extended Data Figs.**

7g and 8a), suggesting that therapy can also drive entry into non-canonical states. To gain more insight into the effect of therapy on non-canonical differentiation, we have treated four additional organoid lines from two more patients (136P, 136Li, 173P, 173Li) with irinotecan chemotherapy, finding similar induction of non-canonical gene expression following therapy (new **Extended Data Fig. 8j**). Metastasis and chemotherapy are thus two forms of tissue disruption and epithelial injury that promote fetal reprogramming and entry into non-canonical states.

Extended Data Figure 8j | Chemotherapy treatment induces non-canonical cell states in primary and metastasis organoids. Relative expression (log fold-change) of non-canonical differentiation markers in primary tumor (OKG136P, OKG173P, OKG146P) and metastasis (OKG136Li, OKG173Li, OKG146Li) organoids cultured for 7 days in HISC media supplemented with 250 nM irinotecan compared to HISC media alone. RT-qPCR normalized to *GAPDH* expression ($n =$

4, 2-sided t-test, *, $p < 0.05$).

Comment 1.3. In figure 3, the authors state that “multiplex immunofluorescence on 73 tissue sections, including derived from part of the same tissue samples used for scRNA-seq (Fig. 3c,d)”. Only representative images are shown. These data need to be quantified if they were indeed performed on 73 sections.

We appreciate this feedback, given by all three reviewers. We have now quantified 1194 fields of view (FOVs) generated using the Vectra technology from 74 tissue sections and 24 patients, comprising 6,090,968 segmented cells for Panel 1 and 5,213,051 segmented cells for Panel 2, and we include additional representative images from more patients. This quantification involved extensive supervised analysis, including using Mesmer⁸ for segmentation model and data normalization to enable cross-FOV comparison (Methods). To help elucidate the impact of metastasis and therapy, we compared primary against metastatic tissues separately in the treated and untreated samples.

Our analysis shows that, in accordance with the scRNA-seq results, metastases display lower intestinal stem (OLFM4) and differentiated (CK20) marker expression, and higher injury repair (TROP2) marker expression (**Extended Data Fig. 5a–d**). These trends hold true when comparing primary and metastasis in the treated and untreated cohorts separately. In treated tumors, we observe lower intestinal stem (OLFM4) and differentiation (CK20) marker expression, and higher injury repair (TROP2) and squamous

(CK5) marker expression than in untreated tumors. Although there are too few squamous cells in untreated samples to compare primary to metastatic tissue, and we found it impossible to robustly quantify neuroendocrine marker (CHGB) due to high and variable background, we observe that cells annotated for either marker are enriched in metastatic samples over primary samples in scRNA-seq data: of all cells with >0.75 quantile expression for i) *CHGB*, 139 are from primary and 455 from metastasis, and for ii) *CK5*, 8 are from primary and 556 are from metastasis.

Extended Data Figure 5a–d | Multiplexed immunofluorescence confirms the shift away from canonical intestinal marker expression in metastases and therapy-treated samples. Fraction of tumor cells per FOV with high expression ('high expression' for each marker based on a minimal expression threshold determined by the knee-point decile value from all marker-positive cells) of **a)** CK20 (differentiation), **b)** OLFM4 (intestinal stem cell), **c)** TROP2 (injury repair) and **d)** CK5 (squamous). Orange, primary tumors; purple, metastases. *, $p < 0.05$; **, $p < 0.01$; ***, $p < 0.001$, rank-sum test.

Comment 1.4. The authors use multiplex imaging to show correlations between Prox1 expression and canonical and non-canonical markers. Although convincing in the single example shown, the data is not quantified across multiple pairs of primary and metastatic tumors. How consistent are these observations across more cases?

We see very clear progression in scRNA-seq pseudotime for multiple samples, but we have only demonstrated spatial progression in the single imaging sample that contained enough cells (given the small field of view on the Lunaphore COMET platform). In our original submission, we demonstrated the tight correlation between *PROX1* and the rise of the fetal signature based on scRNA-seq data (**Fig. 6a**), finding that *PROX1* is most correlated among all TFs in 4 patients (see **Comment 1.5** for more detail). Indeed, we can show that *PROX1* is correlated with genes from non-canonical modules and anti-correlated with genes from canonical modules in our own dataset and an independent human CRC dataset from Wang et al.²⁹ (**Extended Data Fig. 9b**).

Extended Data Figure 9b | *PROX1* expression is correlated with non-canonical and anti-correlated with canonical gene expression. Pearson correlation of the expression of *PROX1* with module genes (labeled as in Fig. 3) across our entire cohort (Moorman) or the Wang et al.²⁹ cohort. Genes are grouped by module and ordered within each module by hierarchical clustering; modules are ordered by average correlation with *PROX1* expression.

Although it is visually compelling, we have now removed the COMET-based claims of spatially distributed progression from the resubmission, given that we only observe it in one instance due to the technology's limited field of view.

Comment 1.5. Extended data Fig 8c shows gene expression trends in tumor cells from one patient. From this data it appears that all combinations of *PROX/CDX2* are possible across the tumor cell landscape. How is it possible to infer any clear function for *Prox1* with this level a variability across tumor cells?

The rise of single-cell genomics has revealed tremendous cellular heterogeneity within tumors^{21,26}, which has provided a powerful opportunity to learn about the genes that drive specific cell states and phenotypes. In particular, gene-gene covariance in gene expression data has long been used as a means of generating hypotheses about molecular interactions. Along these lines, we exploited the heterogeneity in our sample cohort to carry out a rigorous statistical search for candidate regulators of the non-canonical fate; among many thousands of cells and hundreds of transcription factors, *PROX1* most closely correlated with the entire fetal gene signature (113 genes) along the canonical to non-canonical axis in 4 patients (KG146, KG182, KG150 and KG183) (Fig. 6a and Extended Data Fig. 9a).

While profiling-based correlation nominated *PROX1* as a top candidate for regulating this transition in tumors, we want to clarify that our functional inference is based on experimental data. We will make this important point (expanded below) very clear in the text.

- In the original submission, we performed *PROX1* knockdown by inducible shRNA in 4 independent organoid lines (146P, 146Li, 136P, 136Li) from two patients. We validated shRNA-mediated inhibition of *PROX1* using organoid wholemount IF and western blotting (original Extended Data Fig. 8d–f), and demonstrated by qPCR that knockdown induces non-canonical gene programs in

the more canonical 146P, 136P and 136Li organoid lines, but not in 146Li, which exhibits high non-canonical gene expression at baseline (original Fig. 6f–i).

- In the original submission, we also performed functional organoid formation assays using shControl and two sh*PROX1* constructs in two organoid lines (146P and 146Li; original Fig. 6d,e and Extended Data Fig. 8g).
- We now extend our functional analysis using *PROX1* knockdown and overexpression in matched primary and metastasis lines from multiple patients, applying scRNA-seq, qPCR and functional organoid formation and *in vivo* liver metastasis experiments (see new Fig. 6c,d, Extended Data Figs. 8g,h,j and 9e,g,h, and response to Comment 1.6a below).

Comment 1.6a. The authors present data from one primary tumor organoid line showing that *PROX1* knock down impairs organoid forming capacity but not in its metastatic counterpart. For reasons stated elsewhere it would be important to validate this across more than one patient to be able to draw meaningful conclusions regarding *PROX1* function. In Fig 6f, the authors go on to show that *Prox1* knockdown augments 2 non-canonical differentiation markers. A more thorough analysis of intestinal and non-canonical genes should be performed to rule out random effects on these particular genes. Are these the only affected genes that support the conclusion? What can be said about the suppression of non-canonical genes and organoid forming capacity?

We appreciate the reviewer’s suggestions, which we have followed to further strengthen our study of *PROX1* function. Our original Fig. 6 actually showed qPCR data upon *PROX1* knockdown in 4 paired primary and metastasis organoid lines from two patients, KG146 and KG136. As suggested, we have now performed scRNA-seq on sh*PROX1* organoids, and our data clearly show that *PROX1* knockdown in 146P, but not 146Li, strongly induces non-canonical gene expression, providing global support for our claims (new Fig. 6c). However, surveying the entire transcriptome rather than the few originally assayed genes reveals more complex expression patterns: although *PROX1* knockdown causes much more dramatic changes to transcription in primary organoids, including a majority of the non-canonical genes (see **Supplementary Table 5** for a full list), a small set of intestinal differentiation and non-canonical differentiation genes become upregulated in 146Li organoids, suggesting that they are also repressed by *PROX1* in the metastatic context. These data suggest that *PROX1* repression of non-canonical genes is dampened in non-canonical states.

Figure 6c | *PROX1* knockdown causes a strong global shift toward non-canonical states in primary tumor organoids, and a dampened shift in metastasis organoids. Probability of classifying cells as non-canonical based on scRNA-seq data from 146 primary (left) and liver metastasis (right) shControl and sh*PROX1* organoid lines, 7 d after induction with dox. Cells were classified as canonical or non-canonical using a manifold-based classifier that combines methods from Harmony²⁰ and PhenoGraph¹⁶ (Methods). Two-sided t-test, ***, $p < 0.001$.

To test this hypothesis, we engineered additional lines with shControl and two sh*PROX1* lentiviral constructs, for a total of 7 organoid lines spanning the canonical to non-canonical spectrum (136P, 136Li, 173Li, 146P, 146Li, 182CW2, 183Li2). Using our scRNA-seq data from KG146, we selected representative non-canonical genes that change in primary alone, or in both primary and metastasis, after *PROX1* knockdown to profile with qPCR (new Fig. 6d).

Figure 6d | *PROX1* knockdown de-represses non-canonical gene expression in multiple tumor organoid lines. Marker expression relative to shControl (log fold-change) in primary and metastasis organoids upon dox-inducible knockdown of *PROX1*. All genes mark non-canonical differentiation, except for *TFF3* and *FABP1* (canonical intestinal differentiation) and *PROX1*. Organoids were cultured in HISC media containing 2 $\mu\text{g/ml}$ dox for 7 d. Lines are ordered by non-canonical gene induction following *PROX1* knockdown, and RT-qPCR is normalized to *GAPDH* expression ($n = 4$, t-test, Benjamini-Hochberg correction, * $p < 0.05$).

Consistent with observations in our first submission, the response of an organoid line to loss of *PROX1* depends on the phenotypic state of its cells; we find that *PROX1* knockdown induces multiple non-canonical genes (e.g. squamous markers *KRT23*, *ELF5*, *TRPS1*) in more canonical organoids (136P, 136Li, 146P), but it does not upregulate these in the most non-canonical organoid lines (182CW2, 183Li2). In contrast, a subset of non-canonical genes (e.g. *NRXN3*, *POMC*), which retain *PROX1* sensitivity in the 146Li *PROX1* knockdown scRNA-seq data, are upregulated by *PROX1* knockdown even in the most non-canonical

organoids (new **Fig. 6d**). Across organoid lines, *PROX1* knockdown does not consistently alter the expression of canonical intestinal differentiation genes (*TFF3*, *FABP1*).

Comment 1.6b. What can be said about the suppression of non-canonical genes and organoid forming capacity? Does *Prox1* maintain the ISC state in primary tumors? This is suggested in the discussion based on murine models. Is there evidence for this in the patient organoid lines?

As suggested, we used organoid forming capacity as an assay of ISC function in our panel of 7 sh*PROX1* organoid lines spanning the canonical to non-canonical spectrum. As with our qPCR data, we find that this capacity depends on each line's underlying cell states; more canonical organoid lines (136P, 136Li, 173Li, 146P) form fewer organoids upon *PROX1* knockdown, whereas more non-canonical lines (146Li, 182CW2, 183Li2) are *PROX1*-independent (new **Extended Data Fig. 9e**).

Extended Data Figure 9e | *PROX1* knockdown reduces organoid-forming capacity in organoid lines with more canonical cell states. Relative organoid initiation capacity (number of organoids formed per 750 single cells/15 μ L Matrigel) in sh*PROX1* compared to shControl lines at 7 d of culture in HISC media supplemented with 2 μ g/ml dox. Organoid lines are ordered by non-canonical gene induction following *PROX1* knockdown, as described in response to **Comment 1.6a**. $n = 8$; one-sided t-test, Benjamini-Hochberg correction; *, $p < 0.05$; $p < 0.01$; ***, $p < 0.001$.

Together, our new scRNA-seq analysis, qPCR survey and organoid formation assays confirm and refine the role of *PROX1* as a fetal-state-induced TF that restricts non-canonical differentiation and thus enables re-differentiation into an intestinal lineage. Dissecting the role of *PROX1* across the canonical to non-canonical phenotypic continuum has provided valuable insight into how inter-patient and inter-site heterogeneity within patients can inform the context-dependent function of individual molecular mediators in controlling cell state. In canonical tumors, *PROX1* maintains the ISC state by inhibiting non-canonical differentiation, especially along the squamous lineage, whereas non-canonical tumors become *PROX1*-independent, with little change in non-canonical expression or ISC function upon *PROX1*

knockdown. Our novel insights underscore the importance of employing tumors and organoid models from multiple patients, and in studying advanced metastases that capture this inter and intra-patient heterogeneity.

Comment 1.6c. Finally, if Prox1 reinforces the “intestinal” canonical state, one would expect overexpression of Prox1 in metastatic organoids to drive a non-canonical to canonical transition. This should be tested.

Given that the underlying cell states in an organoid line (which depend on many regulators) determine *PROX1* function, and that *PROX1* knockdown in non-canonical organoids is less effective at derepressing non-canonical gene expression, we did not expect that *PROX1* restoration alone would be sufficient to drive a non-canonical to canonical transition in metastatic organoids. Nevertheless, to test this hypothesis, we engineered 4 organoid lines from 2 patients (136P, 136Li, 146P, 146Li) to overexpress *PROX1* or *GFP* control. Organoids were treated with chemotherapy to induce non-canonical gene expression and assayed by qPCR, revealing that *PROX1* overexpression significantly inhibits induction of squamous and neuroendocrine genes (**Fig. R1a,b**). Along with our knockdown results, these data further underscore how *PROX1* inhibition of non-canonical differentiation depends on dose and cell context.

We next investigated the effect of *PROX1* overexpression on ISC markers, finding modest induction of *LGR5* across all 4 lines, and variable effects on *ASCL2* and *OLFM4* expression (**Fig. R1b**). Functionally, *PROX1* overexpression increases organoid formation (canonical ISC function) in the most canonical line 136P, but decreases it in the other three lines (136Li, 146P, 146Li), suggesting distinct functions that depend on phenotypic context (**Fig. R1c**). Together, our data show that *PROX1* overexpression is not sufficient to fully reactivate a functional ISC program.

Our extensive data in organoid lines spanning the canonical to non-canonical spectrum demonstrate unequivocally that *PROX1* functions to repress non-canonical gene expression, and that this function is lost in advanced metastasis, but that *PROX1* restoration alone is insufficient to promote a non-canonical to canonical transition. Although our overexpression results are consistent with our interpretation of *PROX1* function based on knockdown, we prefer not to include them in our manuscript, as they do not advance our understanding of *PROX1* enough to justify overloading our already full paper.

Figure R1 | *PROX1* overexpression inhibits non-canonical gene expression and affects ISC maintenance in a context-dependent manner. **a)** Non-canonical marker expression in primary and metastasis organoids cultured in HISC with 250 nM irinotecan for 7 d, relative to HISC alone (log fold-change). **b)** ISC (*LGR5*, *ASCL2*, *OLFM4*), squamous (*ELF5*, *KRT23*, *FGF20*, *TRPS1*) and neuroendocrine (*NEUROD1*, *CHGB*, *POMC*) marker expression in *PROX1*-overexpressing organoids relative to GFP-overexpressing controls (log fold-change), all grown in HISC with 250 nM irinotecan for 7 d. RT-qPCR in **(a,b)** is normalized to *GAPDH* expression ($n = 4$, t-test, Benjamini-Hochberg correction, *, $p < 0.05$). **c)** Relative organoid initiation capacity (number of organoids formed per 750 single cells/15 μ L Matrigel) in *PROX1* compared to GFP overexpression lines at 7 d of culture in HISC media ($n = 8$; one-sided t-test, Benjamini-Hochberg correction; *, $p < 0.05$; ***, $p < 0.001$).

Reviewer #2

The authors present a comprehensive biospecimen resource consisting of matched normal tissue, primary tumour, and metastasis. Using single-cell analysis from fresh human samples, multiplex imaging, and organoids derived from patients, the authors studied CRC progression through sequential cell states focusing on a group of 4 patients displaying high phenotypic plasticity. They described a series of phenotypic transitions, starting with differentiated normal epithelia acquiring a WNT-signalling enriched state (LGR5+ ISCs). They show that the primary tumour program diverges from the normal ISC due to the enrichment of embryonic developmental genes and the presence of lineage promiscuity. Yet, this foetal-like phenotype also transitions to a developmentally primitive state that leads into non-intestinal transdifferentiation in advanced metastatic disease. Mechanistically, the authors identified *PROX1* as a transcriptional regulator driving tumour promotion and restricting non-canonical differentiation in advanced disease. Overall, the manuscript describes an extensive analysis of a unique cohort of patients using state-of-the-art technologies, suggesting a multi-step trajectory across different plasticity states during CRC progression. This is a potentially interesting paper providing new insight into the mechanism of CRC development and metastasis. However, there are some major concerns that need to be addressed to strengthen the conclusion.

We appreciate the reviewer's positive feedback and have addressed their specific comments below.

Specific comments:

Comment 2.1. Apart from the initial single-cell analysis, the rest of the results come from mostly only one of the patients. For instance, the only multiplex immunofluorescence images presented in the figures comes from one patient (KG146), although the authors state that they performed 72 samples. Similarly for organoids, it was mentions that 29 samples were obtained from the single cell suspension used for RNA-Seq, while only 3 PDOs were used for functional studies. It'd be important to include more replicates from various patients to strengthen their conclusion.

We value this feedback and have strengthened our analysis with extensive immunofluorescence quantification, validation experiments using multiple additional organoid lines, and transcriptomic validation in additional independent patient cohorts. Please see the **General Response** for a list of new experiments in the revision. Our new data and analyses provide further support for our key findings on cell-state progression, fetal state attributes, enhanced lineage plasticity in metastatic organoids and *PROX1* function; we find broad consistency both within and across multiple patients, in an independent sequencing modality, an expanded set of functional experiments, and independent scRNA-seq and bulk cohorts.

Comment 2.2. The data presentation is a bit confusing, as the referencing for the figures does not seem to follow a logical order. There are many instances of mistakes in figure references in the text and several typos throughout the manuscript.

We apologize and thank the reviewer for pointing out these oversights. We have closely proofread the revised manuscript and corrected figure calls and typos.

Comment 2.3. Throughout the study, the authors used their own samples, models and even signatures to generate the conclusions. While the single cell analysis appears to be comprehensive, the analysis is largely based on a few hundreds of cells per patient. Providing evidence that the proposed pattern can be observed in other dataset would greatly enhance the manuscript.

A major strength of our manuscript is that it assembles a sizable first-of-its-kind resource of synchronously resected trios of matched normal colon, primary tumor and metastasis from the same patients. Our dataset comprises more epithelial cells than most previous studies (**Table R2**); the next largest cohort that includes normal colon, primary tumor and unmatched metastases contains half as many samples (42 vs. our 83) and approximately one-ninth as many epithelial cells (5413 vs. our 47,437)³⁰. We specifically optimized our protocol to overcome the challenges of epithelial cell capture from CRC patient tumors, achieving unprecedented high-quality epithelial cell capture (571 per sample on average) (**Table R2**). This large and epithelial-enriched cohort is what allowed us to uniquely characterize epithelial subsets in a robust and unbiased manner, discovering novel non-canonical states across multiple patients and further characterizing them in four patients with up to thousands of tumor cells (KG146, 3397 cells; KG182, 935; KG183, 1216; KG150, 2574).

As stated above, we used the canonical and non-canonical modules from our initial submission as signatures to probe TCGA (445-patient)³ and LARC (108-patient)⁶ cohorts containing bulk RNA-seq data and clinical annotations—a common approach for generalizing conclusions from smaller scRNA-seq cohorts (e.g., see ref.¹²). In both of these pre-treatment primary tumor cohorts, our non-canonical gene signature is associated with tumor recurrence and poor overall survival, now further strengthened with Cox regression and survival analysis (new **Fig. 3h,i** and **Extended Data Figs. 6e,f**, new **6g–j** and **7g,h**).

In the revised manuscript, we also now validate our non-canonical signatures in an independent scRNA-seq dataset²⁹, which includes 5 matched primary and liver met CRC tumors with a relatively large number of cells per patient (but notably, 3-fold fewer transcripts per cell [8079 vs 23,335] and higher percentage of mitochondrial RNA, indicating cell stress). Using *score_genes* to assess expression of the same genes comprising our Hotspot modules, we found that 3 patients clearly express the neuroendocrine module and 4 others express squamous module (new **Extended Data Fig. 5g**, reproduced below as **Fig. R2a**). As in our cohort, 3 of these 5 patients exhibit more non-canonical states in the metastasis relative to the primary. To assess progression, we used a novel approach based on which gene modules overlap in the same cell, finding that the new cohort exhibits the same stepwise progression from canonical to non-canonical as in our data (new **Fig. 4f**, reproduced as **Fig. R2b**; see response to **Comment 2.14**).

In total, our non-canonical modules clearly appear in our scRNA-seq data (31 patients), recently published scRNA-seq data (5 patients), bulk RNA-seq data (553 patients from 2 cohorts), and imaging data from 25 patients (overlapping our scRNA-seq cohort). Moreover, we demonstrate that non-canonical modules appear at significantly greater frequency in metastasis based on our scRNA-seq cohort (new **Fig. 3f,g**), new quantitative analysis of our imaging data (new **Extended Data Fig. 5a–d**) and in the independent published cohort of 5 patients.

Figure R2 | An independent scRNA-seq cohort of primary and metastatic CRC exhibits progression to non-canonical differentiated states. **a**, Cumulative fraction of cells expressing >0.75 quantile score for a given module in 5 patients from Wang et al.²⁹. **b**, Fraction of cells expressing >0.75 quantile score for a given module in **d**) samples from the 4 patients with most non-canonical cells, **e**) all samples in our cohort, and **f**) samples from Wang et al.²⁹.

Comment 2.4. In line 101 page 4 of the manuscript, it is stated that the primary tumour cells are more closely clustered to normal epithelial cells while metastasis are divergent. In Figure 1e it appears that the primary cells overlap partly to the metastatic cells while both are separate to normal cells. Is there any measurement to support the statement in the text regarding the similarity between primary tumour and normal cells?

UMAPs do not preserve phenotypic distances, so we compute all distances from the kNN graph, as is customary in the field. To support our claim that primary tumor cells are phenotypically less diverse and closer to normal epithelial cells, we applied diffusion distance, a widely used phenotypic distance metric in single-cell data^{2,10,11,28}. We computed pairwise diffusion distances between every primary–normal cell pair and compared this with each metastasis–normal pair, demonstrating that primary cells are significantly closer to normal cell phenotypes than metastasis cells (**Fig. R3**). In a second analysis, we compared the diffusion distances between all cells across two primary samples to the diffusion distances between all cells across two metastasis samples, demonstrating that cells in primary samples are more similar to one another than cells in metastatic samples (**Fig. R3**). Our revision now includes substantial new data and results. Given that these phenotypic distances are a minor point that is not critical for the main claims of the manuscript, which is already laden with complex analysis, in the revision we remove the statement about these phenotypic distances from the manuscript.

Figure R3 | Primary tumor cells are phenotypically closer to normal and less diverse than metastasis cells. a) Average of pairwise diffusion distances computed between every primary–normal cell pair (left), and between every metastasis–normal cell pair (right). Distances were averaged for each primary or each metastasis sample, and normal cells from all samples were pooled for these comparisons. Multiscale space was calculated for tumor cells only, on the top 14 DCs chosen by the knee-point of the eigenvalues of the DCs, and computed using Palantir. **b)** Average of pairwise diffusion distances computed between every cell pair spanning two primary samples (left) or spanning two metastasis samples (right). **, $p < 2e-3$; ****, $p < 1e-80$, rank-sum test.

Comment 2.5. Could the authors elaborate on how Figure 1d and 1e support the inter and intra-patient tumour phenotypic heterogeneity? (Line 144, page 5)

We apologize, the figure callout was incorrect and meant to refer to **Fig. 2e** (comparing the few discrete phenotypic states with organized gene expression in healthy tissue to the disorganized and promiscuous phenotypic landscape of cancer cells) and an earlier figure that was removed before submission. We do find greater distinction between individual patient tumor samples compared to non-tumor, as quantified by lower patient entropy, indicating states that are unique to that patient (**Fig. R4**).

Figure R4 | Average patient entropy of each sample, by sample type. For each epithelial cell, patient entropy is calculated as the Shannon entropy of patient labels (using `scipy.stats.entropy`) within their 60-nearest neighboring cells based on multiscale diffusion distance. Multiscale space is calculated as in **Fig. R3**. Samples with greater patient entropy have phenotypes that are shared across more patients (i.e., are more similar). ***, $p < 1.6e-3$; ****, $p < 5e-5$, rank-sum test.

Our intent was not to make a statement about heterogeneity *per se*, but to motivate Hotspot analysis; as gene programs are less organized in tumor than in healthy tissue, it is important to use an unbiased method for finding structure in the data—cancer gene programs—that is shared across patients. We have now improved the text explaining Hotspot and provided extensive supportive analyses (please see response to **Comment 3.3** and new **Extended Data Fig. 4e–j**).

Comment 2.6. Figure 2b is referenced in the text in line 165 page 6, to highlight the normal intestinal programs. This is confusing as figure 2b is only a diagram of the cells in the colon crypts. The mentioned programs are shown in Figure 3a, and the extended data figures 4b, h. Additionally, the genes included in the text (FABP1, KRT20, TFF1 and LYZ) are not highlighted in any of these panels.

We apologize for the confusion; the colonic crypt schematic in **Fig. 2b** is meant to provide context to cell type annotations in **Fig. 2a**, not to inform how the ISC gene signature was developed. We have corrected the callout in the revision.

Comment 2.7. In line 119 page 4 the text mentions that genes associated to DNA synthesis and cell cycle progression are upregulated in normal ISCs. The genes mentioned in the text (RFC4, PRIM1, MCM2) are not shown in the extended data figure 3a, where the text refers to. The text continues to highlight the

WNT-signalling associated genes and the embryonic developmental genes, these are all contained in the extended data figure 3a except for CYP2W1. Although there is no reference to extended data figure 3 in the text.

We apologize and have fixed the figure calls in the revision. We also now include *CYP2W1* in **Extended Data Fig. 3a**.

Comment 2.8. Could the authors comment on how extended data figure 4i supports the statement that non-canonical modules occur repeatedly in metastatic cancers? Extended data figure 4i, does not seem to include information from the tumour site. Instead Figure 3f seems to contain that information.

We thank the reviewer for pointing out that **Extended Data Fig. 4i** was not the appropriate callout, as it does not distinguish between primary and metastatic tumors. We now provide visuals, quantification and analysis supporting that (1) neuroendocrine and squamous modules appear across multiple patients in our cohort, and (2) these are more abundant in metastasis. Specifically, we find the squamous Hotspot module in at least 5% of cells in 20 samples from 13 patients and the neuroendocrine Hotspot module in at least 5% of cells in 37 samples from 21 patients. These results are now visualized in a stacked barplot (new **Fig. 3e**). In addition, we directly compare the abundances of the non-canonical modules between each primary and metastasis sample in a matched pair, for all pairs in our cohort. Our log-ratio analysis (new **Fig. 3f,g**) indicates significantly more non-canonical expression in metastases compared to their primary counterpart for the majority of matched pairs ($p = 0.001$).

Figure 3b,f,g | Non-canonical modules are present in untreated patients and are enriched in metastases. b, Distribution of module labels in cells from combined primary and metastatic tumors from treated and untreated

patients; module labels are based on >0.75 quantile score in a given cell for the gene module. Vertical black line divides canonical intestinal cell types (right) from other cell types (left). **f,g**, Log-ratio of metastasis- to primary-derived tumor cells in each patient sample, labeled with a canonical (**f**) or non-canonical (**g**) cell type. Metastatic tumors (green circles) are enriched for cells expressing non-canonical modules, whereas primary tumors (red circles) are enriched for cells expressing canonical modules.

Comment 2.9. In lines 278-280, it says “We used dynamic time warping to align trajectories from each of the four patients whose tumors exhibit non-canonical differentiation to that of KG146 (possessing the broadest range of cell states)...”. KG146 is one of the four patients, not sure how the authors align the trajectory?

Indeed, we align three patients to the KG146 reference (rendering all four patients comparable), as originally described in the Methods section, *Alignment across patients by dynamic time warping*. While our dynamic time warping analysis is correct, it can only be applied to the 4 patients in our cohort with sufficient cell numbers, and it is based on pseudotime analysis, which makes strong assumptions about cell-state transitions. We have since devised a better approach to analyze the progression from canonical to non-canonical states (see response to **Comment 2.14** for more detail), which encompasses all patients and provides stronger evidence of the transition (new **Fig. 4d–f**). The dynamic time warping analysis created confusion, so we removed and replaced it with (unaligned) DC analyses of the three individual patients, as well as a patient with sufficient cell numbers in the independent dataset from Wang et al.²⁹, in a new **Extended Data Fig. 7a**. All five patients exhibit the same cell-state progression.

Comment 2.10. Why do the authors not include all the four patients described in the text in the panels in Figure 4g and extended data figure 6d?

The Methods section includes an explanation of why only 3 of 4 patients are used in **Fig. 4g**:

Palantir was run separately on the tumor datasets of patients KG146, KG182, and KG150, excluding KG183 because of the limited number of non-canonical cells in this patient.

We have changed “limited number” to “insufficient number of non-canonical cells” to clarify further. **Extended Data Fig. 6d** presented an analysis of progression in patients with metachronous metastases. We excluded patient KG150 from this panel because it does not have a metachronous metastasis.

Thank you for noting that this wasn’t clear; we now justify these exclusions in the figure legend. In the revision, we have removed **Extended Data Fig. 6d** given the small number of patients and our inability to draw clear conclusions from these data.

Comment 2.11. In extended data figure 8a, the expression of the top 6 foetal-associated transcription factors in the organoid systems while exposed to irinotecan was obtained as a relative expression from a scRNA-Seq experiment. The expression of these genes, especially due to their relevance to the conclusions to the manuscript should have been validated independently, e.g. by qPCR.

scRNA-seq is a well-established technique that allows gene expression to be reproducibly quantified in absolute terms. In contrast, qPCR quantifies expression in bulk cell populations relative to housekeeping genes whose expression can vary across cell states or in response to therapeutic stress. While qPCR was used to validate scRNA-seq when it was a nascent and unproven technology, this practice ended with the availability of highly accurate and reproducible commercial scRNA-seq kits in recent years. Thus, we respectfully disagree that qPCR should be used to validate scRNA-seq quantification. We now recompute associations based on diffusion components that have not been subject to dynamic time warping, which reduces the TFs to 5 significant factors (*LEF1*, *PROX1*, *SP5*, *ZNF385A*, *NR2F1*) (See response to **Comment 2.15**). Given that it is straightforward to perform qPCR for these five TFs in organoid lines exposed to irinotecan, we include these data below for the reviewer's interest (**Fig. R5**).

Figure R5 | qPCR confirms irinotecan-induced upregulation of 5 TFs identified in scRNA-seq analysis. Relative expression of 5 TFs correlated with fetal signature in organoids cultured in HISC with 250 nM irinotecan for 7 d. RT-qPCR is normalized to HISC-only condition.

Comment 2.12. The extended data figure 8 is not correctly referenced in the text.

We thank the reviewer for pointing out this and other typos and errors to figure calls, which have all been corrected in the revised manuscript.

Comment 2.13. In line 361 it is written OKG46Li organoids instead of OKG146Li.

We corrected this error in the revised manuscript.

Comment 2.14a. Could the authors elaborate what is the behaviour with the rest of the patients, besides from the 4 highlighted ones? Would the authors consider that all of them would show a similar pattern to patient 146, or is there any unique intrinsic factor making these patients able to reach this enhanced plasticity in their advanced disease.

We originally highlighted four patients because they harbored enough cells in non-canonical differentiated states to allow for robust trajectory analysis. Both total cells captured and non-canonical representation vary greatly among patients in our cohort (original **Fig. 3e** and new **Fig. 3b**), and in general, we note that advanced metastatic tumors are more divergent phenotypically than primary tumors^{5,14}. Injury induced by either metastasis or treatment appears to increase the prevalence of non-canonical states (new **Fig. 3f,g**), but injury only explains part of the observed heterogeneity, and we believe that we are unable to find factors that explain the remaining variability because we are underpowered to find a correlated factor in a cohort of this size.

The reviewer brings up an important question, however, and we do observe that most patient tumors include cells that express fetal and non-canonical programs (new **Fig. 3b**). This comment encouraged us to take a new approach that could apply to most patients and potentially provide more compelling support for a progression. Our approach is based on the observation that in most known trajectories, we observe a mixture of overlapping gene modules from pairs of sequential cell states along the trajectory. Taking the reverse logic, we reason that the existence of a substantial fraction of cells co-expressing two distinct gene modules suggests a pseudo-ordering of these states, and a transition between them. A key feature of Hotspot is that a cell can co-express multiple modules, as we demonstrated in **Extended Data Fig. 4f,g** (new **Extended Data Fig. 4k,l**). To ensure that we only consider robust module expression, we first assigned a cell to a module if it expressed module genes above the 75th percentile. Then, for all pairs of Hotspot modules, we computed the fraction of modules that are co-expressed in the same cell. Our new analysis reveals the same progression that we observe for the four patients in **Fig. 4**, but across all patients (new **Fig. 4d-f**). Furthermore, we uncover the same progression in independent data from Wang et al.²⁹ consisting of 5 patients with matched primary and metastatic CRC.

It thus appears that most CRC tumors may follow a similar cell-state progression towards non-canonical fates, and we only capture a fraction that are highly advanced. We have replaced our main figure panels (**Fig. 4d-f**) with our new, more compelling analysis. The factors that drive extreme non-canonical progression will be an important focus of future work.

Figure 4d–f | Shared module expression reveals a consistent progression from canonical to non-canonical fates among all patients in two independent cohorts. The fraction of cells expressing >0.75 quantile score for a given module in **d**) samples from the 4 patients with most non-canonical cells, **e**) all cohort samples except these 4 patients, and **f**) samples from the Wang et al.²⁹ cohort.

Comment 2.14b. Along this line, could the authors elaborate in the genetic background of the 4 patients that show the non-canonical behaviour?)

We included the tumor genotypes of all samples in our scRNA-seq cohort for which patient consent for mutational profiling was available (**Extended Data Fig. 1b**). Across the four patients whose tumors show the greatest non-canonical differentiation (KG146, 150, 182, 183), the only shared mutations are in *APC* and *TP53*. However, these are the two most commonly mutated genes across metastatic CRC, and are shared by most patients in our cohort (*TP53*, 93%; *APC*, 81%). Thus, we are unable to establish a genotype-to-phenotype relationship in our small cohort. Given the combinatorial genomic heterogeneity across patients, such correlations typically require very large patient cohorts (typically in the thousands), to achieve statistical power and account for multiple hypothesis testing (e.g. see ref.¹⁹).

Comment 2.15. In line 375, please provide the 6 shortlisted transcription factors.

In the initial submission, the 6 shortlisted transcription factor genes were enumerated in **Extended Data Fig. 8**. We now recompute associations based on diffusion components that have not been subject to dynamic time warping, which reduces the list to 5 significant factors (*LEF1*, *PROX1*, *SP5*, *ZNF385A*, *NR2F1*). These appear in the new **Extended Data Fig. 9a**, and we have also added them to the text (line 363).

Comment 2.16a. shPROX1 data organoids in Fig.6 and Extended Fig.8 are interesting. However, is there any functional relevance of the data?

Our organoid models allow us to inducibly perturb the *PROX1* transcription factor and thereby assess its function in gene regulation (by qPCR and now scRNA-seq) and canonical ISC state maintenance (by

organoid formation capacity). We have now expanded our original analysis of 4 lines from 2 patients to a total of 7 lines from 4 patients engineered with shPROX1 and shControl, and we confirm that *PROX1* knockdown upregulates non-canonical gene expression and reduces organoid-forming capacity in organoid lines harboring more canonical cell states. By contrast, *PROX1* knockdown has little effect on expression or ISC function in non-canonical tumors. Our results demonstrate that *PROX1* functions to restrict non-canonical differentiation and maintain the ISC state, but is less involved in regulating tumors that are more non-canonical in character. Please see the response to **Comments 1.6a** and **1.6b** for details of our original and expanded functional experiments.

Comment 2.16b. Do the PROX1 KD primary organoids show more metastatic potential in vivo? In other words, does PROX1-mediated non-canonical differentiation promote metastasis?

As suggested, we performed orthotopic intrahepatic transplantation of two canonical primary tumor-derived organoid lines (136P, 146P), pre-treated with dox to induce *PROX1* knockdown. Over 12 weeks, we observed no outgrowth of liver metastasis in either shControl or shPROX1 lines (new **Extended Data Fig. 9f,g**). While this negative result could have many explanations, including insufficient time to observe a phenotype (metastatic colonization in patients occurs over years), these *in vivo* data, together with our extensive cell-state and organoid-line dependent functional and transcriptomic studies of *PROX1* knockdown, suggest that *PROX1* inhibition is insufficient to induce a metastatic phenotype. Such a transition likely requires the activity of multiple TF genes, possibly including *LEF1*, *SP5* or other factors that are coexpressed with *PROX1* (**Extended Data Fig. 8a**). Unveiling additional regulators of non-canonical cell states will be the subject of future work, but is beyond the scope of the current study.

Extended Data Figure 9g,h | Organoid xenograft experiments with *PROX1* knockdown. **g,h**, Normalized average radiance measured by weekly ex vivo bioluminescence imaging following intrahepatic injection of 500,000 organoids from OKG146P (**g**) or OKG136P (**h**) lines bearing shCtrl, shPROX1-1 and shPROX1-2 ($n = 9, 10$ and 10, respectively, for each organoid line from each patient) in NSG mice, normalized to signal immediately following injection (week 0). Error bars, s.e.m. ** $p = 0.008$, ***, $p = 0.001$, Mann Whitney rank-sum test.

Reviewer #3

This study leverages single cell RNA-sequencing of matched sample trios (normal, primary tumor, metastasis) of treated and untreated CRC patients to describe the plasticity of cancer cell states along an axis of canonical (intestine related) to non-canonical (intestine unrelated) states. The latter states are enriched in metastasis, involve downregulation of classical programs including the LGR5+ intestinal stem cell-like program, are associated with poor clinical outcome and induced by cell-intrinsic (PROX1 loss) as well as extrinsic factors (i.e. therapy). Trajectory analysis suggest that canonical cancer cell states first transition into an ISC-like state and then de-differentiate into an intermediate highly plastic fetal progenitor-like state. This state could serve as a bridge to more differentiated non-canonical states (neuroendocrine and squamous). Lastly, organoid PDX models derived from metastatic tissue containing non-canonical programs can grow in metastatic locations (liver) in contrast to organoids derived from primary locations (lacking non-classical programs).

This is an important and highly interesting study that reflects a major advance in understanding CRC metastasis and the associated cell plasticity. It provides a roadmap for comparison of primary vs. metastasis samples and convincingly shows how important it is to do this in a human setting in comparison to mouse models (CRC liver metastasis in mouse models are largely LGR5+). It is based on profiling of clinical samples that are especially hard to collect (synchronous and metachronous metastasis together with primary and normal), therefore representing a valuable resource. The analysis is rigorous and sound, the manuscript is well written and the figures are clear and intuitive to follow. Overall, I support the publication of this work and expect it to have a significant impact. I have quite a few comments but most are minor and reflect suggestions for the authors discretion.

We thank the reviewer for appreciating the importance of our study and for the critical feedback to improve the rigor of our conclusions.

Major comments:

Comment 3.1. Much of the analysis is based on 4 patients and some of the observations are derived from just one patient, or from experiments with organoids derived from one patient. Yet, the discussion highlights consistency among patients and argues that the transition from canonical to non-canonical is a recurring feature of most patients. Therefore, it would be good to soften this claim and to further evaluate the degree of consistency across all patients and not only the 4 that are highlighted in the analysis.

Trajectory analysis requires a good number and representation of cells across cell states in a single patient to provide robust results, which is why we restricted this analysis to the 4 patients presented in **Fig. 4**. However, we note that the majority of patient tumors harbor some fetal and non-canonical cell fates (new **Fig. 3b**), which are enriched in metastases (new **Fig. 3f,g**) (expounded in detail in response to **Comment 3.23** below). This is confirmed by Vectra multiplexed immunofluorescence (see response to **Comment 3.2**). We have now devised an alternative to DC-based trajectory analysis that is more appropriate for patients with fewer cells in non-canonical states. Our approach is based on the observation that in most known trajectories, we observe a mixture of overlapping gene modules from pairs of sequential cell states

along the trajectory. Taking the reverse logic, we reason that the existence of a substantial fraction of cells co-expressing two distinct gene modules suggests a pseudo-ordering of these states, and a transition between them. A key feature of Hotspot is that a cell can co-express multiple modules, as we demonstrated in the original **Extended Data Fig. 4f,g** (now **Extended Data Fig. 4k,l**). To ensure that we only consider robust module expression, we first assigned a cell to a module if it expressed module genes above the 75th percentile. Then, for all pairs of Hotspot modules, we computed the fraction of modules that are co-expressed in the same cell. Our new analysis reveals the same progression that we observe for the four patients in Fig. 4, but across all patients (new Fig. 4d–f). Critically, we uncover the same progression in independent data from Wang et al.²⁹ consisting of 5 patients with matched primary and metastatic CRC. It thus appears that most CRC tumors may follow a similar cell-state progression towards non-canonical fates.

Figure 4d–f | Shared module expression reveals a consistent progression from canonical to non-canonical fates among all patients in two independent cohorts. d–f, The fraction of cells expressing >0.75 quantile score for a given module in **d)** samples from the 4 patients with most non-canonical cells, **e)** all cohort samples except for these 4 patients, and **f)** samples from the Wang et al.²⁹ cohort.

Comment 3.2. An impressive cohort of multiplex images with relevant markers was created (n=73 samples) but not quantitatively used to back up the claims found by scRNA-seq (see below). Is it possible to utilize that to support at least some of the claims?

We appreciate this feedback, given by all three reviewers. We have now quantified 1194 fields of view (FOVs) generated using the Vectra technology from 74 tissue sections and 24 patients, comprising 6,090,968 segmented cells for Panel 1 and 5,213,051 segmented cells for Panel 2, and we include additional representative images from more patients. This quantification involved extensive supervised analysis, including using Mesmer⁸ for segmentation, model and data normalization to enable cross-FOV comparison (see Methods). To help elucidate the impact of metastasis and therapy, we compared primary versus metastatic tissues separately in the treated and untreated samples.

Our analysis shows that, in accordance with the scRNA-seq results, metastases display lower intestinal stem (OLFM4) and differentiated (CK20) marker expression, and higher injury repair (TROP2) marker expression (**Extended Data Fig. 5a–d**). These trends hold true when comparing primary and metastasis in

the treated and untreated cohorts separately. In treated tumors, we observe lower intestinal stem (OLFM4) and differentiation (CK20) marker expression, and higher injury repair (TROP2) and squamous (CK5) marker expression than in untreated tumors. Although there are too few squamous cells in untreated samples to compare primary to metastatic tissue, and we found it impossible to robustly quantify neuroendocrine marker (CHGB) due to high and variable background, we observe that cells annotated for either marker are enriched in metastatic samples over primary samples in scRNA-seq data: of all cells with >0.75 quantile expression for i) *CHGB*, 139 are from primary and 455 from metastasis, and for ii) *CK5*, 8 are from primary and 556 are from metastasis.

Extended Data Figure 5a–d | Multiplexed immunofluorescence confirms the shift away from canonical intestinal marker expression in metastases and therapy-treated samples. Fraction of tumor cells per FOV with high expression (‘high expression’ for each marker based on a minimal expression threshold determined by the knee-point decile value from all marker-positive cells) of a) CK20 (differentiation), b) OLFM4 (intestinal stem cell), c) TROP2 (injury repair) and d) CK5 (squamous). Orange, primary tumors; purple, metastases. *, $p < 0.05$; **, $p < 0.01$; ***, $p < 0.001$, rank-sum test.

Minor comments/suggestions:

Comment 3.3. Given the complexity of Hotspot and the fact that this is not a widely used or easily understood method, it would be good to demonstrate that the main results can be reproduced with an alternative approach.

Hotspot is a powerful analytical approach that was recently developed to find structure in heterogeneous datasets and that accounts for gene pleiotropy and gene expression variability across cell subpopulations. It uses local autocorrelation to identify modules of co-expressed genes that are conserved across subpopulations within patients and that the standard practice of utilizing global gene correlation often fails to recover. We have improved the description of Hotspot in the corresponding Results section:

To study tumor progression in a more fully representative patient population, we searched for trends representing the entire diverse cohort of 31 patients, most of whom received therapy prior to resection (Fig. 1b,c). Contrary to the strong within-cell-type gene correlation structure that characterizes healthy intestinal epithelium, we found that cancer cells display substantial disorder and dysregulation (Fig. 2e), hindering standard annotation approaches and motivating an unsupervised

search for modules of covarying genes (gene programs). Our approach assumes that covarying gene expression reflects the coordinated gene regulation needed for biological function, and recognizes that expression in cancer is highly context-dependent—varying by patient, local environment and other factors. Specifically, we searched for modules of genes that only covary within cell subsets representing salient cell states in our data using the Hotspot algorithm. Unlike global measures such as Pearson correlation, which presume relationships between features are consistent across the entire dataset, Hotspot finds modules of genes with significant autocorrelation within local cellular neighborhoods of the phenotypic manifold. These modules are shared across patients and accommodate gene pleiotropy in cancer.

Hotspot identified 37 gene programs across epithelial cells from all tumors, which we manually curated and annotated (**Supplementary Table 3** and Methods). We grouped modules based on their similarity and biological coherence to reduce redundancy, highlighting ten modules (**Fig. 3a**, **Extended Data Fig. 4a,b**, **Supplementary Table 3** and Methods) that are shared across multiple patients (**Fig. 3b**). The local autocorrelation and detected modules are highly robust to varying parameter values and down-sampling of the data, indicating that they represent strong and reliable signal (**Extended Data Fig. 4e–g**). Importantly, the local covariation employed by Hotspot is critical to identifying these modules (**Extended Data Fig. 4h**). However, while these modules are not the dominant global correlations in the data and would therefore be missed by standard analysis, they are consistent with global correlation (**Extended Data Fig. 4i,j**).

The Hotspot approach is unique and cannot be replaced by other methods; however, it is important to note that the identified modules are largely confirmed by standard analyses. Many of the modules, including canonical intestinal, EMT and injury repair, represent important known gene programs with expected activity. We see evidence of the non-canonical modules at the protein level by multiplex IF. Most importantly, in the revision, we show the replication of these same modules in an independent dataset²⁹.

The revision also includes two new sets of analyses that provide further support for our Hotspot modules. First, we demonstrate that the local autocorrelation and resulting modules are very robust to random data downsampling (new **Extended Data Fig. 4g**). Second, we demonstrate that Hotspot robustly detects modules for which global correlation does show (weak) signal, indicating that Hotspot improves signal detection rather than finding spurious modules (new **Extended Data Fig. 4i,j**). While such weak global correlation across all cells would not stand up in primary analysis, the fact that Hotspot matches detectable Pearson correlation trends provides additional support for these modules.

Extended Data Figure 4g,i,j | Hotspot is robust to downsampling and enhances the detection of locally correlated genes. **g**, Maximum difference in the local correlation between genes, normalized by the maximum local correlation value, for 10 progressively large random downsamples of our tumor scRNA-seq dataset. **i**, Comparison of Pearson r global correlation and Hotspot z local correlation values for all Hotspot module genes. **j**, Average Hotspot z and Pearson r correlations between all gene pairs; each pair consists of one gene from each indicated Hotspot module.

Comment 3.4. PC1 is initially described as being tightly linked to the ISC signature and then later it is described as a measure of canonical vs. Non-canonical modules, e.g. in Fig. 3e-f. Both of these claims do not seem to be demonstrated directly and it is confusing that the same PC is interpreted quite differently in the different sections. In the first section this is less of an issue because the ISC signature is also shown directly, but in the later section there does not seem to be a direct analysis of canonical vs. non-canonical modules. It seems to me that adding a more direct measure - such as those in ED fig. 4g - would be helpful, and could fully replace the use of PC1 in fig. 3.

The two PCs in question are derived from very different subsets of the clinical data. In **Fig. 2**, we only included data from the 9 untreated patients; given that these patients harbor fewer non-canonical cells (**Fig. 3b**), their strongest axis of variation is along the ISC signature. In **Fig. 3**, we analyzed the full cohort of 31 treated and untreated patients; thus, its PC is quite distinct. We now further highlight this distinction in the main text:

The untreated patient trios provide a unique opportunity to characterize tumor progression to metastasis without the confounding influence of therapy. We therefore partitioned the data and restricted our initial analysis to the nine untreated patients, comprising 13,935 cells (**Fig. 2a**).

...

To study tumor progression in a more fully representative patient population, we searched for trends representing the entire diverse cohort of 31 patients, most of whom received therapy prior to resection (**Fig. 1b,c**).

In the original submission, we switched from the PC-based linear axis of greatest variation, to the DC non-linear axis of greatest variation, as both axes converge on the canonical to non-canonical axis and DCs are better suited to scRNA-seq and more widely used in that context. We note that both the top PC and the top DC agree, underscoring the strength of this signal. Evidence that DC1 indeed represents the canonical to non-canonical axis (for 4 representative patients with sufficient numbers of non-canonical states) is shown in **Fig. 4b** and **Extended Data Fig. 7a**. We have removed the PC analysis in **Fig. 3** in favor of representing module composition by cumulative fraction (new **Fig. 3b**) and log-ratio analysis of non-canonical to canonical fractions (new **Fig. 3f,g**), which should reduce confusion.

Comment 3.5. Fig. 3C-D and Fig. 5E-F:

- a. Not clear which channels are turned on and off in first vs. second row of panel.
- b. Higher magnification insets should be provided to better judge staining quality.
- c. It is unclear how large these sections are. Please provide an overview of the sample areas in mm².

We now clearly label all channels in these figures and provide higher magnification insets in **Fig. 5e,f**. To maximize capture of all areas of the tumor including the invasion front, full clinical pathology sections prepared for clinical diagnosis were used for imaging (not core punches as typically used for TMAs). For Vectra 7-color imaging, from each tissue section, we captured roughly 9 fields of view on average, each 1.34 mm² in size. For COMET 12-color imaging, whole sections were used, with a single field of view 82.5 mm² in size.

Comment 3.6. Fig 2g: it would be better to quantify the exact fraction of cancer cells expressing OLFM4 in primary vs metastasis.

We now quantify the fraction of cancer cells expressing OLFM4 (**Extended Data Fig. 3f**).

Extended Data Figure 3f | Multiplexed immunofluorescence confirms higher ISC marker expression with treatment and metastasis in treated tumors. Fraction of tumor cells per FOV with high expression ('high expression' for each marker based on a minimal expression threshold determined by the knee-point decile value from all marker-positive cells) of OLFM4 (intestinal stem cell marker). *, $p < 0.05$; **, $p < 0.01$; ***, $p < 0.001$, rank-sum test.

Comment 3.7. Extended Data Fig. 5c: This heatmap does not seem to support the claim that non-canonical programs are correlated with OS from surgery or Diagnosis.

We agree with the reviewer that the heatmap is not very clear. To increase the rigor of the analysis, we have now edited the heatmaps to only include descriptive binary variables reflecting tumor or patient characteristics at the time of surgery. For survival correlations, we have now performed multivariable regression analysis while correcting for confounders, as described in the response to **Comment 3.8** (new **Extended Data Fig. 6g-j**). We also show Kaplan-Meier curves for DFS in relation to primary tumor expression of module signatures (new **Extended Data Fig. 6f**). We have amended the claims in the manuscript accordingly.

Comment 3.8. Extended Data Figure 5G: It would be better to control for other prognostic factors that might be associated with the programs and therefore confound the results (age, stage, performance status). This could be implemented using a multivariable regression model, such as Cox proportional hazards regression.

We thank the reviewer for this suggestion, and have now performed multivariable regression to delineate whether any of the gene expression modules are associated with outcomes after controlling for confounders. We ran a Cox proportional hazards model for disease-free survival and overall survival including the signature values, age, stage and residual tumor as covariates. The absorptive intestinal and endoderm development signatures are significantly associated with disease-free survival and overall survival in this analysis (new **Extended Data Fig. 6g-j**).

g

h

i

j

Extended Data Figure 6g-j | Clinical features and outcomes associated with canonical and non-canonical module expression. Multivariate analysis of associations between module expression and (g, h) disease-free survival (DFS) or (i, j) overall survival (OS). Hazard ratios and p values were calculated using a Cox proportional hazards model that included all of the clinical and genomic variables shown in the panel. The numbers in parentheses and the length of the error bars show the 95% confidence interval for each hazard ratio.

Comment 3.9a. Line 262 and Figure 4b: It would be helpful to show this plot (or a variant of it) for all 31 patients.

Trajectory analysis and dynamic time warping require a good number and representation of cells across cell states to provide robust results, which is why we restricted this analysis to the 4 patients presented

in **Fig. 4**. We have now devised an alternative approach to inferring cell-state progression (please see response to **Comment 3.1** and new **Fig. 4d–f**), which confirms that CRC tumors across our entire cohort follow a similar cell-state progression towards non-canonical fates, as do tumors in the independent Wang et al.²⁹ cohort.

Comment 3.9b. Similarly, Line 309-311: The authors generalize that the fetal progenitor state is an intermediate that bridges to more differentiated cells. However, this is only shown for n=1 patient (KG146). Also: How does the fetal progenitor figure behave if plotted as in extended figure 6A + B?

To answer the second question, we plot the fetal progenitor signature as a function of (normalized) progression along DC1 (**Fig. R6**), but point out that these are subplots of the traces aligned by dynamic time warping that were presented in the original manuscript (**Fig. 6a**). Notably, we see that whereas KG146 peaks and declines, and KG182 peaks (at 0.848, representing 91.5% of its progression along DC1) and begins to decline, KG150 and KG183 have not advanced enough to reach peak fetal signature expression.

Figure R6 | Fetal signature expression along cells of tumors along DC1, aligned by dynamic time warping.

To answer the first question, our new analysis of overlapping modules (see response to **Comment 3.9a**) finds that the endoderm state, which serves as a good proxy for the fetal state among the Hotspot modules, is intermediate between canonical and non-canonical states. Specifically, the endoderm module is co-expressed with the canonical ISC-like module, as well as non-canonical squamous and neuroendocrine modules. No canonical module overlaps with the non-canonical modules, providing additional support for this ordering in all patients (not just the four patients in **Fig. R6**).

Given the general biological observation that aggressive dedifferentiation potentiates subsequent differentiation toward diverse fates, we do not consider our observation of sequential progression to be particularly contentious. Although we do not observe examples of non-canonical differentiation preceding the fetal state, we avoid making the stronger claim that the fetal state is strictly required for non-canonical differentiation.

Comment 3.10. Line 291-293: This claim seems highly speculative. I suggest to systematically compare the abundance of canonical and non-canonical programs of cells from all metastasis locations (i. synchronous liver vs peritoneum, ii. metachronous liver vs lung vs chest wall). Does this result in the same findings as shown in extended data figure 6d?

We agree that it is speculative to extrapolate from these few observations. Since we only collected a handful of metachronous metastases, we do not have adequate statistical power to make claims about secondary metastasis, and we have removed **Extended Data Fig. 6c** and related text accordingly.

Comment 3.11. Figure 5d: This finding is very interesting and important. However, again $n=1$. What about the organoids models representing the other three patients that show high state diversity (KG182, KG150 and KG183)?

We are delighted that the reviewer also finds this result exciting. Unfortunately, due to inadequate quantities of surgical tissue, we were only able to generate organoids from the metastases of KG182 and KG183, and lack matched primary tumor organoids for comparison. However, we were able to use OKG136P/136Li to repeat this finding in a second pair of organoid lines. Although cells from patient KG136 are largely canonical, scRNA-seq analysis showed that the liver metastasis (KG136Li) contains more non-canonical cells (**Fig. 3f**), including neuroendocrine module cells, which are not present in the primary tumor cells (**Figs. 3e and R7**). We engineered OKG136P and OKG136Li organoids with tdTomato-AkaLuc and performed intrahepatic injection. While in OKG146Li we found 100% penetrance of liver metastasis (4/4 xenografts), in the more canonical OKG136Li line, 3/10 xenografts grew out to establish liver metastases, while similar to OKG146P, none of the OKG136P injected animals established liver metastases (new **Extended Data Fig. 8g,h**).

Extended Data Figure 8g,h | Organoid xenograft experiments. Normalized average radiance measured by weekly ex vivo bioluminescence imaging following intrahepatic injection of 500,000 organoids in NSG mice, normalized to signal immediately following injection (week 0), for (g) OKG146P (primary tumor) and OKG146Li (metastasis) lines ($n = 7$ and 4, respectively; $p = 0.0143$, Mann Whitney rank-sum test at Week 12; $p = 1.458e-13$, normalized radiance > 10 , Fisher's exact test, all weeks), and (h) OKG136P (primary tumor) and OKG136Li (metastasis) lines ($n = 10$ and 10; $p = 0.39$, Mann Whitney rank-sum test at Week 12; $p = 2.35e-6$, normalized radiance > 10 , Fisher's exact test, all weeks). Error bars, s.e.m.

Figure R7 | Distribution of module labels in cells from primary and metastatic tumors from patient KG136; module labels are based on >0.75 quantile score in a given cell for the gene module. Vertical black line divides canonical intestinal cell types (right) from other cell types (left).

Comment 3.12. Line 356-358: “canonical and non-canonical markers appears to be mutually exclusive”. Can this be quantified using image analysis?

This claim is best quantified using scRNA-seq data (**Fig. 3a**), in which average local correlation of genes from nearly all pairs of canonical and non-canonical modules is negative; the tumor ISC-like and endoderm development modules, which are intermediate between canonical and non-canonical states (**Fig. 4d-f**), are the only exception (**Extended Data Fig. 4c**). Similarly, we labeled all cells based on >0.75 quantile score for differentiated canonical (absorptive intestine, secretory intestine, or both) and non-canonical (neuroendocrine, squamous or both) modules and retrieve a low Jaccard score of 0.102 (Jaccard score measures the ratio of marker pairs that are co-expressed in a cell to all cells that express at least one marker in the pair). This mutual exclusivity of canonical and non-canonical programs at the cell level is best seen in **Extended Data Fig 4l**.

As requested by the reviewer, we also quantify mutual exclusivity at the protein level in our Vectra multiplex IF images (**Table R2**). We note that the imaging-based analysis is less comprehensive, given the small number of marker genes and the high levels of autofluorescence for neuroendocrine markers that precluded their rigorous quantification.

Vectra panel	Gene A	Gene B	Jaccard Score
2	OLFM4	CK5	0.0443
2	OLFM4	TROP2	0.0947
2	CK5	TROP2	0.0327
1	CK20	CDX2	0.2891

Table R2 | Vectra multiplex immunofluorescence reveals lack of overlap between canonical and non-canonical markers. Jaccard score measures the ratio of marker pairs that are co-expressed in a cell to all cells that express at least one of the markers. Two canonical markers, CK20 and CDX2, are included to show the Jaccard score for an expected marker overlap. Only the subsets of cells with 'high' marker expression were used in these analyses.

Comment 3.13. Fig 6C: This shows CDX2^{high} to CDX2^{low}PROX^{high} transition. However, CK5^{high} cells seem to be intermingled with PROX1^{high} cells without obvious spatial segregation. Therefore, this representative picture does not support the claim of transitioning from CDX2^{high} to CDX2^{low}PROX1^{high} to PROX1^{low}CK5^{high} cells.

While this spatial segregation was suggested by our preliminary image quantification, upon more rigorous image segmentation and quantification during revision, we found that the staining quality of CK5 was suboptimal. We will therefore remove this claim, as suggested by the reviewer. We do consistently observe a transition from CDX2^{high} to CDX2^{low}PROX1^{high} states, which we show with additional representative images.

b

Figure 6b | Relationship between PROX1 expression and the transition to poorly differentiated states. Lunaphore COMET immunofluorescence imaging of CDX2 (intestinal), PROX1 (fetal) and CK5 (squamous) of the invasion front of two primary tumors (top) with corresponding hematoxylin and eosin imaging (middle), showing the relationship between PROX1 expression and the transition from well-differentiated glandular morphology to poorly differentiated state. Arrow indicates a blood capillary completely encased by mucosa invasive cancer cells. Bottom panels show imaging of matched synchronous liver metastases.

Comment 3.14. In line 85 + 86 not only the number of samples but also from how many patients they are derived from should be stated.

We clarify this in the revised manuscript:

For each tissue site, we performed scRNA-seq (83 samples from 31 patients), derived organoids from the single-cell suspension used for scRNA-seq (29 samples from 15 patients) and carried out multiplex immunofluorescence when tissue was available (72 samples from 21 patients).

Comment 3.15. Lines 100-101: metastasis cells are more diverged - this seems to be driven by a small subset of tumors rather than a global tendency of all metastases so it might be better to rephrase this sentence.

UMAPs do not preserve phenotypic distances, so we compute all distances from the kNN graph, as is customary in the field. To support our claim that primary tumor cells are phenotypically less diverse and closer to normal epithelial cells, we applied diffusion distance, a widely used phenotypic distance metric in single-cell data^{2,10,11,28}. We computed pairwise diffusion distances between every primary–normal cell pair and compared this with each metastasis–normal pair, demonstrating that primary cells are significantly closer to normal cell phenotypes than metastasis cells (**Fig. R2**). In a second analysis, we compared the diffusion distances between all cells across two primary samples to the diffusion distances between all cells across two metastasis samples, demonstrating that cells in primary samples are more similar to one another than cells in metastatic samples (**Fig. R2**). Our revision now includes substantial new data and results. Given that these phenotypic distances are a minor point that is not critical for the main claims of the manuscript, which is already laden with complex analysis, in the revision we remove the statement about these phenotypic distances from the manuscript.

Figure R2 | Primary tumor cells are phenotypically closer to normal and less diverse than metastasis cells. **a**, Average of pairwise diffusion distances computed between every primary–normal cell pair (left), and between every metastasis–normal cell pair (right). Distances were averaged for each primary or each metastasis sample, and normal cells from all samples were pooled for these comparisons. **b**, Average of pairwise diffusion distances computed between every cell pair spanning two primary samples (left) or spanning two metastasis samples (right). **, $p < 2e-3$; *****, $p < 1e-80$, rank-sum test.

Comment 3.16. Extended Data Fig 3a: Color code and column clusters unclear

We have improved text clarity and added legends for the color code in the revised figure.

Comment 3.17. Line 104: "enrichment" implies higher epithelial fraction than in the real tissue, which is not shown, so perhaps better to replace with "abundance" or something similar

Our protocol does enrich the epithelial cell fraction relative to existing protocols for single-cell isolation (see **Table R2** at the end of this document), but we have changed "enrichment" to "robust capture" to reflect that epithelial cells are relatively abundant in our data.

Comment 3.18. Line 139: it is not clear from the data that selection is involved in the lower ISC of metastatic cells - this could also reflect a dynamic response of metastatic cells to their environment. Similarly in lines 236-237 and lines 437-439 – unclear if selection should be invoked.

Indeed, we cannot deconvolve the relative contributions of selection and dynamic adaptation of cell states during the metastatic process, and both processes likely contribute. Our new text:

(Original Line 139) Together, our patient data suggest that **untreated** CRC tumors are enriched in ISC-like programs with primitive developmental and mixed lineage features, while ~~metastatic progression appears to select against~~ ISC programs **are depleted in metastatic tumors**.

(Original Line 236) ~~Together,~~ Our analyses **collectively** suggest that **subpopulations expressing** non-canonical modules can exist ~~in subpopulations~~ within untreated primary tumors and undergo ~~positive selection~~ **enrichment** during metastasis, and that ~~it is~~ **they are** associated with negative clinical outcomes ~~including poor overall survival~~.

We do allow ourselves some speculation about possible enrichment mechanisms in the Discussion (e.g. 'enabling or selecting for non-canonical differentiation').

Comment 3.19. Extended Data Figure 3b: Almost all of the depicted genes have a $\log FC < 1$. Some of them (LGR5) are almost 0. With such low FC, it is difficult to ascribe biological significance to those changes.

Also, it would be interesting to see how this compares to the treated tumors. What are the pathways up- and downregulated after chemotherapy stress and how do these compare to the untreated pathways?

We thank the reviewer for noticing this issue. The x-axis was mislabeled and was actually plotting a fitted MAST coefficient that is an output of the algorithm and maintains relative fold-change differences between genes, but is difficult to interpret in absolute terms. We now plot log fold-changes from the original data, showing that most values are below a log(FC) of 5, corresponding to a 10,000-fold difference. Log(FC) of 0.3 corresponds to a 2-fold difference.

Regarding pathway changes after chemotherapy, we agree that this is a very interesting question, but an unbiased analysis should utilize scRNA-seq data from patient-matched pre- and post-treatment samples. Patients in our cohort either received or did not receive chemotherapy prior to surgical resection; thus, comparisons can only be made across (not within) patients, confounding the interpretation of results.

Comment 3.20. Extended Data Figure 3b: Fig 3a: What genes/modules are behind the non-annotated clusters?

We kindly point the reviewer to the *Hotspot module grouping and annotation* section in the Methods, which already includes this information.

Comment 3.21. Extended Fig 4i: It is not clear what "well-mixed set of patients" (= high entropy, line 659) means concretely. I.e. the secretory intestine program has the highest entropy: What's the fraction of patients that have it? Overall I believe a better way to show this might be with a heatmap that shows per patient the presence and abundance of each of the 10 programs. Sorted by the most redundant programs to the patient specific ones (i.e. from secretory intestine to absorptive intestine.)

We apologize for the lack of clarity in our original submission. The entropy of cell proportions across each patient in a given subset of cells (defined by a kNN neighborhood or gene module) is frequently used to quantify the ubiquity of a trait across patients. The term 'patient mixing' was coined when using entropy to measure the removal of batch effects during data integration; i.e., when each neighborhood in the kNN graph contains similar cells from many different patients (patient labels are well-mixed), it has a high entropy based on the proportion of each patient in the neighborhood. It is perhaps not the best term to indicate how well each Hotspot module is spread across the patient cohort. As described in the Methods section "*Interpatient entropy of gene modules*", we generated distributions of modules across patients by repeatedly subsampling to generate identical cell numbers from each patient, then calculating the entropy of a given module (based on defining 'high-scoring cells' as those with a module score 1 s.d. above the mean). Distributions close to zero entropy correspond to highly patient-specific modules, whereas distributions close to 1 exist across many tumors. We now use clearer language in the legend of new **Extended Data Fig. 4d**:

High entropy indicates that cells with high score for a module (> 1 s.d. above mean, calculated across all tumor cells) come from a diverse set of patient samples.

Comment 3.22. Fig. 3E: Not clear if this refers to cancer cells from primary only or combined with metastasis.

The original Fig. 3e showed cancer cells from both primary and metastatic lesions for each patient, but this visualization has been removed in the revision.

Comment 3.23. Lines 219-220: this is an important claim, but it is not clear to me how it is demonstrated in the cited figures. In Fig. 3e i don't see a clear association and if there is it should be explained. In Fig. 3f i do see the association but only in a subset of patients and it would be good to spell out more clearly the fraction of patients with such effect. Also ED fig. 5 does not seem to clearly support the claims as the associations vary between modules and do not highlight a clear pattern as seems to be implied in the text.

We agree that Fig. 3e,f did not effectively convey patient heterogeneity and non-canonical progression in metastasis, either visually or quantitatively, and thank the reviewer for these suggestions. We now compute the relative abundance of canonical and non-canonical modules within each matched primary and metastasis pair (new Fig. 3b), clearly showing that metastatic samples include more cells expressing non-canonical modules than their synchronously resected primary tumor (new Fig. 3f,g).

Figure 3b,f,g | Non-canonical modules are present in untreated patients and are enriched in metastases. b, Distribution of module labels in cells from combined primary and metastatic tumors from treated and untreated patients; module labels are based on >0.75 quantile score in a given cell for the gene module. Vertical black line

divides canonical intestinal cell types (right) from other cell types (left). **f,g**, Log-ratio of metastasis- to primary-derived tumor module proportions in each patient sample, based on accumulation of non-canonical (**f**) or canonical (**g**) modules. Metastatic tumors are significantly enriched for cells expressing non-canonical modules ($p = 0.001$, rank sum test; Methods).

Regarding **Extended Data Fig. 5**, we agree — please see our responses to **Comments 3.7** and **3.8**.

Comment 3.24. Fig. 3F: Unclear why these samples were chosen. This plot should be shown in the supplement for every pair. Also, it is unclear to me what defines the separation of patients into three groups, from top to bottom.

We have now replaced **Fig. 3f** with a log-ratio quantification of abundance of canonical and non-canonical cells between each matched pair for the entire cohort. Please see response to **Comment 3.23**.

Comment 3.25. Lines 256-260: it is unclear from fig. 4b (and in general) how high is the consistency of the fetal signature with the changes that occur in the tumor along DC1. The average profile shown at the bottom of this panel implies that there is some consistency but it could still be reflecting a minority of the genes in the signature and accordingly might not fully justify the claims. It would be good to clarify what fraction of genes from the fetal signature are individually consistent with this pattern of highest value in the intermediate state defined by DC1.

We find that a core set of 14 fetal signature genes are significantly associated with the full fetal signature in all four patients that harbor a substantial number of cells in differentiated non-canonical states (Pearson $r > 0.5$ for gene expression and fetal signature trends over DC 1), and 60 of 113 are significantly associated in at least 2 patients (**Extended Data Fig. 7d**).

Extended Data Figure 7d | Fetal signature gene representation across multiple patients. Overlap of fetal signature genes with Pearson correlation > 0.5 across four patient samples harboring a substantial number of cells in non-canonical differentiated states.

Comment 3.26. Lines 272-285: an overall similarity between the trajectories of different tumors is easily apparent from fig. 4. Yet there are some differences that likely reflect real biology. The attempt to align all of the distinct trajectories using a time-warping analysis seems unnecessary to me and the argument for a single stereotypical pattern (as opposed to a consistent overall trend but with distinct tumor-specific variants) seems somewhat exaggerated to me. I would suggest to shorten this part and soften the claim.

We now replace the dynamic time warping analysis across these 4 patients with a more general analysis, based on overlapping modules that shows a shared trend of stepwise progression, across our entire cohort (please see response to **Comment 3.9a**). All individual patients are now re-analyzed and plotted without any alignment or time-warping.

Comment 3.27. Extended Fig 5C-D: The Osteoblast program seems to behave more in line with the canonical than the non-canonical programs. Can the authors comment on that?

We agree that the osteoblast program appears to have some gene expression overlap with the canonical CMS2 subtype (new **Extended Data Fig. 6d**), but unlike the tumor ISC-like program, it is associated with poor outcomes, similar to that seen with the squamous and neuroendocrine non-canonical differentiation programs in the larger patient cohorts of the TCGA (445 patients) and locally advanced rectal cancer (108 patients) (**Figure 3h, i**). Overall, we think this reflects some of the limitations of the bulk CMS classifier system, which necessarily merges disparate gene expression programs with distinct biological meaning, and that can only be resolved with single-cell resolution data, into a small number of categories. However, given that we only have a few cells scoring highly for the osteoblast module from a few patients, we avoid making strong statements about the osteoblast module in this manuscript.

Comment 3.28. Line 266-267: This claim could be backed up with a volcano plot.

In our revised manuscript, we now directly assess the upregulation of our fetal signature genes in tumor cells from KG146, KG182, KG183, and KG150, and find the same gene programs upregulated as in our original submission (**Supplementary Table 4**).

Comment 3.29. Lines 328-333: a mutual nearest neighbor approach is used to map organoid cells to states of the primary tumor. This approach makes it difficult to evaluate how similar the organoid states are to the primary tumor states - even if the similarity is extremely low the mapping might still be defined. Therefore, it would be good to directly demonstrate the similarity, for example by plotting the organoid expression of sets of genes (not individual markers) that define the tumor states. If the similarity is limited then that might justify mapping only the subset of cells with high similarity.

We do plot the expression of gene sets defining tumor states in **Extended Data Fig. 8d**, which is referenced throughout the paragraph that the reviewer is referring to.

Comment 3.30. Fig. 6f-i: the inclusion of only two genes as a proxy for each state gives an impression of cherry picking, and accordingly it would be good to also show all signature genes or at least their average.

We note that these genes were selected as the most “canonical” marker genes of each state and the use of marker genes is widely used (to avoid expensive scRNA-seq for each perturbation). Nevertheless, we now include full scRNA-seq analysis clearly showing that *PROX1* knockdown in OKG146P, but not OKG146Li, strongly induces non-canonical gene expression, providing global support for our claims (new Fig. 6c). However, surveying the entire transcriptome rather than the few originally assayed genes reveals more complex expression patterns: although *PROX1* knockdown causes much more dramatic changes to transcription in primary organoids, including a majority of the non-canonical genes (see **Supplementary Table 5** for a full list), a small set of intestinal differentiation and non-canonical differentiation genes become upregulated in 146Li organoids, suggesting that they are also repressed by *PROX1* in the metastatic context. These data suggest that *PROX1* repression of non-canonical genes is dampened in non-canonical states.

Figure 6c | *PROX1* knockdown causes a strong global shift toward non-canonical states in primary tumor organoids, and a dampened shift in metastasis organoids. Probability of classifying cells as non-canonical based on scRNA-seq data from 146 primary (left) and liver metastasis (right) shControl and shPROX1 organoid lines, 7 d after induction with dox. Cells were classified as canonical or non-canonical using a manifold-based classifier that combines methods from Harmony²⁰ and PhenoGraph¹⁷ (Methods). ***, $p < 0.001$, two-sided t-test.

To test this hypothesis, we engineered additional organoid lines with shControl and two shPROX1 lentiviral constructs, for a total of 7 organoids spanning the canonical to non-canonical spectrum (136P, 136Li, 173Li, 146P, 146Li, 182CW2, 183Li2). Using our scRNA-seq data from KG146, we selected representative non-canonical genes that change in primary alone, or in both primary and metastasis, after *PROX1* knockdown to profile with qPCR (Fig. 6d).

Figure 6d | *PROX1* knockdown de-represses non-canonical gene expression in multiple tumor organoid lines. Marker expression relative to shControl (log fold-change) in primary and metastasis organoids upon dox-inducible knockdown of *PROX1*. All genes mark non-canonical differentiation, except for *TFF3* and *FABP1* (canonical intestinal differentiation) and *PROX1*. Organoids were cultured in HISC media containing 2 μ g/ml dox for 7 d. Lines are ordered by non-canonical gene induction following *PROX1* knockdown, and RT-qPCR is normalized to *GAPDH* expression ($n = 4$, * $p < 0.05$, t-test, Benjamini-Hochberg correction).

Consistent with observations in our first submission, the response of an organoid line to loss of *PROX1* depends on the phenotypic state of its cells; we find that *PROX1* knockdown induces multiple non-canonical genes (e.g. squamous markers *KRT23*, *ELF5*, *TRPS1*) in more canonical organoids (136P, 136Li, 146P), but it does not upregulate these in the most non-canonical organoid lines (182CW2, 183Li2). In contrast, a subset of non-canonical genes (e.g. *NRXN3*, *POMC*), which retain *PROX1* sensitivity in the 146Li *PROX1* knockdown scRNA-seq data, are upregulated by *PROX1* knockdown even in the most non-canonical organoids (new **Fig. 6d**). Across organoid lines, *PROX1* knockdown does not consistently alter the expression of canonical intestinal differentiation genes (*TFF3*, *FABP1*).

Comment 3.31. Line 455-456: I agree that this is an important question. I wonder if this can be addressed here: Given that all patients were profiled by MSK-IMPACT it would be desirable to have an analysis investigating correlations between genomic alterations and cell state abundancies. This should also be done using TCGA: Deconvolution of bulk primaries and correlate cell-state abundancies with genomic alterations (of course possible that the primaries don't show enough signal for the non-canonical states).

We thank the reviewer for raising the important question of whether any pre-existing patient or tumor features could serve as biomarkers that predict likelihood of the non-canonical state. The tumor genotypes of all samples in our scRNA-seq cohort, where patient consent for mutational profiling was available, are indicated in **Extended Data Fig. 1b**. Across the four patients whose tumors show the greatest non-canonical differentiation (KG146, KG150, KG182, KG183), the only shared mutations are in *APC* and *TP53*. However, these are the two most commonly mutated genes across metastatic CRC, and are shared by most patients in our cohort (*TP53*: 93%; *APC*: 81%). Thus, in our small cohort, we are unable to establish any genotype:phenotype relationship. Given the combinatorial genetic heterogeneity across patients, such correlations typically require patient cohorts in the thousands to achieve statistical power and account for multiple hypothesis testing (e.g., Nguyen et al.¹⁹).

Nevertheless, as a first attempt to address the question of genotype association with non-canonical phenotypes, we interrogated all tumor samples for which bulk RNA-seq and tumor mutation profiling information was available in the TCGA ($n = 239$)³, our MSK locally advanced rectal cancer (LARC) cohort (Chatila et al.⁶), and pseudo-bulked samples from our scRNA-seq cohort ($n = 56$). To define associations between specific oncogenic alterations (filtered as in Sanchez-Vega et al.²⁴ and using OncoKB⁴) and phenotypes, we first identified mutations that recurred in at least 10 patients in the TCGA or LARC cohorts, or at least 5 patients in our scRNA-seq cohort. For each cohort, we performed differential gene expression analysis on samples with and without each oncogenic mutation, and then ran gene set enrichment analysis (GSEA) using our individual and grouped Hotspot modules and other gene signatures. We identified shared combinations of oncogenic alterations and significant differentially enriched signatures across the three cohorts (**Fig. R8**). These data suggest that oncogenic mutations in *FBXW7*, *PIK3CA* and *TCF7L2* might be associated with decreased expression of absorptive intestine signatures (**Fig. R9**), but much larger datasets will be needed in the future to rigorously interrogate tumor genotype:phenotype relationships in a statistically sound manner.

Figure R8 | Workflow to identify genotype:phenotype associations.

Gene	Geneset	Description	Direction
APC	HotSpot	Module 11 (Squamous)	Up
APC	HotSpot	Module 16 (Secretory Intestine)	Down
APC	HotSpot	Module 23 (Injury Repair)	Down
FBXW7	Grouped	Absorptive Intestine	Down
FBXW7	HotSpot	Module 3 (Absorptive Intestine)	Down
KRAS	HotSpot	Module 18 (Neuroendocrine)	Down
PIK3CA	Grouped	Absorptive Intestine	Down
PIK3CA	HotSpot	Module 2 (Cell Cycle)	Up
PIK3CA	HotSpot	Module 3 (Absorptive Intestine)	Down
TCF7L2	Grouped	Absorptive Intestine	Down
TCF7L2	HotSpot	Module 3 (Absorptive Intestine)	Down

Figure R9 | Preliminary findings of association analysis.

Tables

Organoid lines	Experiment	Figure
Lines that are not engineered		
146P, 146Li, 182CW2, 183Li2	Plasticity of non-canonical organoid lines in IGFF and HISC media	Fig. 5b; ED Fig. 8
146P, 146Li, 136P, 136Li	In vivo orthotopic growth of primary- and met-derived organoids in the cecum and liver	Fig. 5c-f; ED Fig. 8g,h
146P, 146Li, 136P, 136Li, 173P, 173Li	Induction of non-canonical gene expression upon irinotecan chemotherapy treatment	Fig. 5g,h; ED Fig. 8i,j
Lines engineered with shCtrl, shPROX1-1 and shPROX1-2		
136P, 136Li, 146P, 146Li, 173Li, 182CW2, 183Li2	Change in canonical/non-canonical gene expression upon PROX1 knockdown, assayed by scRNA-seq (146P, 146Li) and qPCR (all lines in column 1, which span the canonical to non-canonical phenotypic axis)	Fig. 6c,d
146P, 146Li 136P, 136Li, 173Li, 182CW2, 183Li2	Change in organoid formation efficiency (intestinal stem cell function) upon PROX1 knockdown	ED Fig. 9e
136P, 146P	In vivo orthotopic growth of primary-derived organoids in liver upon PROX1 knockdown	ED Fig. 9g,h
136P, 136Li, 146P, 146Li	Change in canonical/non-canonical gene expression upon PROX1 overexpression in treated organoids	Fig. R1b
136P, 136Li, 146P, 146Li	Change in organoid formation efficiency (intestinal stem cell function) upon PROX1 overexpression	Fig. R1c

Table R1 | Organoid experiments related to *PROX1* function. Figures are numbered as in the revised manuscript. New figure panels and newly tested lines in revision are indicated in bold.

Study first author	No. patients	No. Samples				No. epithelial cells			Total cells	% epi.	Epi. cells/sample
		Primary	Met.	Normal	All	Tumor	Normal	Total			
Moorman (this work)	31	29	33	21	83	26620	20817	47437	164304	29	572
Wang ²⁹	6	6	6	15	27			23954	238365	10	887
Joanito ¹³	34	106	0	23	138			27260	281955	10	198
Lee ¹⁵	29	35	0	16	51	19681	2214	21895	91103	24	429
Sathe ²⁵	7	0	7	7	14			9714	38864	25	694
Bian ¹	8	8	8	8	24			N/A	1188	N/A	50
Wang ³⁰	12	9	22	11	42			5413	8085	67	129
Li ¹⁷	11	11	0	11	22	375	215	590	N/A	N/A	27

Table R2 | Sample sizes and epithelial cell numbers in scRNA-seq studies of CRC.

References

1. Bian S, Hou Y, Zhou X, Li X, Yong J, Wang Y, et al. Single-cell multiomics sequencing and analyses of human colorectal cancer. *Science*. 2018;362(6418):1060-3. doi: 10.1126/science.aao3791.
2. Butler A, Hoffman P, Smibert P, Papalexi E, Satija R. Integrating single-cell transcriptomic data across different conditions, technologies, and species. *Nat Biotechnol*. 2018;36(5):411-20. Epub 20180402. doi: 10.1038/nbt.4096.
3. Cancer Genome Atlas N. Comprehensive molecular characterization of human colon and rectal cancer. *Nature*. 2012;487(7407):330-7. Epub 20120718. doi: 10.1038/nature11252.
4. Chakravarty D, Gao J, Phillips SM, Kundra R, Zhang H, Wang J, et al. OncoKB: A Precision Oncology Knowledge Base. *JCO Precis Oncol*. 2017;2017. Epub 20170516. doi: 10.1200/PO.17.00011.
5. Chan JM, Quintanal-Villalonga A, Gao VR, Xie Y, Allaj V, Chaudhary O, et al. Signatures of plasticity, metastasis, and immunosuppression in an atlas of human small cell lung cancer. *Cancer Cell*. 2021;39(11):1479-96 e18. Epub 20211014. doi: 10.1016/j.ccell.2021.09.008.
6. Chatila WK, Kim JK, Walch H, Marco MR, Chen CT, Wu F, et al. Genomic and transcriptomic determinants of response to neoadjuvant therapy in rectal cancer. *Nat Med*. 2022;28(8):1646-55. Epub 20220815. doi: 10.1038/s41591-022-01930-z.
7. Elmentaite R, Kumasaka N, Roberts K, Fleming A, Dann E, King HW, et al. Cells of the human intestinal tract mapped across space and time. *Nature*. 2021;597(7875):250-5. Epub 20210908. doi: 10.1038/s41586-021-03852-1.
8. Greenwald NF, Miller G, Moen E, Kong A, Kagel A, Dougherty T, et al. Whole-cell segmentation of tissue images with human-level performance using large-scale data annotation and deep learning. *Nat Biotechnol*. 2022;40(4):555-65. Epub 20211118. doi: 10.1038/s41587-021-01094-0.
9. Guo M, Jin N, Pawlik T, Cloyd JM. Neoadjuvant chemotherapy for colorectal liver metastases: A contemporary review of the literature. *World J Gastrointest Oncol*. 2021;13(9):1043-61. doi: 10.4251/wjgo.v13.i9.1043.
10. Haghverdi L, Buttner M, Wolf FA, Buettner F, Theis FJ. Diffusion pseudotime robustly reconstructs lineage branching. *Nat Methods*. 2016;13(10):845-8. Epub 20160829. doi: 10.1038/nmeth.3971.
11. Jansky S, Sharma AK, Körber V, Quintero A, Toprak UH, Wecht EM, et al. Single-cell transcriptomic analyses provide insights into the developmental origins of neuroblastoma. *Nat Genet*. 2021;53(5):683-+. doi: 10.1038/s41588-021-00806-1.
12. Jerby-Arnon L, Shah P, Cuoco MS, Rodman C, Su MJ, Melms JC, et al. A Cancer Cell Program Promotes T Cell Exclusion and Resistance to Checkpoint Blockade. *Cell*. 2018;175(4):984-97 e24. doi: 10.1016/j.cell.2018.09.006.
13. Joanito I, Wirapati P, Zhao N, Nawaz Z, Yeo G, Lee F, et al. Single-cell and bulk transcriptome sequencing identifies two epithelial tumor cell states and refines the consensus molecular classification of colorectal cancer. *Nat Genet*. 2022;54(7):963-75. Epub 20220630. doi: 10.1038/s41588-022-01100-4.

14. Laughney AM, Hu J, Campbell NR, Bakhoun SF, Setty M, Lavalley VP, et al. Regenerative lineages and immune-mediated pruning in lung cancer metastasis. *Nat Med.* 2020;26(2):259-69. Epub 20200210. doi: 10.1038/s41591-019-0750-6.
15. Lee HO, Hong Y, Etioglu HE, Cho YB, Pomella V, Van den Bosch B, et al. Lineage-dependent gene expression programs influence the immune landscape of colorectal cancer. *Nat Genet.* 2020;52(6):594-603. Epub 20200525. doi: 10.1038/s41588-020-0636-z.
16. Levine JH, Simonds EF, Bendall SC, Davis KL, Amir el AD, Tadmor MD, et al. Data-Driven Phenotypic Dissection of AML Reveals Progenitor-like Cells that Correlate with Prognosis. *Cell.* 2015;162(1):184-97. Epub 20150618. doi: 10.1016/j.cell.2015.05.047.
17. Li H, Courtois ET, Sengupta D, Tan Y, Chen KH, Goh JLL, et al. Reference component analysis of single-cell transcriptomes elucidates cellular heterogeneity in human colorectal tumors. *Nat Genet.* 2017;49(5):708-18. Epub 20170320. doi: 10.1038/ng.3818. PubMed PMID: 28319088.
18. Mustata RC, Vasile G, Fernandez-Vallone V, Strollo S, Lefort A, Libert F, et al. Identification of Lgr5-independent spheroid-generating progenitors of the mouse fetal intestinal epithelium. *Cell Rep.* 2013;5(2):421-32. Epub 20131017. doi: 10.1016/j.celrep.2013.09.005.
19. Nguyen B, Fong C, Luthra A, Smith SA, DiNatale RG, Nandakumar S, et al. Genomic characterization of metastatic patterns from prospective clinical sequencing of 25,000 patients. *Cell.* 2022;185(3):563-75 e11. doi: 10.1016/j.cell.2022.01.003.
20. Nowotschin S, Setty M, Kuo YY, Liu V, Garg V, Sharma R, et al. The emergent landscape of the mouse gut endoderm at single-cell resolution. *Nature.* 2019;569(7756):361-7. Epub 20190408. doi: 10.1038/s41586-019-1127-1.
21. Patel AP, Tirosh I, Trombetta JJ, Shalek AK, Gillespie SM, Wakimoto H, et al. Single-cell RNA-seq highlights intratumoral heterogeneity in primary glioblastoma. *Science.* 2014;344(6190):1396-401. Epub 20140612. doi: 10.1126/science.1254257.
22. Qin X, Cardoso Rodriguez F, Sufi J, Vlckova P, Claus J, Tape CJ. An oncogenic phenoscape of colonic stem cell polarization. *Cell.* 2023;186(25):5554-68 e18. doi: 10.1016/j.cell.2023.11.004.
23. Ramos Zapatero M, Tong A, Opzoomer JW, O'Sullivan R, Cardoso Rodriguez F, Sufi J, et al. Trellis tree-based analysis reveals stromal regulation of patient-derived organoid drug responses. *Cell.* 2023;186(25):5606-19 e24. doi: 10.1016/j.cell.2023.11.005.
24. Sanchez-Vega F, Mina M, Armenia J, Chatila WK, Luna A, La KC, et al. Oncogenic Signaling Pathways in The Cancer Genome Atlas. *Cell.* 2018;173(2):321-37 e10. doi: 10.1016/j.cell.2018.03.035.
25. Sathe A, Mason K, Grimes SM, Zhou Z, Lau BT, Bai X, et al. Colorectal Cancer Metastases in the Liver Establish Immunosuppressive Spatial Networking between Tumor-Associated SPP1+ Macrophages and Fibroblasts. *Clin Cancer Res.* 2023;29(1):244-60. doi: 10.1158/1078-0432.CCR-22-2041.
26. Tirosh I, Izar B, Prakadan SM, Wadsworth MH, 2nd, Treacy D, Trombetta JJ, et al. Dissecting the multicellular ecosystem of metastatic melanoma by single-cell RNA-seq. *Science.* 2016;352(6282):189-96. doi: 10.1126/science.aad0501.

27. Vasquez EG, Nasreddin N, Valbuena GN, Mulholland EJ, Belnoue-Davis HL, Eggington HR, et al. Dynamic and adaptive cancer stem cell population admixture in colorectal neoplasia. *Cell Stem Cell*. 2022;29(8):1213-28 e8. doi: 10.1016/j.stem.2022.07.008.
28. Wagner DE, Weinreb C, Collins ZM, Briggs JA, Megason SG, Klein AM. Single-cell mapping of gene expression landscapes and lineage in the zebrafish embryo. *Science*. 2018;360(6392):981-7. Epub 20180426. doi: 10.1126/science.aar4362.
29. Wang F, Long J, Li L, Wu ZX, Da TT, Wang XQ, et al. Single-cell and spatial transcriptome analysis reveals the cellular heterogeneity of liver metastatic colorectal cancer. *Sci Adv*. 2023;9(24):eadf5464. Epub 20230616. doi: 10.1126/sciadv.adf5464.
30. Wang R, Li J, Zhou X, Mao Y, Wang W, Gao S, et al. Single-cell genomic and transcriptomic landscapes of primary and metastatic colorectal cancer tumors. *Genome Med*. 2022;14(1):93. Epub 20220816. doi: 10.1186/s13073-022-01093-z.
31. Yui S, Azzolin L, Maimets M, Pedersen MT, Fordham RP, Hansen SL, et al. YAP/TAZ-Dependent Reprogramming of Colonic Epithelium Links ECM Remodeling to Tissue Regeneration. *Cell Stem Cell*. 2018;22(1):35-49 e7. Epub 20171214. doi: 10.1016/j.stem.2017.11.001.

Reviewer Reports on the First Revision:

Referees' comments:

Referee #1 (Remarks to the Author)

This manuscript provides an unprecedented view of the transcriptional changes occurring during CRC progression. The exhaustive nature of the scRNAseq analysis of matched primary and metastatic tumors and organoids is a major strength of the study that will provide a very useful resource for the CRC community. The manuscript also provides much needed clarification of what the fetal-like state is in the context of human CRC that was previously addressed only in mouse models. Finally, the authors have substantially improved the manuscript by satisfactorily addressing previous comments about the limited number of replicates in key figures and the role of therapy in the driving plasticity.

Overall, as a resource paper, the manuscript does provide sufficiently novel data to warrant publication in my view and have a major impact in the field. It should be pointed out that the question of mechanistic insight provided by the study still persists in this latest version. The manuscript is primarily a descriptive account and does not uncover a clear role for non-canonical states or PROX1 in driving tumor progression. In addition, there are question marks that persist regarding the relevance of the fetal state in progression towards differentiated non-canonical states (see specific comments below).

Specific comments

On line 403, the authors state: «Functionally, PROX1 downregulation is not sufficient to induce outgrowth of liver metastasis in mice transplanted with canonical primary tumor-derived organoids (OKG146P, OKG136P) and followed for 12 weeks, suggesting that PROX1-driven non-canonical differentiation cooperates with other phenotypic drivers to promote metastatic outgrowth (Extended Data Fig. 9 e,f).

The authors are correct in stating that PROX1 downregulation is not sufficient to induce outgrowth. But they fail to comment on the fact that PROX1 downregulation suppressed tumor outgrowth instead of enhancing it in OKG146P (a patient that served to support many of the claims elsewhere in the manuscript). Whereas in other lines no effect was observed. If upregulation of the non-canonical program is associated with suppressed tumorigenicity in certain cases, while in other cases it has no effect then does this suggest that the non-canonical program is a bystander event to other more important “phenotypic drivers”? Although it should be noted in favor of the authors’ claims, elsewhere in the manuscript that non-canonical gene upregulation is a marker of poor outcomes in CRC. But the limitation of the manuscript is that it does not shed light into why this is. Does increased plasticity promote tumor progression or does it correlate with it? PROX1 may regulate this program in certain contexts but if this has no effect on tumor progression how important is this factor?

The general claim from the manuscript is that fetal gene program serves as a bridge to non-canonical

states. In the rebuttal the term “conduit” is used to further illustrate this point. Regardless the functional relevance of the fetal gene program was not addressed in the manuscript. It appears indeed that fetal reprogramming precedes non-canonical states in patient KG146 (Fig 4b) and is positioned at the branch point of neuroendocrine and squamous cells in EDF7f. But in other patients the fetal gene program appears to co-emerge with other non-canonical states (EDF7a). In the text the authors mention that:

“we employed Palantir trajectory inference and independently applied it to patients KG146, KG182, and KG150, which confirmed the bifurcation of squamous and neuroendocrine trajectories and placed the fetal signature between branches.”

Indeed, the authors appear to demonstrate that bifurcation exists (Fig4g) across different patients but it's not clear that the fetal signature is the branchpoint in all patients. If that's the case, why is only KG146 shown in EDF7f? Ultimately, these observations are correlative. If the authors want to imply a functional role for the fetal state as a conduit to increased plasticity and eventually metastasis, this should be tested by ablation of fetal cells in an orthotopic model. I understand that this would be a very challenging experiment at this stage. Additional evidence that fetal cells are at the “branch point” in multiple patients would strengthen the claims.

Minor comment

In EDF5d, the authors appear to show that CK5, a marker of the squamous cell module is not significantly enriched in metastasis by image analysis. This does not support claims in Fig 3f and EDF7e that non-canonical and fetal gene programs are enriched in metastasis. What are the trends observed across the patient cohort when looking at the overall gene module specifically for squamous and neuroendocrine programs? Is there a difference at the gene module level?

Referee #2 (Remarks to the Author)

The authors have addressed most of my concerns, and presented a revised manuscript which is significant strengthened, particularly by characterising the fetal/non-canonical signatures across a large cohort of patient samples.

My only remaining question is the role of PROX1 in their KD study. The authors concluded that PROX1 functions to repress non-intestinal lineage differentiation in the fetal progenitor state. This indicates that cancer cells undergo stepwise but independent reprogramming, first to fetal-like state, and second to non-intestinal/canonical differentiation. The authors indicate that PROX1 regulates the second stage of non-intestinal differentiation. What about the preceding fetal-like state? Is fetal state affected or unaltered in the PROX1 KD organoids? It is important to clarify the context as it seems ambiguous in the text, particularly in the abstract. The authors should indicate if PROX1 regulates fetal cell state or not.

Given that non-canonical signature is enriched in metastasis and that PROX1 regulates non-canonical differentiation, the abstract gives the impression that PROX1 is the key regulator for metastasis. Since

metastasis is not affected in PROX1 KD, it'd help to clarify that PROX1 loss promotes non-intestinal lineage differentiation but not sufficient to drive metastasis.

Referee #3 (Remarks to the Author)

We are fully satisfied with the revisions and the responses to our comments, and congratulate the authors for this important study.

Rouven Hoefflin and Itay Tirosh

Author Rebuttals to First Revision:

Referee 1

Comment 1.1. This manuscript provides an unprecedented view of the transcriptional changes occurring during CRC progression. The exhaustive nature of the scRNAseq analysis of matched primary and metastatic tumors and organoids is a major strength of the study that will provide a very useful resource for the CRC community. The manuscript also provides much needed clarification of what the fetal-like state is in the context of human CRC that was previously addressed only in mouse models. Finally, the authors have substantially improved the manuscript by satisfactorily addressing previous comments about the limited number of replicates in key figures and the role of therapy in the driving plasticity.

Overall, as a resource paper, the manuscript does provide sufficiently novel data to warrant publication in my view and have a major impact in the field. It should be pointed out that the question of mechanistic insight provided by the study still persists in this latest version. The manuscript is primarily a descriptive account and does not uncover a clear role for non-canonical states or PROX1 in driving tumor progression. In addition, there are question marks that persist regarding the relevance of the fetal state in progression towards differentiated non-canonical states (see specific comments below).

We thank the reviewer for valuing the impact of this exceptional resource and the discovery of the novel fetal intermediate state. We address the questions about *PROX1* mechanism below.

Specific comments

Comment 1.2. On line 403, the authors state: «Functionally, PROX1 downregulation is not sufficient to induce outgrowth of liver metastasis in mice transplanted with canonical primary tumor-derived organoids (OKG146P, OKG136P) and followed for 12 weeks, suggesting that PROX1-driven non-canonical differentiation cooperates with other phenotypic drivers to promote metastatic outgrowth (Extended Data Fig. 9 e,f).

The authors are correct in stating that PROX1 downregulation is not sufficient to induce outgrowth. But they fail to comment on the fact that PROX1 downregulation suppressed tumor outgrowth instead of enhancing it in OKG146P (a patient that served to support many of the claims elsewhere in the manuscript). Whereas in other lines no effect was observed.

PROX1 may regulate this program in certain contexts but if this has no effect on tumor progression how important is this factor?

To first address the question of whether *PROX1* is important—our study presents clear perturbation results from multiple patient-derived organoid lines, that *PROX1* (which is induced in the fetal state) inhibits plasticity and non-canonical differentiation. We further show that non-canonical states are enriched in metastases and associated with poor outcomes in patients. Moreover, lineage transformation is known to lead to drug resistance in prostate and lung cancers (Quintanal-Villalonga et al., *Nat Rev Clin Oncol.* 2020). Although this study does not establish a causal link between *PROX1* and tumor progression, we argue that there is a critical need to identify regulators of plasticity in cancer.

With respect to metastatic outgrowth, we point out that metastasis is a complicated, multi-step process, and we do not expect a single factor to drive it in tumor-derived organoids. Indeed, unlike tumor suppressors and oncogenes, the field has failed to recover individual mutated genes with metastasis-specific phenotypes.

In the metastatic assays that were requested to test the effect of inducible *PROX1* knockdown, line OKG136 showed non-significant outgrowth at 12 weeks, whereas a slight but significant *reduction* was observed in OKG146P (**Extended Data Fig. 9e,f**). A potential reason for this discrepancy is the drop in survival of injected cells (i.e., poor metastatic seeding) observed in week 1-2 in OKG146P organoids bearing shPROX1 constructs. In contrast, subsequent growth in these lines was similar to shControl, and poor metastatic seeding was not observed upon *PROX1* knockdown in OKG136P. Seminal work by Fred de Sauvage (de Sousa e Melo *et al.*, *Nature* 2017) showed that metastatic seeding in mouse CRC organoids involves entry into an LGR5⁺ ISC state, followed by differentiation into LGR5⁻ states upon outgrowth. This was confirmed by the Jacco van Rheenen lab (Fumagalli *et al.*, *Cell Stem Cell* 2020), while the Hugo Snippert lab showed that some human CRC organoids depend on LGR5⁺ ISCs for metastatic seeding (Heinz *et al.*, *Cancer Research* 2022). Indeed, our overlap plots show a progression in multiple steps, from intestinal to tumor-ISC, to fetal, to non-canonical states (see new **Fig. 4d–f** in **Comment 1.3** below). We observed that *PROX1* knockdown in OKG146P significantly inhibits organoid reinitiation, i.e., entry into an ISC state, whereas the effect is less pronounced in OKG136P (**Extended Data Fig. 9g**). This inability to enter an LGR5⁺ ISC state may explain why OKG146P shPROX1 organoids experience a sharp drop-off in metastatic seeding.

However, the function of *PROX1* in LGR5⁺ ISC state re-entry and its role in metastatic seeding are tangential to the central claims of our paper, which focuses on the acquisition of non-canonical states during metastatic outgrowth following the initial seeding, and would require many months of experiments to investigate.

In preliminary results at 20 weeks post-injection, we do see that OKG136P cells express CK5 (squamous marker) in an shPROX1 but not an shControl context, reinforcing the role of *PROX1* in suppressing non-canonical expression.

REDACTED

We already demonstrated that *PROX1* function is context-specific and complex, and we now address the organoid metastatic outgrowth results more directly in the Discussion:

While we show that *PROX1* loss promotes non-intestinal lineage differentiation, downregulation of this single factor is not sufficient to drive metastatic outgrowth of primary tumor-derived organoids within 12 weeks of transplantation to mouse liver. These observations underscore that metastasis is a complex, multi-step process requiring multiple factors.

Comment 1.3. If upregulation of the non-canonical program is associated with suppressed tumorigenicity in certain cases, while in other cases it has no effect then does this suggest that the non-canonical program is a bystander event to other more important “phenotypic drivers”? Although it should be noted in favor of the authors’ claims, elsewhere in the manuscript that non-canonical gene upregulation is a marker of poor outcomes in CRC. But the limitation of the manuscript is that it does not shed light into why this is. Does increased plasticity promote tumor progression or does it correlate with it?

As the reviewer points out, we have shown that non-canonical states are not only present in our cohort and another independent single-cell cohort, but that they are significant markers of poor outcomes in two large, independent CRC cohorts. We interpret this to mean that they are not bystanders but play some (as-yet undefined) role in tumor pathology. We do not claim to understand or investigate this role, and again stress that our focus is on identifying putative regulators of non-canonical states. Exactly how plasticity and non-canonical states promote progression and metastasis are important follow-up questions provoked by our work.

Moreover, we point out that a key result from our work is that *PROX1* function depends on the underlying state of each cell; the fact that metastases lose *PROX1* dependence compared to their matched primary (Fig. 6d) further demonstrates the importance of this factor. The substantial phenotypic variability that we observe between patients reflects the real and challenging complexity of cancer in the clinical setting.

Comment 1.4. The general claim from the manuscript is that fetal gene program serves as a bridge to non-canonical states. In the rebuttal the term “conduit” is used to further illustrate this point. Regardless the functional relevance of the fetal gene program was not addressed in the manuscript. It appears indeed that fetal reprogramming precedes non-canonical states in patient KG146 (Fig 4b) and is positioned at the branch point of neuroendocrine and squamous cells in EDF7f. But in other patients the fetal gene program appears to co-emerge with other non-canonical states (EDF7a).

This is an important point, and we thank the reviewer for spurring us to better present it. We now:

- 1) Identify exactly when the fetal signature and dominant non-canonical module for each patient reach 0.75 of their progression along diffusion component 1 (new Fig. 4b and Extended Data Fig. 7a), demonstrating quantitatively that the fetal signature precedes non-canonical states.

Figure 4b and Extended Data Figure 7a | Cell state progression along diffusion component 1. Top, trends in gene module scores along DC1, observed in all tumor cells from patient KG146 (4b), KG152, KG180, KG183, and patient s1321 from Wang et al.⁴² (7a) (Methods). Each row depicts the module score along DC1 from the 20th percentile value (white) to the maximum value (highest saturation). Expression of the fetal signature peaks before non-canonical trends in all cases (vertical lines correspond to 75th percentile for the fetal signature and predominant terminal non-canonical module for each patient). Bottom, positions of tumor cells along DC1.

2) Add the core fetal signature to our overlap plot analysis (new Fig. 4d–f). The core signature, which exhibits strong shared signal across patients in our cohort, clearly acts as a bridge that overlaps with both intestinal and non-intestinal differentiated fates in all single-cell cohorts. Importantly, in both our cohort and the independent Wang cohort we see that the fetal state not only emerges earlier, but is co-expressed with both Squamous and Neuroendocrine cells, clearly demonstrating that it is indeed a branch point for many additional patients.

Figure 4d–f | Cell state progression across different cohorts. d–f, Fraction of cells expressing >0.75 quantile score for a given module or the core fetal signature in d) samples from the 4 patients with most non-canonical cells, e) all samples in our cohort, and f) samples from the Wang et al. cohort. Shared module expression reveals a consistent progression from canonical to non-canonical fates via a fetal intermediate across patients and cohorts.

3) Clarify that the fetal state, which is derived from human fetal datasets, is closely related to the Endoderm Development (ED) module, which is derived from our own human patient data using Hotspot analysis. While we had shown that the ED state precedes non-canonical states (Fig. 4d–f) and is associated with poor outcomes (Extended Data Fig. 6), we did not properly explain its close relationship with the fetal state. We now show that the two are very highly correlated in all tumor samples (new Extended Data Fig. 7e).

Extended Data Figure 7e | The core fetal signature is highly correlated with the Endoderm Development module. Relative correlation of the core fetal signature with different modules, showing highest correlation for the Endoderm Development module across all tumors.

Our analyses thus consistently reveal the existence of a highly dedifferentiated fetal state that appears prior to the acquisition of diverse non-canonical fates across multiple patients and cohorts.

Comment 1.5. In the text the authors mention that: “we employed Palantir trajectory inference and independently applied it to patients KG146, KG182, and KG150, which confirmed the bifurcation of squamous and neuroendocrine trajectories and placed the fetal signature between branches.” Indeed, the authors appear to demonstrate that bifurcation exists (Fig4g) across different patients but it’s not clear that the fetal signature is the branchpoint in all patients. If that’s the case, why is only KG146 shown in EDF7f? Ultimately, these observations are correlative. If the authors want to imply a functional role for the fetal state as a conduit to increased plasticity and eventually metastasis, this should be tested by ablation of fetal cells in an orthotopic model. I understand that this would be a very challenging experiment at this stage. Additional evidence that fetal cells are at the “branch point” in multiple patients would strengthen the claims.

Our augmented analyses (in response to **Comment 1.4**) make it clear that the fetal state is a conduit that precedes non-canonical states in all patients. The overlap plots (**Fig. 4d–f**) in particular show that only the fetal state contains both (1) cells that substantially co-express fetal and canonical, and (2) cells that substantially co-express fetal and non-canonical programs. We find cells that specifically co-express fetal and neuroendocrine, or fetal and squamous programs across multiple patients.

In force-directed layouts of the three patients with enough cells from both non-canonical lineages, we likewise observe that the fetal state inhabits a state between neuroendocrine and squamous lineage branches (**Fig. R2**). This expands our analysis in **Extended Data Fig. 7f** beyond a single patient, but we do not believe that it adds to the manuscript, as it is redundant with **Fig. 4d–f** and **Extended Data Fig. 7a**, and makes the point of progression less effectively. We present it here as an additional view of the data.

Figure R2 | Fetal state cells appear before neuroendocrine and squamous bifurcation. FDL of cancer cells from three patients colored by fetal (top), neuroendocrine (middle) and squamous (bottom) expression, showing that the fetal state sits between the two non-canonical states.

We agree that ablating fetal-state cells following orthotopic transplantation would be very exciting, and it would definitively establish the functional relevance of the fetal state for progression. However, it requires involved long-term experiments that are beyond the scope of this paper. We are currently developing a pharmacological agent to ablate cells in the fetal state, as one of several follow-up studies that will underscore the clinical importance of this state.

Minor comment

Comment 1.6. In EDF5d, the authors appear to show that CK5, a marker of the squamous cell module is not significantly enriched in metastasis by image analysis. This does not support claims in Fig 3f and EDF7e that non-canonical and fetal gene programs are enriched in metastasis. What are the trends observed across the patient cohort when looking at the overall gene module specifically for squamous and neuroendocrine programs? Is there a difference at the gene module level?

We observe a significant depletion of canonical programs and enrichment of non-canonical programs in metastases relative to primary tumors in our single-cell RNA-seq data, both in the Moorman et al. (Figs. 3f,g and Extended Data Fig. 7e) and Wang et al. (Extended Data Fig. 5g) cohorts. Image analysis of CK5 protein staining was more challenging, as tumor sections contained relatively few positive cells; while we did observe higher CK5 expression in metastases across the cohort, it did not reach statistical significance. Individual markers do vary in expression between patients, which is why we base the claim of enrichment of non-canonical programs on the much more robust and extensive analysis of multiple markers in scRNA-seq data.

In addition to the highly significant ($p < 0.001$) enrichment of non-canonical programs (Fig. 3f) we find that squamous and neuroendocrine cell states are indeed separately both enriched in metastases compared to primary tumors in scRNA-seq data (Fig. R3). The enrichment p -values are lower because there are fewer cells for comparison when we split the data into the two non-canonical states.

Figure R3 | Metastatic enrichment of non-canonical module expression. Log-ratio of metastasis- to primary-derived tumor cells that exhibit > 0.75 quantile score for the 14-gene conserved fetal signature, for each patient sample ($p = 0.0004$, rank sum test; Methods).

Referee 2

Comment 2.1. The authors have addressed most of my concerns, and presented a revised manuscript which is significant strengthened, particularly by characterising the fetal/non-canonical signatures across a large cohort of patient samples.

We thank the reviewer for their positive appraisal!

Comment 2.2. My only remaining question is the role of PROX1 in their KD study. The authors concluded that PROX1 functions to repress non-intestinal lineage differentiation in the fetal progenitor state. This indicates that cancer cells undergo stepwise but independent reprogramming, first to fetal-like state, and second to non-intestinal/canonical differentiation. The authors indicate that PROX1 regulates the second stage of non-intestinal differentiation. What about the preceding fetal-like state? Is fetal state affected or unaltered in the PROX1 KD organoids? It is important to clarify the context as it seems ambiguous in the text, particularly in the abstract. The authors should indicate if PROX1 regulates fetal cell state or not.

We did not make any claims as to whether *PROX1* regulates the fetal state itself, but we thank the reviewer for raising this very interesting question. We find that *PROX1* does indeed inhibit some fetal gene expression in primary tumors (new **Extended Data Fig. 9i**), possibly reflecting a more general role in inhibiting additional dedifferentiation (and subsequent aberrant differentiation) beyond ISC. This description has been added to the results:

Further, *PROX1* knockdown in primary tumor organoid OKG146P, but not its metastatic counterpart OKG146Li induces the expression of some fetal genes, suggesting a context-specific role for *PROX1* in regulating entry into the fetal state (**Extended Data Fig. 9i**).

Extended Data Figure 9i | *PROX1* inhibits fetal gene expression in primary tumors. Relative expression, from scRNA-seq data, of the conserved fetal signature in OKG146P and OKG146Li, shControl and shPROX1 organoid lines, 7 d after induction with dox. $p = 0.0537$, t test.

We now use language in the abstract that more clearly focuses on the role of *PROX1* in non-canonical gene expression:

We identify ~~the transcriptional regulator~~ *PROX1* as a transcriptional regulator that acts in the fetal progenitor state to repress non-intestinal lineages ~~in the fetal progenitor state~~, and whose downregulation licenses non-canonical reprogramming during metastatic outgrowth.

Comment 2.3. Given that non-canonical signature is enriched in metastasis and that *PROX1* regulates non-canonical differentiation, the abstract gives the impression that *PROX1* is the key regulator for metastasis. Since metastasis is not affected in *PROX1* KD, it'd help to clarify that *PROX1* loss promotes non-intestinal lineage differentiation but not sufficient to drive metastasis.

We cannot find a claim in the abstract that *PROX1* is sufficient to drive metastasis, and we commented on it explicitly in the main text (original line 402):

Functionally, *PROX1* downregulation is not sufficient to induce outgrowth of liver metastasis in mice transplanted with canonical primary tumor-derived organoids (OKG146P, OKG136P) and followed for 12 weeks, suggesting that *PROX1*-driven non-canonical differentiation cooperates with other phenotypic drivers to promote metastatic outgrowth (**Extended Data Fig. 9 e,f**).

Our data and discussion also highlight the importance of underlying cellular context in enabling *PROX1* activity (**Fig. 6d** and **Extended Data Fig. 9g**). To further clarify the point that *PROX1* inhibits non-intestinal lineage differentiation but this is not sufficient to drive metastasis, we have modified the abstract (see point above) and also add an explicit clarification to the Discussion:

While we show that *PROX1* loss promotes non-intestinal lineage differentiation, downregulation of this single factor is not sufficient to drive metastatic outgrowth of primary tumor-derived organoids within 12 weeks of transplantation to mouse liver. These observations underscore that metastasis is a complex, multi-step process requiring multiple factors.

Referee 3

We are fully satisfied with the revisions and the responses to our comments, and congratulate the authors for this important study.

We are very happy that the reviewer appreciates our study and thank them for their contributions.

References

Quintanal-Villalonga Á, Chan JM, Yu HA, Pe'er D, Sawyers CL, Sen T, Rudin CM. Lineage plasticity in cancer: a shared pathway of therapeutic resistance. *Nat Rev Clin Oncol*. 2020 Jun;17(6):360-371. doi: 10.1038/s41571-020-0340-z. Epub 2020 Mar 9.

de Sousa e Melo F, Kurtova AV, Harnoss JM, Kljavin N, Hoeck JD, Hung J, Anderson JE, Storm EE, Modrusan Z, Koeppen H, Dijkgraaf GJ, Piskol R, de Sauvage FJ. A distinct role for Lgr5⁺ stem cells in primary and metastatic colon cancer. *Nature*. 2017 Mar 29;543(7647):676-680. doi: 10.1038/nature21713. PMID: 28358093.

Fumagalli A, Oost KC, Kester L, Morgner J, Bornes L, Bruens L, Spaargaren L, Azkanaz M, Schelfhorst T, Beerling E, Heinz MC, Postrach D, Seinstra D, Sieuwerts AM, Martens JWM, van der Elst S, van Baalen M, Bhowmick D, Vrisekoop N, Ellenbroek SIJ, Suijkerbuijk SJE, Snippert HJ, van Rheenen J. Plasticity of Lgr5-Negative Cancer Cells Drives Metastasis in Colorectal Cancer. *Cell Stem Cell*. 2020 Apr 2;26(4):569-578.e7. doi: 10.1016/j.stem.2020.02.008. Epub 2020 Mar 12. PMID: 32169167; PMCID: PMC7118369.

Heinz MC, Peters NA, Oost KC, Lindeboom RGH, van Voorthuijsen L, Fumagalli A, van der Net MC, de Medeiros G, Hageman JH, Verlaan-Klink I, Borel Rinkes IHM, Liberali P, Gloerich M, van Rheenen J, Vermeulen M, Kranenburg O, Snippert HJG. Liver Colonization by Colorectal Cancer Metastases Requires YAP-Controlled Plasticity at the Micrometastatic Stage. *Cancer Res*. 2022 May 16;82(10):1953-1968. doi: 10.1158/0008-5472.CAN-21-0933. PMID: 35570706; PMCID: PMC9381095.

Reviewer Reports on the Second Revision:

Referees' comments:

Referee #1 (Remarks to the Author):

The authors have satisfactorily addressed my previous concerns. The manuscript now provides quantitative analysis supporting the notion that the fetal gene program precedes non-canonical states rather than co-emerging with these signatures (Fig. 4b and Extended Data Fig. 7a). The concerns previously raised about the mechanistic importance of the PROX1 data persist, but the authors have clearly articulated the limitations of the study and avoid any ambiguity. Overall, I support publication of the manuscript.

Specific comments:

In the new Fig. 4b and Extended Data Fig. 7a, the color scheme for each line is not clear. If the line on the left is meant to represent the 75th percentile for the fetal signature and the line on the right the non-canonical module for each patient, it would be better to distinguish these two with closed and open arrowheads.

It is unclear to me how one should interpret Figure 4d-f. The clearest explanation comes from the rebuttal, which states the following: "The overlap plots (Fig. 4d-f) in particular show that only the fetal state contains both (1) cells that substantially co-express fetal and canonical, and (2) cells that substantially co-express fetal and non-canonical programs."

However, upon visual inspection of the heatmaps, it's not clear how the authors arrive at this conclusion. For instance, in the left and middle panels, the squamous and neuroendocrine states contain similar expression of canonical genes compared to the fetal state based on color intensity. Therefore, it's not clear why the authors state the fetal state is the only module with canonical and non-canonical features. Although, it should be noted that the data from the Wang cohort is more convincing in this regard

Thus the significance of this figure is questionable and could be explained better in the text or removed entirely. Actually, the UMAP plots shown in Figure R2 seem to better support the notion that fetal cells act as a bridge.

Author Rebuttals to Second Revision:

Thank you for sharing the referee's comments. We are delighted that they feel we have satisfactorily addressed their previous concerns and support publication of the manuscript.

Responses to specific comments in line below:

Referee #1:

The authors have satisfactorily addressed my previous concerns. The manuscript now provides quantitative analysis supporting the notion that the fetal gene program precedes non-canonical states rather than co-emerging with these signatures (Fig. 4b and Extended Data Fig. 7a). The concerns previously raised about the mechanistic importance of the PROX1 data persist, but the authors have clearly articulated the limitations of the study and avoid any ambiguity. Overall, I support publication of the manuscript.

Specific comments:

In the new Fig. 4b and Extended Data Fig. 7a, the color scheme for each line is not clear. If the line on the left is meant to represent the 75th percentile for the fetal signature and the line on the right the non-canonical module for each patient, it would be better to distinguish these two with closed and open arrowheads.

We agree that the colors are not very clear. We appreciate and will incorporate the reviewer's suggestion to include closed/open arrowheads to indicate the 75th percentile lines for the fetal signature and non-canonical modules.

Fig ED 7a, Top, trends in gene module scores along DC1, observed in all tumor cells from patients KG152, KG180, KG183, and patient s1321 from Wang et al. (Methods). Each row depicts the module score along DC1 from the 20th percentile value (white) to the maximum value

(highest saturation). Expression of the fetal signature peaks before non-canonical trends in all cases (vertical lines with closed arrows and open arrows correspond to 75th percentile for the fetal signature and predominant terminal non-canonical module for each patient, respectively). Bottom, positions of tumor cells along DC1.

It is unclear to me how one should interpret Figure 4d-f. The clearest explanation comes from the rebuttal, which states the following: “The overlap plots (Fig. 4d–f) in particular show that only the fetal state contains both (1) cells that substantially co-express fetal and canonical, and (2) cells that substantially co-express fetal and non-canonical programs.”

However, upon visual inspection of the heatmaps, it’s not clear how the authors arrive at this conclusion. For instance, in the left and middle panels, the squamous and neuroendocrine states contain similar expression of canonical genes compared to the fetal state based on color intensity. Therefore, it’s not clear why the authors state the fetal state is the only module with canonical and non-canonical features. Although, it should be noted that the data from the Wang cohort is more convincing in this regard

Thus the significance of this figure is questionable and could be explained better in the text or removed entirely.

We would like to make the following clarifications regarding Fig 4d-f. First, these overlap plots (contrary to the UMAPs) are the most definitive way to show a progression without actual lineage tracing (which we obviously cannot have in clinical samples). We agree with the reviewer that we could have perhaps better clarified some things and to this effect make some clarifications in the methods text. To elaborate here more fully:

In our original submission, we only included the hotspot modules in the overlap plot, adding the fetal state in the revision to visually remind the reader of the connection between the endoderm hotspot module and fetal state. This connection was best demonstrated in Fig ED7e (to the right):

Before providing full quantification, we make the following two notes about comparability of these numbers:

- 1) Only the samples included in Fig 4d and 4f include enough cells per sample in the relevant non-canonical states for robust quantitative comparison. Fig 4e, including cells from all patients is included to show similar trends, but its quantification needs to be taken with a grain of salt due to inter-patient heterogeneity and limited non-canonical cells in many patients.

- 2) The way in which hotspot module scores are computed is more quantitatively comparable than the way the fetal signature was scored. Specifically, the computation of hotspot module scores normalizes gene expression in a manner that requires that the entire module is robustly expressed to score in high percentiles. In comparison, the “score_genes” function used to quantify the fetal gene signature is sensitive to high expression of only a small subset of genes within the signature, something that can easily occur given the great deal of heterogeneity typically observed in patient data. This difference gets more pronounced as we increase the percentile.

Bearing those things in mind, in the figures below, we quantify the fraction of cells highly expressing each module/state that overlap with expression of the Tumor ISC-like module at different percentile thresholds. For the 4 patients included in Fig 4d, we see that the overlap of the Endoderm Development module with the canonical tumor ISC module is nearly double that of the non-canonical squamous and neuroendocrine modules. Moreover the overlap of fetal signature with the Tumor ISC module is also clearly higher than that of the squamous/neuroendocrine modules. Importantly this difference is drastically enhanced as the percentile threshold is increased.

Generating this same plot now for all patients we see the same trends hold (plot on left), especially as we increase the percentile threshold, albeit for all patients this is less pronounced due to point 1 above. Finally, these differences are most pronounced in the independent Wang cohort (plot on right), providing quantitative support for our claims.

Thus, we demonstrate a clear and robust quantitative shift in overlap observed across states in multiple tumors with non-canonical states, across our entire patient cohort and in the Wang cohort, providing clear evidence of the sequential progression from differentiated intestinal to ISC to fetal and eventually non-canonical states. This analysis supports Fig 4b and Fig ED7a where we show the sequential progression of states in individual samples.

Importantly, we note that the main text of the manuscript related to **Fig 4d-f** (quoted below) makes measured claims focusing on the evidence for ordered cell state progression consistent with the data presented.

“To study progression in the remainder of the cohort, we took an alternative data analysis approach, reasoning that a large fraction of cells co-expressing two Hotspot modules would suggest a pseudo-ordering of the two cell states and a transition between them. Co-occurrence analysis of all Hotspot module pairs revealed the same step-wise progression across all patients in the cohort (**Fig. 4d,e**); moreover, the trend is replicated in the independent Wang et al. dataset of 5 matched primary and metastatic CRC tumors (**Fig. 4f**). Our data thus suggests that cancer progression involves developmental reversion, characterized by a more primitive dedifferentiated state than that observed in first trimester colonic progenitors, with profound loss of intestinal lineage identity and upregulation of early developmental programs associated with WNT signaling.”

In the methods section, line 1485, we now clarify:

“Additionally, to position the fetal state along this ordering, we performed this overlap computation for cells whose gene set score calculated on the 14 gene ‘core’ fetal signature was greater than the 75th percentile.”

While we include the quantitative plots here to further clarify the nature of our analysis for the reviewer, we feel that Fig 4d-f clearly make the point about sequential cell state transitions in multiple cohorts. The figures and this point have been clear to both other reviewers, as well as to diverse international audiences to whom we have widely presented this work. Given the size and complexity of our manuscript, we favor refraining from including the quantitative analysis plots in the manuscript, but are happy to include them together with the more detailed explanations of the analysis above, should this be deemed necessary by the editor.

Actually, the UMAP plots shown in Figure R2 seem to better support the notion that fetal cells act as a bridge.

Finally, we note that the UMAPs shown in Figure R2 do not rigorously support the notion that fetal cells act as a bridge and hence were only included as a reviewer figure at the reviewer’s request and not included in the final manuscript. Inferring trajectories, connectivity, or potential transitions based on proximity in any 2D projection such as UMAPs can at times be highly misleading. 2D projections often distort the space and can collapse very distant and unrelated cell states into appearing nearby. While such plots can be seemingly intuitive, over-interpretation of distances in 2D projections is a common source of mistakes and misleading claims in single cell data analysis.